Doubling the known diversity of a remote island fauna: marine bivalves of the Juan Fernández and Desventuradas oceanic archipelagos (Southeastern Pacific Ocean)

Zelaya Diego Gabriel 1 2
Güller Marina 1 2 mariguller@gmail.com
http://orcid.org/0000-0002-9554-1947 Bieler Rüdiger 3
1 Consejo Nacional de Investigaciones Científicas y Técnicas (CONICET) , Buenos Aires , Argentina
2 Universidad de Buenos Aires, Facultad de Ciencias Exactas y Naturales, Departamento Biodiversidad y Biología Experimental , Buenos Aires , Argentina
3 Integrative Research Center, Field Museum of Natural History , Chicago , United States of America
Oehlmann Jörg
Electronic publication date: 2024 Jun 28
Publication date: 2024
Volume: 12
Electronic Location ID: e17305
Received 2024 Feb 1; Accepted 2024 Apr 4
Copyright: © 2024 Zelaya et al.
Copyright year: 2024
Copyright holder: Zelaya et al.
License: This is an open access article distributed under the terms of the Creative Commons Attribution License, which permits unrestricted use, distribution, reproduction and adaptation in any medium and for any purpose provided that it is properly attributed. For attribution, the original author(s), title, publication source (PeerJ) and either DOI or URL of the article must be cited.
License URL: https://creativecommons.org/licenses/by/4.0/

Keywords: Biodiversity, Endemism, Expeditions, Marine protected areas, Mollusca, Collections, Oceanic islands, Chile, Bivalvia

Funding: National Geographic 5257-96 Encyclopedia of Life U.S. National Science Foundation (NSF) DEB-0732854 Bivalve-Tree-of-Life (BivAToL) Field Museum Bass Senior Visiting Fellowship The IOC97 cruise received support from National Geographic grant 5257-96 (to Mark Westneat and Brian Dyer) and funding from The John D. and Catherine T. MacArthur Foundation for the Encyclopedia of Life (via Mark Westneat). Relevant bivalve research at the Field Museum was supported by the U.S. National Science Foundation (NSF) award DEB-0732854 to Rüdiger Bieler for the Bivalve-Tree-of-Life (BivAToL) project. Research visits by Diego Gabriel Zelaya and Marina Güller to FMNH were funded by a Field Museum Bass Senior Visiting Fellowship. The funders had no role in study design, data collection and analysis, decision to publish, or preparation of the manuscript.

==============================
Juan Fernández and Desventuradas are two oceanic archipelagos located in the southeastern Pacific Ocean far off the Chilean coast that received protected status as marine parks in 2016. Remoteness and access difficulty contributed to historically poor biodiversity sampling and limited associated research. This is particularly noticeable for bivalves, with most prior regional publications focused on single taxa or un-illustrated checklists. This study investigates marine bivalves collected between the intertidal and 415 m depth during (1) the 1997 IOC97 expedition aboard the M/V Carlos Porter, with special focus on scuba-collected micro-mollusks of both archipelagos, (2) two expeditions by the R/V Anton Bruun (Cruise 12/1965 and Cruise 17/1966), and (3) Cruise 21 of USNS Eltanin under the United States Antarctic Program, which sampled at Juan Fernández in 1965. Also, relevant historical material of the British H.M.S. Challenger Expedition (1873–1876), the Swedish Pacific Expedition (1916–1917), and by German zoologist Ludwig H. Plate (1893–1895) is critically revised. A total of 48 species are recognized and illustrated, including 19 new species (described herein) and six other potentially new species. The presence of two species mentioned in the literature for the region (Aulacomya atra and Saccella cuneata) could not be confirmed. The genera Verticipronus and Halonympha are reported for the first time from the Eastern Pacific, as are Anadara and Condylocardia from Chilean waters. Lectotypes are designated for Arca (Barbatia) platei and Mytilus algosus. These findings double the number of extant bivalve species known from the Juan Fernández and Desventuradas archipelagos, highlighting the lack of attention these islands groups have received in the past. A high percentage of species endemic to one or both archipelagos are recognized herein, accounting for almost 78% of the total. The newly recognized level of bivalve endemism supports the consideration of Juan Fernández and Desventuradas as two different biogeographic units (Provinces or Ecoregions) of the Eastern Pacific Ocean.

Introduction

Recent decades have seen international efforts that greatly improved our knowledge of the global marine fauna, by intensifying sampling in previously understudied regions and by formally inventorying the species-level diversity (e.g., Mora et al., 2011; Appeltans et al., 2012; Kennedy et al., 2019; WoRMS Editorial Board, 2024). Much work remains to be done, and the faunas of many regions are still imperfectly known (Costello et al., 2010). Among these lesser-known regions are the remote Juan Fernández and Desventuradas archipelagos of the Southeast Pacific Ocean. The islands are volcanic seamounts resulting from hotspots of the oceanic Nazca Plate (Bello-González, Contreras-Reyes & Arriagada, 2018), with ages spanning about 1–4 mya (Stuessy et al., 1984; González-Ferrán, 1987; Philippi & Rodrigo, 2020). Far offshore the Chilean coast, the groups are under the influence of subtropical and subantarctic waters (Dyer & Westneat, 2010). Juan Fernández is located about 650 km west of Valparaíso, Chile (33°S), and is composed of three main islands: Robinson Crusoe (=Más a Tierra), Alejando Selkirk (=Más Afuera), and Santa Clara, together with several smaller islets. The Desventuradas, about 840 km west of Antofagasta Chile (26°S), are comprised of the island of San Ambrosio and the San Félix group, the latter consisting of San Félix, Islote González, and Roca Catedral de Peterborough. The Juan Fernández archipelago is sparsely inhabited and nowadays reliant on fishing and tourism; the Desventuradas have no civilian population but host a small detachment of the Chilean navy. The two archipelagoes, together with Salas y Gómez and Rapa Nui farther west in the Pacific, constitute Insular Chile (Islas Oceánicas Chilenas, IOC) and administratively belong to Chile’s Valparaiso Region. Both archipelagoes have been under formal protection since 2016, when marine parks were established for the Juan Fernández archipelago, covering the Crusoe and Selkirk Seamounts and a network of marine parks named “Lobería Selkirk”, “El Arenal”, “Tierra Blanca”, and “El Palillo”. At the same time, the Desventuras became part of the Nazca-Desventuradas Marine Park, the largest such park of Chile (Sernapesca (Servicio Nacional de Pesca y Acuicultura), 2021).

The scientific study of Juan Fernández marine fauna began in the 19th century. The majority of studies focused on fishes (Pequeño & Sáez, 2000; Pequeño & Lamilla, 2000; Dyer & Westneat, 2010; Ramírez et al., 2013; and references therein) and comparatively few studies have addressed mollusks and other invertebrates of the region. Bivalve research in the region has a particularly sparse history: Juan Fernández became a resupply and rest stop for sailing ships in the 18th and 19th centuries. During such visits, some specimens were collected by captains and crews and subsequently described in British publications. One such bivalve example is Arca angulata King, 1832 (now Arca fernandezensis Hertlein & Strong, 1943), described from “a collection formed by the Officers of H.M.S. Adventure and Beagle employed between the years 1826 and 1830 in surveying the southern coasts of South America” (King, 1832: 332). The first comprehensive bivalve data from the archipelago stem from the deep-water materials collected during the British H.M.S. Challenger Expedition (1873–1876) (published by Smith, 1885) and from intertidal and shallow-subtidal collections made by the German zoologist Ludwig H. Plate (1893–1895) (published by Stempell, 1899). Summarizing the prior work, Dall (1909) recognized nine marine bivalve species from Juan Fernández. Collecting by the Swedish Pacific Expedition (1916–1917) led to additional species and other data provided by Odhner (1922). In the second half of the 20th century, Stuardo, Saelzer & Rosende (1970), Villarroel (1971), Roth (1975), Osorio (1979), Villarroel & Stuardo (1998), and Coan (2000) described some individual new species and provided additional distributional records for others. Rozbaczylo & Castilla (1987) compiled the then-available information, reporting 14 species of bivalves from Juan Fernández archipelago. Adding information from specimens obtained during the Southeast Pacific Biological Oceanographic Project (SEPBOP) Anton Bruun cruise 17 of 1966 (and apparently also including material from the Anton Bruun cruise 12 of 1965, see below), Bernard, McKinnell & Jamieson (1991) extended this number to a list of 31 named and unnamed species, without however providing any descriptions or illustrations. As will be shown below, some identifications by Bernard, McKinnell & Jamieson (1991) were surprisingly incorrect (even at the family or order levels), which likely is explained by the fact that the work was published after the death of the lead author and might have been based on unfinished work.

Without a natural harbor or permanent freshwater sources, the Desventuradas archipelago has seen even less attention; it is considered one of the least explored sites in the Pacific Ocean (National Geographic Society, 2013). Published information on its bivalve fauna is limited to a listing of 13 described and undescribed species by Bernard, McKinnell & Jamieson (1991), based on a preliminary analysis of Anton Bruun cruise 17, and two species listed by Tapia-Guerra et al. (2021: table S2).

In this context, it is not surprising that the marine fauna of Juan Fernández and Desventuradas has been widely regarded as poorly known (e.g., Rozbaczylo & Castilla, 1987; Bernard, McKinnell & Jamieson, 1991; Ramírez & Osorio, 2000; Vargas-Gaete et al., 2014). To improve our understanding of the zoological diversity of both archipelagoes, an expedition (IOC-97) was launched in 1997 as a collaborative effort between the Field Museum of Natural History in Chicago and three Chilean institutions, the Universidad del Mar (Viña del Mar), Museo Nacional de Historia Natural (Santiago), and Universidad Austral de Chile (Valdivia). Special focus of IOC97 was on collecting fishes, with the senior author of this article (RB) focusing on marine invertebrates, particularly mollusks. The results of the fish survey were published by Dyer & Westneat (2010). The analysis of the extensive molluscan material obtained during IOC-97 remained hampered by limited comparative knowledge of the Chilean mainland coastal fauna. For Bivalvia, this information became available with the work of Valentich-Scott, Coan & Zelaya (2020).

The objective of this contribution is to refine the current knowledge of the bivalve fauna from the Juan Fernández and Desventuradas archipelagos and, based on that information, to discuss the endemicity and faunistic affinities of these areas.

Materials and Methods

Specimen sources and station data

The majority of the material studied herein stems from four expeditions for which the material has been deposited in the collections of FMNH, MNHN-CL, LACM, and USNM; see acronyms, below (Fig. 1). A Supplemental File provides detailed location and accession data for the expedition material studied herein.

Figure 1 Bivalve collection sites in Juan Fernández and Desventuradas archipelagos.

Symbols correspond to stations of IOC-97 expedition (circles), R/V Anton Bruun 12 and 17 cruises (triangles) and additional museum material and bibliographic records (diamonds). (Manually designed using Ocean Basemap (Esri) and Bathymetric contours (GEBCO_2023) from the U.S. National Centers for Environmental Information Bathymetric Data Viewer, https://www.ncei.noaa.gov/maps/bathymetry).

IOC97

The IOC97 (Islas Oceánicas de Chile, 1997) expedition was undertaken aboard the M/V Carlos Porter of the Chilean Instituto Fomento Pesquero (IFOP) for the period of February 17 to March 12, 1997. The following IOC97 stations resulted in bivalve samples investigated for this study (all by Rüdiger Bieler, scuba, unless otherwise noted):

IOC97-12: 26°20′05″ S, 79°53′25″ W, Caleta Las Moscas, N coast of San Ambrosio Island, Desventuradas, 10–16 m; rocky slope with volcanic boulders and rocks bordering on sand plain, some large macroalgae; 21 February 1997.

IOC97-13: [same site/date as IOC97-12]; sediment at base of rock wall in 18.2 m depth.

IOC97-16: 26°20′18″ S, 79°53′54″ W, between Caleta Patos and Punta Saliente, in shallow inlet, NE side of San Ambrosio Island, Desventuradas, 5–19 m; rock cliff dropping to sea, volcanic rock rubble and large macroalgae (kelp); 22 February 1997.

IOC97-18: 26°20′20.23″ S, 79°53′52.04″ W, Caleta Potal, N coast of San Ambrosio Island, Desventuradas, intertidal to 10 m; volcanic rock rubble, macroalgae, some sand; 23 February 1997.

IOC97-22: 26°17′23″ S, 80°06′34″ W, N coast of San Félix Island, Desventuradas, 10–14 m; volcanic rock, rubble, macroalgae; 24 February 1997.

IOC97-26: 26°17′15.18″ S, 80°6′44.16″ W, between caves and Roca Catedral de Peterborough, NW of San Félix Island, Desventuradas, 25–35 m; 25 February 1997.

IOC97-29: 26°17′28.19″ S, 80°6′37.77″ W; N coast of San Félix Island, Desventuradas, 20–25 m; rock, sand, macroalgae; 26 February 1997.

IOC97-30: 26°17′24.14″ S, 80°6′36.22″ W, N coast of San Félix Island, Desventuradas, intertidal to 12.2 m; 26 February 1997.

IOC97-30A: [same site as IOC97-30] sediment from bottom collected in 12.2 m; 26 February 1997.

IOC97-32: 26°17′9.50″ S, 80°6′7.85″ W; W of Punta Negra, N coast of San Félix Island, Desventuradas, intertidal to 20 m; large volcanic rocks, some sand pockets; 27 February 1997.

IOC97-39: 26°16′52.86″ S, 80°6′48″ W; between San Félix Island and Roca Catedral de Peterborough, Desventuradas, 40–110 m; sandy bottom; collected by IOC-97 team using bottom trawl; 28 February 1997.

IOC97-44: 33°38′27.6″ S, 78°49′22.8″ W; NE point of Cumberland Bay, N of Robinson Crusoe Island, Juan Fernández, 1–14 m; volcanic rock reef with sheer walls with much algal coverage; bottom with sand, rubble and algal mats, 4 March 1997.

IOC97-44A: [same site as IOC97-44]; volcanic rocks at bottom (10–14 m) with arcids, sediment from bottom at 14 m; 4 March 1997.

IOC97-48: 33°42′34.8″ S, 80°46′30″ W, about 100 m off Punta Iman, N of Alejandro Selkirk Island, Juan Fernández, 18–21 m; rocky bottom with some large volcanic boulders and sand pockets; 5 March 1997.

IOC97-48A: [same site/date as IOC97-48]; macromollusks from rocks, sediment in sand pockets at 22 m.

IOC97-50: 33°45′37.14″ S, 80°45′8.1″ W; east side of Alejandro Selkirk Island, Juan Fernández, 20–25 m; rocky bottom, sparse algal coverage, some large (3 m) volcanic boulders, sand pockets; 6 March 1997.

IOC97-50A: [same site/date as IOC97-50]; sediment samples from sand pockets in 25 m depth.

IOC97-57: [same site as IOC97-44]; in 1–10 m depth; 7 March 1997.

IOC97-57A: [same site/date as IOC97-57]; sediment from 7.3 m depth.

IOC97-59: 33°38′28.21″ S, 78°49′6.83″ W, E of Cumberland Bay, N coast of Robinson Crusoe Island, Juan Fernández, 2–25 m; collected by IOC-97 team by scuba; 7 March 1997.

IOC97-62: 33°36′01″–33°36′12″ S, 78°53′42″–78°53′08″ W; W side of N tip of Robinson Crusoe Island, Juan Fernández, 50–80 m; collected by IOC-97 team using bottom trawl; 8 March 1997.

IOC97-64: [same site as IOC97-62].

IOC97-66A: 33°40′20″ S, 78°56′27″ W, off S shore of Bahia Padres, SW point of Robinson Crusoe Island, Juan Fernández, 17–18 m; volcanic rock, sand patches, algae; sediment sample; 8 March 1997.

IOC97-67: 33°38′25.96″ S, 78°49′18.73″ W, NE point of Cumberland Bay, N coast of Robinson Crusoe Island, Juan Fernández, 20–25 m; volcanic rocks, algal mats, sand pockets; 9 March 1997.

IOC97-68A: [same site/date as IOC97-67] sediment from sand pockets in 24.7 m depth; 9 March 1997.

R/V Anton Bruun Cruise 12 (December 1965)

R/V Anton Bruun Cruise 12 focused on the flora and fauna of San Félix and Juan Fernández Islands, with Robert L. Wisner serving as chief scientist (Wisner, 1966). The party spent 3 days (4–7 December 1965) at San Félix, with five scuba-based fish poison stations (9–35 m), 1 otter trawl (75 m) and many hook & line (fish) collections. It spent 6 days (10–16 December 1965) at Juan Fernández, conducting nine scuba poison stations (3–30 m), eight otter trawls (80–200 m), one beam trawl haul (135 m), one Isaacs-Kidd midwater trawl haul (530 m), and three free vehicle set lines (1,100–2,400 m). “No appraisal of the invertebrate fauna was made at the time at either island, but the collections have been sent to the U.S.” (Wisner, 1966: 6). The cruise report does not provide station details for the activities by scuba, otter [=40-foot shrimp] trawl, or beam [=5-foot double-beam with 1/8 inch mesh] trawl (see lots USNM 846082 and 880644 discussed herein). Specimens from 10 collecting events were obtained from LACM for this study, with data taken from collection labels and enhanced with information in Wisner (1966) and from data on USNM collection labels:

LACM 1965-94: “26°17′ S, 80°05′ W”, NW side of San Félix Island, Desventuradas, intertidal; leg. Diane E. Robbins, Janet Haig, John Hall, David Wallen; 5 December 1965 [stated coordinates too far east and not matching described intertidal].

LACM 1965-95: “26°17.5′ S, 80°05.6′ W”, NW side of San Félix Island, Desventuradas, 10–45 ft [3–13.7 m]; leg. Sylvia E. Taylor, Alan Chapman; 5–6 December 1965 [stated coordinates too far inland].

LACM 1965-96: 33°38′ S, 78°49′ W, Cumberland Bay, Robinson Crusoe Island, Juan Fernández, intertidal; leg. Diane E. Robbins; 11–12 December 1965.

LACM 1965-97: 33°38′ S, 78°49′ W, Cumberland Bay, Robinson Crusoe Island, Juan Fernández, 10–30 ft [3–9.1 m]; leg. Sylvia E. Taylor; 11 December 1965.

LACM 1965-98: 33°37.5′ S, 78°49.7′ W, off Cumberland Bay, Robinson Crusoe Island, Juan Fernández, trawled (150 m); “station MV65-IV-47”; 12 December 1965 [depth information from lot USNM 76215].

LACM 1965-99: 33°37′18″ S, 78°50′20″ W, 0.5 mi NW of San Carlos Point, Robinson Crusoe Island, Juan Fernández, 30–70 ft [9.1–21.3 m]; rocky, leg. Sylvia E. Taylor, “station 240”; 12 December 1965.

LACM 1965-100: 33°38′ S, 78°49′ W, Cumberland Bay, Robinson Crusoe Island, Juan Fernández, 46 m; “grab 130”; 12 December 1965.

LACM 1965-101: 33°34–41′ S, 78°45–55′ W, off W side of Robinson Crusoe Island, Juan Fernández, 130–180 m; trawled, 13–15 December 1965 [lot USNM 764195 from “station MV65-IV-54” is dated 13 December, USNM 764199 from “station MV65-IV-63” with starting and ending coordinates of 33°41.2′ S, 78°57′ W to 33°40.7′S, 78°51.8′ W is dated 15 December 1965, and USNM 679522-23 from “station 255, 130–170 m” are dated 15 December 1965].

LACM 1965-102: “approximately 33°38′ S, 78°49′ W”, Chamelo Point, Robinson Crusoe Island, Juan Fernández, intertidal; 16 December 1965.

LACM 1965-103: “approximately 33°38′ S, 78°49′ W”, SE of Bacalao Point, Robinson Crusoe Island, Juan Fernández, 85 ft [25.9 m]; leg. Sylvia E. Taylor; 16 December 1965.

R/V Anton Bruun Cruise 17 (July 1966)

R/V Anton Bruun Cruise 17 focused on the relationships between benthic organisms and their environments in the region. Bandy (1967) provided the cruise report; Roger R. Seapy and Robert M. Woollacott sampled the benthic invertebrates. The party spent 2 days (11–12 July 1966) at Desventuradas, employing four Campbell grab stations, and 1 each of Ockelmann dredge, Newell dredge, and Menzies trawl. It spent 2 days (17–18 July 1966) at Juan Fernández, conducting 10 Campbell grab stations and one Newell grab station. Specimens from six collecting events were obtained from LACM for this study, with data taken from collection labels and checked against information in Bandy (1967):

LACM 1966-98: 26°20′ S, 80°03′ W, SE off San Félix Island, Desventuradas, 415 m; station 675H, by Campbell grab; 12 July 1966.

LACM 1966-99: 26°20′ S, 80°02′ W, SE off San Félix Island, Desventuradas, 170–160 m; station 676B, by Menzies trawl; 12 July 1966. Bernard.

LACM 1966-100: 33°38′ S, 78°50′ W, off Robinson Crusoe Island, Juan Fernández, 62 m; station 680E, by Campbell grab; 18 July 1966.

LACM 1966-101: 33°38′ S, 78°50′ W, off Robinson Crusoe Island, Juan Fernández, 255 m; station 680H, by Campbell grab; 18 July 1966.

LACM 1966-102: 33°38′ S, 78°48′ W, off Robinson Crusoe Island, Juan Fernández, 188 m; station 680I, by Campbell grab; 18 July 1966.

LACM 1966-103: 33°38′ S, 78°46′ W, off Robinson Crusoe Island, Juan Fernández, 210 m; station 680J, by Campbell grab; 18 July 1966.

USNS Eltanin Cruise 21 (November 1965)

United States Antarctic Program (USAP) cruise 21 of the USNS Eltanin took place from 23 November 1965 to 7 January 1966. Under chief scientist George R. Toney, efforts included sediment-core sampling and heat-flow stations (Sandved, 1966; Houtz, Aitken & Cruises, 1973). Start and end ports were Valparaiso and Punta Arenas, Chile. The early part of the cruise included trawl stations in the Juan Fernández archipelago, with material deposited at the USNM.

Station 21-203: 33°45′00″ S, 80°40′48″ W, Alejandro Selkirk Island, Juan Fernández, 79–91 m, Blake trawl; 26 November 1965.

Station 21-205: 33°43′12″ S, 80°43′12″ W, Alejandro Selkirk Island, Juan Fernández, 128–183 m, Blake trawl; 26 November 1965.

Type deposition

All primary types of new species obtained during the IOC97 expedition have been transferred to the collections of the Museo Nacional de Historia Natural de Chile (MNHN-CL). Details are listed under each species treatment and in the Supplemental Table.

Scanning electron microscopy

The shell material was cleaned of encrustations by immersion in an ultrasonic water bath and examined using scanning electron microscopy (SEM) to observe sculpture and hinge and prodissoconch detail. Specimens to be imaged were mounted on conductive carbon tabs, coated with gold, and examined using a Leo EVO 60 SEM at FMNH. Specimens of Condylocardiidae sp A. were imaged, without prior coating, at low voltage (1 kV) using a Hitachi SU7000 SEM at FMNH.

Other

For the previously unnamed species, only empty shell material was collected, thus providing no opportunity to obtain relevant molecular data under current technology. Our species descriptions therefore focus on morphological characters of the shell as well as organic components of the hinge and periostracum when present.

For the analysis of faunistic affinities, Amygdalum sp., Entodesma sp. and Bathyarca corpulenta were excluded, for reasons explained below.

The electronic version of this article in Portable Document Format (PDF) will represent a published work according to the International Commission on Zoological Nomenclature (ICZN), and hence the new names contained in the electronic version are effectively published under that Code from the electronic edition alone. This published work and the nomenclatural acts it contains have been registered in ZooBank, the online registration system for the ICZN. The ZooBank LSIDs (Life Science Identifiers) can be resolved and the associated information viewed through any standard web browser by appending the LSID to the prefix http://zoobank.org/. The LSID for this publication is: urn:lsid:zoobank.org:pub:571610DE-8F2D-4CB4-B527-8B999F6CB098. The online version of this work is archived and available from the following digital repositories: PeerJ, PubMed Central SCIE, and CLOCKSS.

Museum acronyms

ANSP – Academy of Natural Sciences of Drexel University, Philadelphia, USA

FMNH – Field Museum of Natural History, Chicago, USA

GNM – Göteborgs Naturhistoriska Museum, Gothenburg, Sweden

LACM – Natural History Museum of Los Angeles County, Los Angeles, USA

MCZ – Museum of Comparative Zoology at Harvard University, Cambridge, USA

MHNG – Muséum d’histoire naturelle Genève, Switzerland

MNHN – Muséum national d’Histoire naturelle, Paris, France

MNHN-CL – Museo Nacional de Historia Natural, Santiago, Chile

NHMUK – Natural History Museum, London, United Kingdom

NMV – Museum of Victoria, Melbourne, Australia

NSMT – National Museum of Nature and Science, Tokio, Japan

NZNM – Museum of New Zealand—Te Papa Tongarewa, Wellington, New Zealand

SGO.Pi – Colección de Paleontología de Invertebrados, Museo Nacional de Historia Natural de Chile, Santiago, Chile

SMNH – Swedish Museum of Natural History, Stockholm, Sweden

USNM – National Museum of Natural History-Smithsonian Institution, Washington D.C., USA

ZMB – Museum für Naturkunde Berlin (Zoological Collections), Berlin, Germany

Text conventions

The specimen listings under each newly described species include the entire material seen.

av, avs–associated shell valve/s (a matching pair of valves, dry-preserved; sometimes still in closed condition)

spec, specs–whole specimen/s, wet-preserved

v, vs–single shell valve/s

Results

Taxonomic account of bivalve species

NUCULIDAE

Nucula fernandensis Villarroel, 1971

Fig. 2

Figure 2 Family Nuculidae.

(A–O) Nucula fernandensis Villarroel, 1971; (A–D) LACM 1966-101.1; (E, F) LACM 1966-98.1; (G–L) LACM 1966-100.1; (M) (SEM) FMNH 322327; (N, O) (SEM) FMNH 322329. Scale bars: (A–L, M) = 2 mm; (N, O) = 500 µm.

Nucula fernandensis Villarroel, 1971: 159–171, pl. 1, figs. 1, 1A, 2, 2B; pl. 2, figs. 3, 4; pl. 3, figs. 5, 6, 7A, 7B (line drawings; shell and gross morphology).

Nucula fernandensis, – Cekalovic & Artigas, 1981: 80. Rozbaczylo & Castilla, 1987: 176 [listed only]. Bernard, McKinnell & Jamieson, 1991: 36 (listed only). Valentich-Scott, Coan & Zelaya, 2020: 46, pl. 16.

Nucula (Linucula) fernandensis, – Bernard, 1983: 10 (listed only).

Nucula (Nucula) fernandensis, – Villarroel & Stuardo, 1998: 128–129, figs. 23–26 (stomach, drawing), 99–101 (shell SEM).

Nucula pisum, – Bernard, McKinnell & Jamieson, 1991: 36 (listed only). (Non Sowerby, 1833).

Type locality: 33°35′0″ S, 78°31′2″ W (off Robinson Crusoe Island), Juan Fernández archipelago, 220 m (“220–280 m” fide Villarroel & Stuardo (1998)).

Type material: Two possible paratypes studied herein (MZUC 10387 and MZUC 10388).

Other material studied: Desventuradas: San Félix: LACM 1966-98.1 (2 vs, Figs. 2E, 2F). Juan Fernández: Alejandro Selkirk: IOC97-50A (FMNH 322329: 9 avs, 3 vs, with Figs. 2N, 2O). Robinson Crusoe: IOC97-44A (FMNH 322328: 1 av); IOC97-66A (FMNH 322327: 12 avs, 8 vs, with Fig. 2M); MNHN-CL MOL 101609 ex FMNH 327897: 3 avs); LACM 1966-100.1 (3 avs, 48 vs, with Figs. 2G–2L); LACM 1966-101.1 (1 av, 6 vs, with Figs. 2A–2D); LACM 1966-102.1 (3 vs).

Other published records: Topotypic specimens (Villarroel & Stuardo, 1998).

Distribution: Juan Fernández and Desventuradas archipelagos.

Description: Shell up to 4.5 mm L, triangular to subovate, longer than high, somewhat inflated, inequilateral, solid. Anterior end projected, posterior end abbreviated. Umbo large, high and wide, posteriorly displaced, opisthogyrate. Antero-dorsal margin long, convexly sloping, forming a continuous curve with anterior margin. Ventral margin widely arcuated. Posterior margin flattened. Postero-dorsal margin short, sloping straight. Dissoconch sculptured with numerous, narrow and flat radial riblets and strong, irregular growth disruptions. Radial sculpture stronger on posterior than anterior and median areas. Shell surface shiny, whitish; periostracum yellowish pale. Hinge plate solid, with two series of strong, bluntly pointed teeth, reducing in size toward the umbo. Anterior series composed of 7 to 12 teeth, posterior series of 3 to 6 teeth; both series separated by a minute resilifer. Inner shell surface nacreous. Inner margin finely crenulated.

Comments: The status of the type material of Nucula fernandensis is problematic. In the original description, Villarroel (1971) announced the planned deposition of the holotype in MNHN-CL (no number provided). The specimen was never received at that institution (O. Galvez Herrera in litt., June 2022). The author also stated that an unspecified number of paratypes were present in the MZUC collection (registration under number “4578”) and provided dimensions for 18 of them (Villarroel, 1971: table 1). However, lot 4578 does not appear in the MZUC type catalog (Cekalovic & Artigas, 1981) and could not be located during a personal visit to that collection in March 2013. It is unclear whether the originally published paratype lot was lost or the repository number was erroneous or changed subsequently. Cekalovic & Artigas (1981) reported two other lots as paratypes: MZUC 10387 and MZUC 10388. We had an opportunity to study these lots and their current labels indeed state “paratypes”. The first of these was also mentioned by Villarroel & Stuardo (1998), who, in addition, listed eight other lots as having paratype status: MZUC 4577, 4580, 10295, 10296, 10297, 10298, 10299, 10300. Although topotypic and identified by the original author, the type status of this material is uncertain. Even with these uncertainties, the adequate illustration of this species provided by Villarroel (1971) and Villarroel & Stuardo (1998) makes a neotype designation unnecessary.

The specimens studied herein from the Anton Bruun expedition are indistinguishable from the holotype of N. fernandensis (figured by Villarroel, 1971: pl. 1, figs. 1, 1a). Bernard, McKinnell & Jamieson (1991) listed this species from Juan Fernández under the names N. fernandensis and Nucula pisum Sowerby, 1833. The latter, however, does not occur here. It differs by having a lower and narrower umbo, more delicate and pointed hinge teeth, and a larger resilifer. Villarroel (1971) reported as an additional difference the presence of a gastric caecum in N. fernandensis, which is absent in N. pisum.

The present study provides the first record of N. fernandensis from Desventuradas archipelago.

TINDARIIDAE

Tindaria sanfelixensis n. sp.

Fig. 3

Figure 3 Family Tindariidae.

(A–J) Tindaria sanfelixensis n. sp.; (A, B) Holotype, LACM 3819; (C, D, I, J) Paratypes, LACM 3820; (E–H) Paratype, FMNH 312472. Scale bar: (A–J) = 2 mm.

Type locality: 26°20′ S, 80°03′ W, SE off San Félix Island, Desventuradas, 415 m (R/V Anton Bruun Cruise 17, station 675H, by Campbell grab, 12 July 1966; LACM 1966-98).

Type material: Holotype (LACM 3819: 1 v, Figs. 3A, 3B) and 21 paratypes (LACM 3820: 3 avs, 14 vs, + 2 fragments, with Figs. 3C, 3D, 3I, 3J; FMNH 312472 ex LACM 3820 (2 vs, with Figs. 3E–3H); MNHN-CL MOL 101610 ex LACM 3820 (2 vs), all from type locality.

Distribution: Only known from Desventuradas archipelago.

Diagnosis: Shell ovate, with low umbo and comarginal sculpture increasing in solidness towards ventral margin. Hinge plate narrow, with anterior and posterior series of teeth in contact.

Description: Shell up to 4.8 mm L, ovate, longer than high, inequilateral, moderately solid, somewhat inflated. Anterior end short, posterior end projected. Umbo low, broad, anteriorly located, prosogyrate. Antero-dorsal margin, anterior, ventral and posterior margins forming a continuous curve. Postero-dorsal margin long, almost straight. Lunule and escutcheon indistinct, unmarked. Dissoconch with fine comarginal sculpture, increasing in strength toward ventral margin, where it forms low, regularly separated cords. Shell surface whitish, shiny. Hinge plate narrow, with two series of teeth decreasing in size towards the umbo, where they remain in contact. Anterior series composed of 10 teeth, posterior series of about 18 teeth, more delicate than anterior teeth. Ligament completely external, small, delicate, opisthodetic. Inner shell surface whitish, porcellaneous. Inner shell margin smooth. Pallial sinus entire.

Etymology: Named for the type locality, San Félix Island, Desventuradas; adjective.

Comments: The material studied herein, ranging in shell length from 2.3 to 4.8 mm, was previously identified as Nucula grayi d’Orbigny, 1846 by Bernard, McKinnell & Jamieson (1991), for which Valentich-Scott, Coan & Zelaya (2020) designated a lectotype. It is a member of the Nuculidae, belonging to the genus Ennucula Iredale, 1931. The specimens studied herein differ strikingly from that species and other nuculids by having the anterior and posterior series of hinge teeth in contact (not separated by a resilifer), and by lacking nacre interiorly. They fit the concept of Tindariidae as defined by Valentich-Scott, Coan & Zelaya (2020).

Tindaria salaria Dall, 1908 is the only species of the genus thus far known from Chilean waters. The type material of that species consists of three valves and two shell fragments. Raines & Huber (2012: figs. 3F–3H) figured two of these valves for the first time. Additional photographs of the syntypes are available at the Smithsonian National Museum of Natural History’s site (https://collections.nmnh.si.edu). The new species described herein resembles the two smaller syntypes of T. salaria in general shell outline but differs strikingly by having a narrower hinge plate and more delicate teeth. The largest syntype has a higher shell and a more triangular (“nuculiform”) shell outline than Tindaria sanfelixensis n. sp. To date, Tindaria salaria is restricted to off Salas and Gomez archipelago (Villarroel & Stuardo, 1998; Raines & Huber, 2012).

In general shell outline, Tindaria sanfelixensis n. sp. also resembles the northwestern Atlantic Tindaria amabilis (Dall, 1889), the Antarctic T. antarctica Thiele, 1931, the northeastern Pacific Tindaria compressa Dall, 1908, and the east Indian Tindaria sundaensis Knudsen, 1970. However, the first two species clearly differ from T. sanfelixensis n. sp. by having coarser comarginal sculpture evenly distributed all along the dissoconch; Tindaria compressa differs by having a higher umbo and a large “central” tooth in the left valve, which fits into a deep socket in the right valve (see Knudsen, 1970); and T. sundaensis differs by having a narrow triangular pit between the anterior and posterior series of teeth. In addition, radial sculpture on the anterior and posterior parts of the shell was described for T. antarctica, T. compressa, and T. sundaensis, whereas it is absent in T. sanfelixensis n. sp.

Another Chilean species originally placed in Tindaria is Malletia (Tindaria) virens Dall, 1890. However, based on the anatomical information provided by Villarroel & Stuardo (1998) and the ligament characteristics, Valentich-Scott, Coan & Zelaya (2020) excluded this species from Tindaria and reallocated it to Pseudoneilonella (Neilonellidae).

NUCULANIDAE

Ledella costulata n. sp.

Fig. 4

Figure 4 Family Nuculanidae.

(A, B) Ledella costulata n. sp., holotype, LACM 3821. Scale bar: (A, B) = 1 mm.

Type locality: 26°20′ S, 80°03′ W, SE off San Félix Island, Desventuradas archipelago, 415 m. (R/V Anton Bruun Cruise 17, station 675H, by Campbell grab, 12 July 1966; LACM 66-98).

Type material: Holotype (LACM 3821: 1 v, Figs. 4A, 4B).

Distribution: Only known from Desventuradas archipelago.

Diagnosis: Shell ovate, projected in a long, wide posterior rostrum. Dissoconch with comarginal sculpture, increasing in strength ventrally. Resilifer extending entire hinge plate height.

Description: Shell 3.7 mm L, ovate, longer than high, inequilateral, solid. Anterior end broadly rounded, posterior end rostrate. Umbo stout, slightly anteriorly displaced, opisthogyrate. Antero-dorsal and postero-dorsal margins sloping at similar angles, the anterior one convex, the posterior straight. Ventral margin widely curved, forming a well-marked sinuation before reaching the rostrum. Rostrum bluntly pointed, long, wide, posteriorly directed. Lunule indistinct. Escutcheon narrow, flanked by ridges. Dissoconch sculptured with comarginal lines, increasing in solidness towards the ventral margin, where they originate strong, regularly separated cords. Shell surface whitish, shiny. Hinge plate thick, with two series of teeth decreasing in size towards the umbo. Anterior series composed of nine teeth, posterior series of seven teeth. Anterior and posterior series of teeth separated by a deep, rectangular resilifer, extending for the entire hinge plate height. Inner shell surface whitish, porcellaneous. Inner shell margin smooth. Pallial sinus absent.

Etymology: Latin costulatus, -a, -um, bearing small ribs; adjective.

Comments: In general shell outline and shell morphology, Ledella costulata n. sp. closely resembles the northeast Atlantic and Mediterranean Ledella messanensis (Jeffreys, 1870) (see https://naturalhistory.museumwales.ac.uk) and the Brazilian Ledella elfica Viegas, Benaim & Absalão, 2014, Ledella legionaria Viegas, Benaim & Absalão, 2014, and Ledella spocki Viegas, Benaim & Absalão, 2014 (as figured in their original descriptions). However, Ledella costulata n. sp. has strong comarginal cords on the ventral part of the dissoconch, a condition different from the almost completely smooth shell surface of all other known species. In addition, Ledella legionaria and L. spocki have much smaller resilifers than L. costulata n. sp.

Saccella cuneata (Sowerby, 1833)

Comments: The only published record of the occurrence of this species in Juan Fernández archipelago comes from 33°35′ S, 78°31′12″ W, 220–280 m (Villarroel & Stuardo, 1998). The material could not be located during a personal visit to the MZUC collection in March 2013.

MYTILIDAE

Amygdalum sp.

Figs. 5A–5D

Figure 5 Family Mytilidae.

(A–D) Amygdalum sp., LACM 1965-98.1. (E–L) Gregariella exilis (Philippi, 1847); (E, F) FMNH 322289; (G–J) FMNH 322290; (K, L) FMNH 322288. (M–T) Modiolus aurum Osorio, 1979, FMNH 322262. (U, V) Perumytilus purpuratus (Lamarck, 1819), MCZ 143489. (W–Z) Semimytilus patagonicus (Hanley, 1843); (W, X) MCZ 250131; (Y, Z) lectotype of Mytilus algosus Gould, 1850 (designated herein), MCZ 216829. Scale bars: (A–D, U, V) = 5 mm; (E–L) = 2 mm; (M–T) = 2 cm; (W–Z) = 10 mm.

Amygdalum americanum, – Bernard, McKinnell & Jamieson, 1991: 36 [listed only].

Material examined: Juan Fernández: Robinson Crusoe: LACM 1965-98.1 (10 avs, 1 v; partially damaged, with dried tissue, with Figs. 5A–5D).

Distribution: Uncertain (see Comments).

Description: Shell to 14.9 mm L, ovate, longer than high, markedly inequilateral, thin. Posterior end higher than anterior end. Umbo close to anterior end, low, rounded. Anterior margin relatively broad, evenly arched with ventral and posterior margins. Dorsal margin long, almost straight. Dissoconch smooth. Shell surface shiny, white, with narrow, widely spaced, anastomosing, brownish radial lines at posterior end. Nepionic shell grayish, hyaline, clearly distinct from the rest of the shell. Periostracum thin. Hinge plate narrow, edentulous. Nymph long, narrow. Inner shell surface whitish. Inner margin smooth.

Comments: Oliver (2001) recognized three informal groups of species in Amygdalum. In general shell outline and color pattern, the Juan Fernández specimens studied herein fit within the group in which Oliver (2001) included the Caribbean Amygdalum dendriticum (Megerle von Mühlfeld, 1811) [regarded by Beu (2004) as a junior synonym of A. arborescens (Fischer von Waldheim, 1807), the type species of the genus], the Indo-Pacific Amygdalum peasei (Newcomb, 1870), the Japanese Amygdalum plumeum (Kuroda & Habe, 1971) (=A. peasei according to Huber (2010) and Raines & Huber (2012)), the Australian Amygdalum beddomei (Iredale, 1924) (a junior synonym of Amygdalum striatum (Hutton, 1873) fide Beu (2004)) and the eastern Pacific Amygdalum americanum (Soot-Ryen, 1955). The Juan Fernández specimens studied herein (previously identified as A. americanum by Bernard, McKinnell & Jamieson, 1991) resemble the similar-sized Japanese specimen figured by Oliver (2001) as Amygdalum plumeum and the Eastern Island specimen figured by Raines & Huber (2012) as Amygdalum peasei. At present, the taxonomy of these nominal species remains unclear. In fact, some authors suggested that all these names could correspond to a single, widely distributed species (see Beu, 2004). Under this scenario, we refer to the material examined herein as Amygdalum sp.

Gregariella exilis (Philippi, 1847)

Figs. 5E–5L

Mytilus exilis Philippi, 1847: 120.

Gregariella chenui, – Soot-Ryen, 1955: 13. Soot-Ryen, 1959: 23 [listed only; “record needs confirmation”]. Bernard, McKinnell & Jamieson, 1991: 36 [listed only]. (Non Récluz, 1842).

Gregariella opifex, – Osorio & Bahamonde, 1970: 192 [listed only]. Rozbaczylo & Castilla, 1987: 176 [listed only]. (Non Say, 1825, non Philippi, 1847).

Modiolaria (Gregariella) opifex, – Odhner, 1922: 221. (Non Say, 1825, non Philippi, 1847).

Mytilus exilis, – Coan & Kabat, 2017: 49 [information on type material].

Gregariella coarctata, – Valentich-Scott, Coan & Zelaya, 2020: 109, pl. 36. (Non Carpenter, 1857).

Type locality: “Orae Chilensis et Peruvianae.” (Chilean and Peruvian coasts).

Type material: One syntype of Mytilus exilis studied herein (MNHN-CL 100031).

Other material studied: Desventuradas: San Félix: IOC97-26 (FMNH 322294: 1 av, 20 vs; MNHN-CL MOL 101611 ex FMNH 327898: 20 vs); IOC97-29 (FMNH 322288: 20 vs, with Figs. 5K, 5L); IOC97-30 (FMNH 322295: 1 v); IOC97-32 (FMNH 322296: 3 vs); LACM 1965-95.1(3 avs, 5 vs; with dried tissue); LACM 1966-98.2 (fragments); LACM 1966-99.1 (1 v juv., fragments). San Ambrosio: IOC97-12 (FMNH 322291: 1 v, 1 fragment; juvenile); IOC97-13 (FMNH 322292: 5 vs, juvenile); IOC97-18 (FMNH 322293: 1 fragment). Juan Fernández: Alejandro Selkirk: IOC97-48A (FMNH 322297: 1 v, juvenile); IOC97-50A (FMNH 322299: 2 vs). Robinson Crusoe: IOC97-44A (FMNH 322289: 4 vs, with Figs. 5E, 5F); IOC97-57A (FMNH 322298: 1 v); IOC97-68A (FMNH 322290: 5 vs, with Figs. 5G–5J); LACM 1965-100.1 (1 v); LACM 1966-100.2 (3 vs); “Masatierra”, 30–45 m (SMNH 1227: 4 spec, mentioned by Odhner (1922) as Modiolaria (Gregariella) opifex Philippi).

Other published records: Bahía Cumberland, Juan Fernández (Valentich-Scott, Coan & Zelaya, 2020).

Distribution: Only known with certainty from Juan Fernández and Desventuradas archipelagos. The occurrence of this species in Perú and Chile mainland (from where the specimens described by Philippi (1847) apparently came) is uncertain.

Description: Shell up to 14 mm L, trapezoidal to subovate, longer than high, inflated, markedly inequilateral, thin. Posterior end higher than anterior end. Umbo very broad, low, almost at anterior end, dorsally located. Anterior margin extremely short, slightly arcuated. Ventral margin straight to markedly sinuated by byssal embayment, which is associated with a median sulcus along outer shell surface. Posterior margin gently curved. Dorsal margin straight to slightly convex. Posterior area of shell wide, limited by an obscure fold running from the umbo to the junction of posterior and ventral margins. Dissoconch sculptured with prominent comarginal folds and numerous, fine but solid radial riblets anterior and posteriorly. Radial sculpture of the posterior area densely packed, stronger than that of anterior area, forming small granulations in the intersection with comarginal sculpture. Median area lacking radial sculpture. Shell surface whitish, iridescent in smaller specimens, dull in larger, well-preserved specimens, purple iridescent in eroded specimens. Periostracum thick, dehiscent, yellowish to dark brown, projected in numerous, long, branched setae on the posterior area. Hinge thickened anterior to the umbo; posteriorly narrow, slightly widening distally; with small, dysodont teeth, and 4–8 stronger tubercles at posterior end. Nymph narrow, long. Inner shell surface nacreous. Inner margin finely crenulated at anterior, ventral and ventral part of posterior margins; dorsal part of posterior margin with stronger crenulations.

Comments: Our study of the Juan Fernández specimens previously mentioned in the literature as well as the material obtained during IOC97 revealed the presence of a single species of Gregariella in this area. However, three different names were previously applied to this entity: “Gregariella opifex Philippi, 1847”, Gregariella chenui (Récluz, 1842), and Gregariella coarctata (Carpenter, 1857). The first was reported by Odhner (1922), who erroneously attributed the authorship of this species to Philippi. As pointed out by Coan & Kabat (2017), the author of G. opifex is Say (1825), who described the species based on a specimen collected at Minorca in the Mediterranean Sea. Philippi (1847) tentatively identified as “Modiola opifex Say” specimens from Brazil. The figure of the material he studied (Philippi, 1847: pl. 2, fig. 7) differs from the specimen illustrated by Say (1825: pl. 9, fig. 2) in having a much shorter and rounded anterior margin, and a considerably lower shell. There is no doubt that the specimens studied by Philippi (1847) belong to Gregariella. However, the identity of Modiola opifex Say is an enigma (Huber, 2010): some authors (e.g., Soot-Ryen, 1955) placed it in Gregariella, whereas others (e.g., Palazzi, 1981; Kleemann, 1983) questioned such generic placement, and Say’s species is currently considered a nomen dubium (MolluscaBase, 2024).

The drawing provided by Philippi (1847: pl. 2, fig. 7) for the Brazilian material clearly differs from the Gregariella species occurring in the Juan Fernández archipelago by having a much narrower posterior area of the shell, a more projected anterior end, and a higher anterior margin, which results in a more dorsally displaced umbo. Soot-Ryen (1959) suggested that Odhner’s (1922) record of Gregariella opifex “Philippi” from Juan Fernández could correspond to the same entity that he (Soot-Ryen, 1955) had identified as Gregariella chenui, a species he reported as ranging from Monterrey, California to Bahía de la Independencia, Perú. However, Coan & Valentich-Scott (2012) excluded this taxon from the Eastern Pacific, restricting its distribution to the western Atlantic, where it ranges from its type locality in Bahía, Brazil, to Florida, USA (Huber, 2010). Coan & Valentich-Scott (2012) recognized for the Eastern Pacific only two valid species of Gregariella: G. denticulata (Dall, 1871) and G. coarctata. These species are readily distinguished by their shell outline, particularly by the posterior margin: G. denticulata has an obliquely straight, rapidly sloping posterior margin, which results in a triangular shell outline, whereas in G. coarctata the posterior margin is convex and more slowly sloping, resulting in an ovate-elongate shell outline. In this context, Valentich-Scott, Coan & Zelaya (2020) identified specimens from Juan Fernández as G. coarctata.

Gregariella coarctata was originally described from Mazatlán, Mexico, and the Galápagos Islands. Specimens from various sites in the regions of the type localities were figured by various authors (e.g., Olsson, 1961: pl. 16, figs. 4–4d; Keen, 1971: fig. 133; Coan & Valentich-Scott, 2012: pl. 41; Hendrickx et al., 2014: fig. 8; López-Rojas et al., 2017: fig. 3F; Valentich-Scott, 1998: fig. 5.19). All these specimens appear conspecific with additional material from the Panamic Province that we studied in museum collections (e.g., FMNH 103022, specimens from Sonora, Mexico). However, all these show a consistently higher anterior shell margin than the Juan Fernández specimens, which results in a more dorsal placement of the umbo. Furthermore, the posterior margin in these specimens is straight or only slightly convex, slopes fast, and results in a pointed posterior end, which is as high as the anterior end. By contrast, in the Juan Fernández specimens the posterior margin is markedly convex and evenly curved, thus resulting in a rounded posterior end, which also is positioned much higher than the anterior end. The significance of these differences is difficult to interpret in the context of the currently imperfect knowledge of Gregariella species, although we expect them to reflect species-level distinction.

The Juan Fernández specimens closely resemble the primary type material of Mytilus exilis Philippi, 1847 (MNHN-CL 100031, syntype), originally described from the Peruvian-Chilean coast. That nominal species was previously considered as a possible synonym of Perumytilus purpuratus (Lamarck, 1819) (e.g., Bernard, 1983; Coan & Valentich-Scott, 2012). However, we consider these two species as distinct and confirm the placement of M. exilis in Gregariella, a placement also suggested by Valentich-Scott, Coan & Zelaya (2020).

Modiolus aurum Osorio, 1979

Figs. 5M–5T

Modiolus aurum Osorio, 1979: 199, figs. 1–7 (shell, gross morphology).

Modiola plumescens Dunker, – Odhner, 1922: 221. Soot-Ryen, 1959: 23 (“records need confirmation”). Osorio & Bahamonde, 1970: 192 (listed only). (See comments on the name Modiola plumescens, below).

Modiolus aurum, – Bernard, 1983: 19 (listed only). Rozbaczylo & Castilla, 1987: 176 (listed only). Bernard, McKinnell & Jamieson, 1991: 36 (listed only). Ramírez & Osorio, 2000: 6 (listed only). Valentich-Scott, Coan & Zelaya, 2020: 115, pl. 39 (holotype and 1 paratype).

Mytilus pilosus, – Stempell, 1899: 221 (specimens from Juan Fernández). Dall, 1909: 258 (listed only). Lamy, 1936–1937: 110–111 (with reference to Stempell and Dall). (Non Reeve, 1858).

Type locality: 33°37′ S, 78°49′ W, Bahía Cumberland, Robinson Crusoe Island (Juan Fernández archipelago).

Type material: Holotype (MNHN-CL 100230), two paratypes (MNHN-CL 100231, MNHN-CL 100232), “one hundred additional paratypes in the author’s [C. Osorio’s] collection”; five paratypes (NMV F 96495) (the latter not mentioned in the original description).

Other material studied: Desventuradas: San Félix: LACM 1965-94.1 (2 vs); LACM 1965-95.2 (2 vs). San Ambrosio: IOC97-16 (FMNH 322265: 2 avs). Juan Fernández: Alejandro Selkirk: Eltanin Station 21-203 (USNM 887909: 6 specs, 23 avs, 2 vs; mostly juvenile). Robinson Crusoe: IOC97-44 (FMNH 322262: 2 avs, Figs. 5M–5T; FMNH 322266: 1 spec); MNHN-CL MOL 101612 ex FMNH 327899: 1 av, 4 vs); IOC97-44A (FMNH 322270: 6 vs, fragments); IOC97-57 (FMNH 322263: 3 avs); IOC97-57A (FMNH 322264: 5 vs); IOC97-66A (FMNH 322269: 3 vs, juvenile); IOC97-68A (FMNH 322268: 5 vs); LACM 1965-96.1 (2 avs); LACM 1965-97.1 (2 avs with dried tissue); LACM 1965-101.2 (7 avs, 1 v; partly with dried tissue); LACM 1966-100.3 (4 vs, juvenile).

Other published records: Bahía Padres, Juan Fernández (Stempell, 1899). “Masatierra”, 30–45 m (Odhner, 1922). Playa “El Palillo” and Playa “El Pangal”, Robinson Crusoe, Juan Fernández; and Alejandro Selkirk Island (Osorio, 1979).

Distribution: Only known from Juan Fernández and Desventuradas archipelagos.

Description: Shell to 38.2 mm L, triangular, longer than high, flat, markedly inequilateral, moderately solid. Posterior end much higher than anterior end. Umbo almost at anterior end, rounded, low, only slightly outstanding from dorsal margin. Anterior margin extremely short, not distinctly separated from ventral margin. Ventral margin straight to variably sinuated by byssal embayment. Posterior margin gently curved, high. Dorsal margin long, straight, oblique, forming an obscure angulation at the junction with posterior margin. Posterior area of shell somewhat depressed. Dissoconch only sculptured with low, irregular growth marks; surface yellowish, dull. Periostracum thick, golden-yellowish to brown; projected in long, simple, pointed or spatuliform setae; the latter kind of setae only present in the dorsal area of larger specimens. Hinge plate extremely narrow, lacking tubercles or any other teeth-elements. Nymph long, extending for about half of the total dorsal margin length. Inner shell surface whitish, somewhat nacreous. Inner margin smooth.

Comments: Stempell (1899) identified as Mytilus pilosus specimens from Juan Fernández and Iquique. None of these specimens could be located at ZMB (C. Zorn in litt., July 2022), where other specimens studied by Stempell are housed. Stempell’s records were subsequently copied by Dall (1909) and Lamy (1936–1937). Most probably, the Juan Fernández records were based on Modiolus aurum, a species still undescribed at the time. Modiolus aurum resembles “Mytilus” pilosus in general shell outline and by having the periostracum projected in setae. However, according to the original description (Reeve, 1857–1858: species 35) M. pilosus strikingly differs by being sculptured with numerous, strong, serrated radial ribs. Mytilus pilosus is currently considered a nomen dubium (Huber, 2015, electronic appendix).

Odhner (1922) identified a specimen collected in Juan Fernández as “Modiola plumescens Dunker, 1868”. Osorio (1979) reported that this specimen was lost and, in view of the great similarities of M. plumescens and Modious aurum, suggested that Odhner’s (1922) record might have been a misidentification. It should be noted that the name Modiola plumescens, although widely used (e.g., Huber, 2010, as “Modiolus plumescens (Dunker, 1868)” was never introduced by Dunker. As discussed by Bieler & Petit (2012: 27, 65), the name was first listed as a nomen nudum by Schmeltz (1864) and formally introduced as Modiola tumescens [sic] by Clessin (1886–1890).

Perumytilus purpuratus (Lamarck, 1819)

Figs. 5U and 5V

Modiola purpurata Lamarck, 1819: 113.

Mytilus ovalis Lamarck, 1819: 121. Hupé, 1854–1858: 312–313 (listed only).

? Mytilus exaratus Philippi, 1847: 119.

Mytilus (Aulacomya) purpuratus, – Stempell, 1899: 226.

Modiolus purpuratus, – Dall, 1909: 258 [listed only].

Brachidontes purpuratus, – Soot-Ryen, 1955: 45, pl. 4, fig. 18 and text fig. 30. Soot-Ryen, 1959: 28 [listed only]. Bernard, 1983 [listed only]. Skoglund, 2001: 16–17 [listed only].

Perumytilus purpuratus, – Olsson, 1961: 117, pl. 12, fig. 1 and pl. 14, figs. 1, 1B. Osorio & Bahamonde, 1970: 192 [listed only]. Marincovich, 1973: 9, Fig. 6. Ramorino & Campos, 1979: 207–218, pls. 1, 2. Lozada & Reyes, 1981: 147–154. Alamo & Valdivieso, 1997: 100, fig. 232. Guzmán, Saá & Ortlieb, 1998: 63–64. Aldea & Valdovinos, 2005: 395, fig. 10D. Pérez-Garcia et al., 2010: 199–205. Coan & Valentich-Scott, 2012: 119, pl. 37. Uribe et al., 2013: 215. Trovant et al., 2015: 60–74. Oyarzún et al., 2016: 375–385 (in part). Paredes et al., 2016: 132. Valentich-Scott, Coan & Zelaya, 2020: 105–107, pl. 35 [neotype].

Figure 6 Family Arcidae.

(A–D) Acar pusilla (Sowerby, 1833), LACM 1965-94.2. (E–J) Acar bernardi n. sp.; (E–F) Holotype, LACM 3822; (G, H) Paratype, LACM 3823; (I, J) Paratype, FMNH 312477. (K, M) Anadara stempelli n. sp., holotype, ZMB 51988a. (N–P) Bathyarca corpulenta (Smith, 1885); Syntype of Arca (Barbatia) corpulenta Smith, 1885, NHMUK 1889.11.11.131. (Q–EE) Tetrarca fernandezensis (Hertlein & Strong, 1943); (Q–U) Syntype of Arca (Arca) fernandezensis Hertlein & Strong, 1943, NHMUK 1969.202; (V) FMNH 322274; (W, X) FMNH 322271; (Y, Z) LACM 10496; (AA–CC) FMNH 322272; (DD, EE) FMNH 322273. Scale bars: (A–J, AA–CC, DD, EE) = 2 mm; (K–M) = 5 mm; (N–P) = 10 mm; (Q–Z) = 2 cm.

Type localities: 9.359° S 78.425° W, Bahía Huaynuna, Ancash, Perú (see Valentich-Scott, Coan & Zelaya, 2020) (Modiola purpurata). Mers du Pérou [Perú] (Mytilus ovalis). Unknown: Mytilus exaratus.

Type material: Neotype of Modiola purpurata (=syntype of Mytilus ovalis) (MHNG-Moll 50635). Holotype of Mytilus exaratus (MNHN-CL 100030).

Material studied: Juan Fernández: Robinson Crusoe: 33°38′09.13″ S, 78°49′22.09″ W, Cumberland Harbor (MCZ 143489: 1 spec, Figs. 5U, 5V).

Distribution: Estero Zarumilla, Tumbes, Perú (03°30′ S) (Dall, 1909) to Coliumo, Bío Bío, Chile (36°36′ S) (Trovant et al., 2015); and Juan Fernández (this study).

Description: Shell to 50 mm L, subovate to subtrigonal, longer than high, inflated, markedly inequilateral, thick. Posterior end higher than anterior end. Umbo at or close to anterior end, broadly rounded, low. Anterior margin extremely short, not distinctly separated from ventral margin. Ventral margin straight to markedly curved. Posterior margin gently curved, high. Dorsal margin long, straight, oblique, forming an angulation at the junction with posterior margin. Dissoconch sculptured with radial ribs and comarginal threads. Radial ribs moderate to strong in central and dorsal areas, much thinner ventrally. Shell surface purple. Periostracum thick, dark brown to black. Hinge plate narrow, with several, small tubercles. Nymph long, strong. Inner shell surface whitish to purple, shiny to silky. Inner margin strongly crenulate.

Comments: Perumytilus purpuratus was historically regarded as a widespread species in South America, occurring from Perú in the Pacific Ocean to Golfo San Matías in the Atlantic Ocean (Prado & Castilla, 2006). However, based on data from three molecular markers (COI, 18S, and 28S), Trovant et al. (2015) recognized two clades: a “North Clade”, restricted to the Peru-Chile Province (in the Pacific coast, north of Coliumo, Bío Bío, Chile (36°36′ S)) and a “South Clade” occurring in the Magellanic Province (south of La Misión, Los Lagos, Chile (39°48′ S) at the Pacific coast, and extending along southern Argentina, at the Atlantic coast). This distinction is consistent with the result of the microsatellite marker used by Pérez et al. (2008) and the sperm polymorphism reported by Briones et al. (2012). Although no morphological differences between northern and southern clades were detected, Valentich-Scott, Coan & Zelaya (2020) restricted the usage of Perumytilus purpuratus to the northern clade, suggesting that a different name might be needed for the southern clade. The two articulated valves studied herein represent the only record of a member of Brachidontinae thus far known from Juan Fernández. These specimens are tentatively identified as P. purpuratus.

Semimytilus patagonicus (Hanley, 1843)

Figs. 5W–5Z

Mytilus patagonicus Hanley, 1842–1846: 236 (based on d’Orbigny MS).

Mytilus patagonicus d’Orbigny, 1834–1848: 646–647; 1847: pl. 85, figs. 12, 13. Clessin, 1886–1890: 82–83, pl. 18, figs. 5, 6. Dall, 1909: 258 [listed only]. Carcelles, 1950: 76, pl. 4, fig. 70. Carcelles & Williamson, 1951: 329 [listed only].

Mytilus algosus Gould, 1846–1850: 344; 1852: 450; 1860: pl. 41, figs. 566, 566a. Johnson, 1964: 39.

Mytilus dactyliformis Hupé, 1854–1858: 310. Hupé, 1854–1858: pl. 5, figs. 6, 6A. Dall, 1909: 258 [listed only]. Carcelles & Williamson, 1951: 329 [listed only].

Mytilus splendens Dunker, 1857: 358. Clessin, 1886–1890: 86. Dall, 1909: 258 [listed only].

Modiola splendens, – Reeve, 1857–1858: Modiola plate 7, species 37.

Mytilus cuneiformis Reeve, 1857–1858: Mytilus plate 5, species 18. Stempell, 1899: 221. (Non Hanley, 1843).

? Mytilus angustanus, – Reeve, 1857–1858: Mytilus plate 9, species 36. Clessin, 1886–1890: 43, pl. 13, figs. 5, 6. (Non Lamarck, 1819).

? Mytilus similis Clessin, 1886–1890: 82, pl. 16, figs. 3, 4. (Non Münster, 1841).

Mytilus edulis patagonicus, – von Ihering, 1907: 411.

Modiola pseudocapensis Lamy, 1931: 305–306.

Modiolus (Modiolus) nonuranus Pilsbry & Olsson, 1935: 16, pl. 1 fig. 3. Olsson, 1961: 115, pl. 17, fig. 10. Clench & Turner, 1962: 105.

Semimytilus algosus, – Soot-Ryen, 1955: 25–26, text figs. 8, 9, 14–16, pl. 4, fig. 17. Soot-Ryen, 1959: 25–26. Olsson, 1961: 114–115, pl. 14, fig. 8. Kensley & Penrith, 1970: 17–20, figs. 2–4. Osorio & Bahamonde, 1970: 191 [listed only]. Marincovich, 1973: 9, fig. 7. Bernard, 1983: 19 [listed only]. Alamo & Valdivieso, 1997: 100, 103, fig. 233. Guzmán, Saá & Ortlieb, 1998: 64. Skoglund, 2001: 16–17 [listed only]. Aldea & Valdovinos, 2005: 395, fig. 10H. Villegas, Stotz & Laudien, 2006: 25–31. Coan & Valentich-Scott, 2012: 120, pl. 38. Uribe et al., 2013: 216. Bigatti, Signorelli & Schwindt, 2014: 241–246. Paredes et al., 2016: 132. Valentich-Scott, Coan & Zelaya, 2020: 101, pl. 33. Ma et al., 2020a: 507–515; 2020b: 1–13.

Semimytilus nonuranus, – Olsson, 1961: 115, pl. 17, fig. 10.

Mytella speciosa, – Bernard, McKinnell & Jamieson, 1991: 36 [listed only]. (Non Reeve, 1857).

Modiolus patagonicus, – Zelaya, 2015: 253.

Semimytilus patagonicus, – Signorelli & Pastorino, 2021a: 55–63; 2021b: 173 [authorship].

Type localities: Patagonia [Hanley, based on d’Orbigny’s manuscript]; elaborated by d’Orbigny as îles de Las Gamas [=Gamma Island, 40°30′ S 062°12′ W] et de los Chanchos [Jabali Island, 40°35′ S 62°12′ W], baie de San-Blas [San Blas Bay, Argentina] (Mytilus patagonicus). Perú (Mytilus algosus; here restricted). “Ad litus Peruanus” (Perú coast) (Mytilus splendens). Bay of Guayaquil [Ecuador] (Mytilus cuneiformis). Walfisch (=Walvis Bay, Namibia) (Modiola pseudocapensis) [fide Huber, 2010]. Beach of Nonura Bay, near Punta Aguja, Peru (Modiolus (Modiolus) nonuranus). Unknown localites: Mytilus dactyliformis; Mytilus similis.

Type material: Mytilus patagonicus: lectotype and four paralectotypes (NHMUK 1854.12.4.805.5). Mytilus algosus: one lectotype (MCZ 216829, designated herein, Figs. 5Y, 5Z), two paralectotypes (MCZ 154352), and one paralectotype (ANSP 55842). Modiola pseudocapensis: three syntypes, Ponta Gea [Mozambique] (MNHN-IM 25767), six syntypes Walfisch [=Walvis Bay, Namibia] (MNHN-IM 25765, MNHN-IM 25766). Modiolus (Modiolus) nonuranus: two syntypes (ANSP 164612) and five syntypes (ANSP 164613). Mytilus dactyliformis, Mytilus splendens, Mytilus cuneiformis, and Mytilus similis: types not found.

Material studied: Juan Fernández: Robinson Crusoe: 33°38′09.13″ S, 78°49′22.09″ W, Cumberland Harbor (MCZ 250131: 2 avs, with Figs. 5W, 5X).

Distribution: In the eastern Pacific from Manta, Manabí, Ecuador (1°S) (Soot-Ryen, 1955) to Maicolpué, Chile (40°35′ S) (Oyarzún et al., 2020), and Juan Fernández (Soot-Ryen, 1955; this study). In the eastern Atlantic (introduced): from Luanda, Luanda, Angola (8.8° S) (Ma et al., 2020b) to Hermanus, Western Cape, South Africa (34.4° S) (Ma et al., 2020a). In the western Atlantic, only mentioned from Bahía San Blas, Buenos Aires, Argentina (40.5° S) (d’Orbigny, 1834–1848) and Puerto Madryn, Chubut, Argentina (42.2°S) (Bigatti, Signorelli & Schwindt, 2014). Introduced in the Indian Ocean, the species was reported from Mozambique (NaGISA Project, 2018).

Description: Shell to 74 mm L, ovate-elongate, longer than high, inflated, markedly inequilateral, thin. Posterior end higher than anterior end. Umbo almost at anterior end, broad, low, only slightly outstanding from dorsal margin. Anterior margin extremely short, not distinctly separated from ventral margin. Ventral margin nearly straight. Posterior margin gently curved, high. Dorsal margin long, straight, oblique, forming an angulation at the junction with posterior margin. Dissoconch only sculptured with thin growth lines; surface whitish to purple. Periostracum thick, light greenish to dark brown. Hinge plate narrow, lacking tubercles or any other teeth-elements. Nymph long, narrow. Inner shell surface whitish to purple, somewhat nacreous. Inner margin smooth.

Comments: Semimytilus algosus was the name predominantly used to refer to this species, until Signorelli & Pastorino (2021a) considered that taxon a junior synonym of Semimytilus patagonicus. The latter was originally described from the Atlantic coast of South America (Bahía San Blas), where, judging from d’Orbigny’s (1834–1848: 647) comment (“elle est excellente à manger”), it appears to have been abundant. However, there are no subsequent records of this species at the Atlantic coast of South America, except for specimens found attached to a squid-fishing vessel during an in-water hull cleaning (Bigatti, Signorelli & Schwindt, 2014). At present, the species is apparently not living in this area (Signorelli & Pastorino, 2021a).

The type locality of Mytilus algosus has been historically unclear: Gould (1846–1850) originally described the species as coming from the “South Seas”, and later (Gould, 1852–1860) specified “Feejee islands” [=Fiji]. Soot-Ryen (1955) regarded M. algosus as a South American species, consequently considering “Fiji” as an incorrect provenance. Considering that the U.S. Exploring Expedition–as part of which the material studied by Gould was collected–visited the surroundings of only two localities in the Southeastern Pacific (Valparaiso in Chile and Callao in Peru), Soot-Ryen (1955) considered “it therefore safe to make Valparaiso the type locality”, but did not indicate the reason why he chose this locality over Callao. Signorelli & Pastorino (2021a) considered the locality selected by Soot-Ryen (1955) as wrong, alternatively correcting it to “Callao, Peru”, based on the provenance of the three syntypes of M. algosus they studied (MCZ 154352: two loose valves; MCZ 216829: one syntype, consisting of two loose valves). However, the only information currently available in the MCZ original book register and specimen lot labels, is “Perú”. In addition, there is another supposed syntype of M. algosus (ANSP 55842), labeled as coming from “Pasco, Perú”, a site in fact visited by some members of the U.S. Exploring Expedition (Wilkes, 1845), although not located on the coast. In view of the prior confusion concerning the exact provenance of the type material, a lectotype is here designated for M. algosus (lot MCZ 216829; figured in Signorelli & Pastorino, 2021a: figs. 3A–3D). This lectotype designation consequently fixes the type locality of the species as “Perú”.

Pilsbry & Olsson (1935) described Modiolus (Modiolus) nonuranus from the “Beach of Nonura Bay, near Punta Aguja, northern Peru (Olsson), type 164612 ANSP; also at Punta Capullana […]. Paratypes in Olsson collection.” They provided dimensions (39.0 and 37.5 mm L) for two specimens, of which one was illustrated. The specimens from Punta Capullana were described as small (up to 30 mm). Signorelli & Pastorino (2021a) described the type series as composed of the “holotype (ANSP 164612) and two paratypes (ANSP 164613, ANSP 164614).” However, ever since the time of cataloging (on 13 February 1935), the lot ANSP 164612 has contained 4 valves (=2 specimens) (G. Rosenberg, in litt., February 2023), which matches the original account of two differently sized shells in the original “type” lot. These qualify as syntypes, as does lot ANSP 164613 from the type locality. Lot ANSP 164614 stems from Punta Capullana.

In the eastern Atlantic, Semimytilus patagonicus has been known since the end of the 1920s and it was described by Lamy (1931) under the name Modiola pseudocapensis. The exact date of arrival of this South American species to Africa and the vector allowing that process remain unknown, and it was not until recent times that the conspecificity of these two entities was confirmed (de Greef, Griffiths & Zeeman, 2013). Today, S. patagonicus spans along the African coasts for more than 25° latitude (Ma et al., 2020b).

The specimens examined as part of this study represent the only record of this species from Juan Fernández archipelago to date. These specimens were previously mentioned by Soot-Ryen (1955) as Semimytilus algosus and by Bernard, McKinnell & Jamieson (1991) as Mytella speciosa.

Aulacomya atra (Molina, 1782)

Comments: The only record of this species in Juan Fernández archipelago comes from Osorio (2002), who included this archipelago in the geographic distribution of the species without providing specific data.

ARCIDAE

Acar pusilla (Sowerby, 1833)

Figs. 6A–6D

Byssoarca pusilla Sowerby, 1833: 18.

Arca pusilla, – Philippi, 1860: 176. Dall, 1909: 252 [listed only].

? Arca gradata, – Stempell, 1899: 220. (Non Broderip & Sowerby, 1829).

Barbatia (Acar) pusilla, – Marincovich, 1973: 8, fig. 2. Alamo & Valdivieso, 1997: 93 [listed only]. Guzmán, Saá & Ortlieb, 1998: 60.

Barbatia pusilla, – Bernard, 1983: 15 [listed only]. Bernard, McKinnell & Jamieson, 1991: 36 [listed only].

Acar pusilla, – Reinhart, 1939: 41–42, pl. 3, figs. 2a, 2b. Rost, 1955: 191–192, pl. 12, fig. 13. Soot-Ryen, 1959: 20. Osorio & Bahamonde, 1970: 189 [listed only]. Nielsen, 2013: 50, figs. 8g, 8h. Paredes et al., 2016: 134 [listed only]. Valentich-Scott, Coan & Zelaya, 2020: 121, pl. 40.

Type locality: “Iquiqui, Peruviae” [=Iquique, Chile].

Type material: Holotype (NHMUK 1969.236).

Other material studied: Desventuradas: San Félix: LACM 1965-94.2 (2 vs; Figs. 6A–6D).

Distribution: San Bartolo, Lima, Perú (12°18′ S) to Bahía de la Herradura de Guayacán, Coquimbo, Chile (30°00′ S) (Valentich-Scott, Coan & Zelaya, 2020), and Desventuradas archipelago; with a published record from Isla La Plata, Manabí, Ecuador (1°18′ S), see Comments below.

Description: Shell to 12 mm L, trapezoidal, longer than high, markedly inflated, slightly inequilateral, solid. Posterior end higher than anterior end. Umbo low but wide, truncated, subcentrally located, prosogyrate. Dorsal margin straight, forming well-marked angulations at the junctions with anterior and posterior margins. Anterior margin broadly rounded. Ventral margin almost straight, not distinctly separated from anterior and posterior margins, with variably developed byssal embayment. Posterior margin oblique, somewhat arched. Posterior area of shell depressed; umbonal carina strong, rounded. Dissoconch sculptured with about 30 heavy radial ribs and finer comarginal sculpture. Radial ribs separated by narrow interspaces and somewhat projected from shell margins. Comarginal sculpture of low and thin cords near the umbo, gradually increasing in height towards ventral margin, originating bars and scales when crossing over radial sculpture. Shell surface whitish. Hinge plate solid, wider posteriorly than anteriorly; ventral margin arched; with 15 striated teeth, arranged in two continuous series, perpendicular to hinge line. Exterior cardinal area narrow. Ligament elongate, mostly posterior. Inner shell surface porcelaneous, strongly crenulated outside pallial line.

Comments: Stempell (1899) identified specimens from Cavancha, Iquique, Chile as Acar gradata (under Arca). The specimens he studied were not figured, but according to Coan & Valentich-Scott (2012), the range of A. gradata does not extend south of Isla Galápagos, Ecuador. The only Acar species thus far known from the Chilean coastline is A. pusilla. The specimen figured by Huber (2010: 131) as Acar pusilla shows a more trapezoidal shell outline, with obliquely truncated posterior margin, and narrower and more uniform radial sculpture than the holotype of the species (figured by Reinhart, 1939: 2a, b). Due to the lack of provenance information with Huber’s specimen, the identity of this material could not be determined. Rost (1955) identified as Acar pusilla seven specimens collected at Isla [de] la Plata, Ecuador (1.3° S). Valentich-Scott, Coan & Zelaya (2020) were unable to locate these specimens or any other specimen of this species north of Perú. We are uncertain about the northern distributional limit for this species. It may be occasionally found in Equatorial waters or, alternatively, Rost’s (1955) record may be reflecting another case of cryptic speciation, which according to Marko & Moran (2009) is a common phenomenon within this genus.

Acar bernardi n. sp.

Figs. 6E–6J

Hiatella solida, – Bernard, McKinnell & Jamieson, 1991: 36 [listed only]. (Non Sowerby, 1834).

Type locality: 26°20′ S, 80°03′ W, SE off San Félix Island, Desventuradas, 415 m (R/V Anton Bruun Cruise 17, station 675H, by Campbell grab, 12 July 1966; LACM 1966-99).

Type material: Holotype (LACM 3822: 1 v, Figs. 6E, 6F) and two paratypes (LACM 3823: 1 v, Figs. 6G, 6H; FMNH 312477 ex LACM 3823: 1 v, Figs. 6I, 6J) from the type locality.

Distribution: Only known from Desventuradas archipelago.

Diagnosis: Shell trapezoidal, sculptured with wide radial ribs and thin comarginal sculpture, the latter increasing in height ventrally, forming lamellae.

Description: Shell to 9.0 mm L, trapezoidal, longer than high, inflated, inequilateral, solid. Posterior end higher than anterior end. Umbo small in smaller specimens, broad in larger specimens, anteriorly located, prosogyrate. Dorsal margin straight, forming angulations at the junctions with anterior and posterior margins. Anterior margin curved. Ventral margin almost straight, not distinctly separated from anterior margin. Posterior margin obliquely truncated, slightly arched. Posterior area of shell wide, depressed; umbonal carina indistinct. Dissoconch sculptured with strong, low but wide radial ribs and thin comarginal sculpture. Radial ribs separated by interspaces wider than ribs width. Comarginal sculpture consisting of cords near the umbo, gradually increasing in height, forming low lamellae towards ventral margin; originating bars when crossing over radial sculpture. Shell surface whitish. Hinge plate narrow, somewhat wider posteriorly than anteriorly; ventral margin forming a weak angle; with about 20 teeth, perpendicular to hinge line. Anterior and posterior series of teeth continuous. Exterior cardinal area narrow. Inner shell surface porcelaneous, strongly crenulated outside pallial line.

Etymology: Named for Frank R. Bernard (1940–1989), biologist and head of the Shellfish Section and the Salmon and International Section of the Pacific Biological Station in Nanaimo, Canada, who published a preliminary listing of the bivalves collected by the Anton Bruun expeditions to the Desventuradas.

Comments: Acar bernardi n. sp. closely resembles Acar gradata Broderip and Sowerby, 1829 (a syntype of the latter figured by Coan & Valentich-Scott, 2012: pl. 50), from which it differs by having a shorter and higher shell and narrower and higher comarginal sculpture forming lamellae (instead of comarginal cords, as in A. gradata). The trapezoidal shell outline, with anteriorly displaced umbo, markedly projected posterior end, and truncated posterior margin clearly allow the distinction of Acar bernardi n. sp. from Acar pusilla. The present material was previously identified by Bernard, McKinnell & Jamieson (1991); as Hiatella solida. However, the taxodont hinge clearly excludes these specimens from the Hiatellidae.

Anadara stempelli n. sp.

Figs. 6K–6M

Arca (Barbatia) platei Stempell, 1899: 220 (in part).

Type locality: Juan Fernández archipelago, intertidal.

Type material: Holotype (ZMB 51988a: 1 v, Figs. 6K–6M).

Distribution: Only known from Juan Fernández archipelago.

Diagnosis: Shell ovate-elongate, with low umbo, wide posterior area and uniformly crenulated inner shell margin.

Description: Shell to 12.6 mm L, ovate-elongate, longer than high, only slightly inflated, inequilateral, solid. Posterior end higher than anterior end. Umbo prominent, low but wide, anteriorly located, prosogyrate. Dorsal margin straight, forming prominent angulations at the junctions with anterior and posterior margins. Anterior margin broadly rounded. Ventral margin widely curved, not distinctly separated from anterior and posterior margins. Posterior margin sinuous, sloping obliquely. Posterior area of shell somewhat depressed, slightly auriculate; umbonal carina indistinct. Dissoconch sculptured with about 35 coarse and flat radial ribs and finer comarginal sculpture. Radial ribs simple, similar in solidness all along shell surface, producing undulations in shell margins. Comarginal sculpture originating weak bars over radial sculpture. Shell surface pale cream with brownish blotches. Periostracum dark brown, dehiscent. Hinge plate moderately solid, widening at anterior and posterior ends; ventral margin weakly arched. Two continuous series of small teeth, increasing in size anterior and posteriorly, obliquely oriented with respect to hinge line, present. Anterior and posterior series with 16 teeth each. Exterior cardinal area narrow, slightly depressed. Ligament amphidetic, well extended anteriorly and posteriorly. Inner shell surface porcelaneous, strongly crenulated outside pallial line, with weaker radial lines inside pallial line.

Ethymology: Named for the German biologist Carl Ludwig Walter Stempell (1869–1938), who in 1899 published a foundational work on the Juan Fernández bivalve fauna.

Comments: The material on which this new species is based stems from a mixed collection lot that served as syntype series for Arca (Barbatia) platei Stempell, 1899 (currently in genus Kamanevus, see below). The new species described herein clearly differs from K. platei by having a greater number of hinge teeth, which are oriented obliquely (instead of parallel) to the hinge line. In addition, the inner shell margin of the new species is strongly crenulated, whereas that of K. platei is smooth. This represents the first record of the genus Anadara in Chilean waters. Among the Anadara species occurring in the eastern Pacific, Anadara stempelli n. sp. is morphologically most similar to A. formosa (Sowerby, 1833) [syntype figured in Coan & Valentich-Scott, 2012: pl. 56], from which it differs by having a less projected posterior end, resulting in a shorter and higher shell outline, and by its wider posterior area. Furthermore, the inner shell margin crenulations in Anadara stempelli n. sp. are uniform in solidness, whereas in A. formosa there are strong anterior and posterior crenulations and more delicate ventral crenulations.

Bathyarca corpulenta (Smith, 1885)

Figs. 6N–6P

Arca (Barbatia) corpulenta Smith, 1885: 263, pl. 17, figs. 5, 5b.

Bathyarca corpulenta, – Bernard, 1983: 16 [listed only]. Valentich-Scott, Coan & Zelaya, 2020: 125, pl. 41 [syntype from Philippines].

Comments: Smith (1885) described B. corpulenta based on specimens collected during the H.M.S. Challenger Expedition in deep waters of Oceania, Indonesia, mid-Pacific Ocean, and Juan Fernández. At present, the affinities of this material with specimens of Bathyarca reported from other parts of the world remain uncertain. Knudsen (1970) included Bathyarca corpulenta (together with Arca (Barbatia?) imitata Smith, 1885, Bathyarca abyssorum Verrill & Bush, 1898, Arca strebeli Melvill & Standen, 1907, Arca (Bathyarca) nucleator Dall, 1908, and Arca (Bathyarca) corpulenta pompholyx Dall, 1908) in the synonymy of Bathyarca orbiculata (Dall, 1881), considering the latter as a widespread species, occurring in the Atlantic and Pacific Oceans, as well as in the Antarctic waters. However, this interpretation was not shared by Oliver & Allen (1980), who regarded B. corpulenta as a distinct, valid species, restricted to the Pacific Ocean. Coan & Valentich-Scott (2012) and Valentich-Scott, Coan & Zelaya (2020) accepted the synonymy of Arca (Bathyarca) corpulenta pompholyx Dall, 1908 with B. corpulenta, consequently considering the species extending to the North Pacific Ocean. Bathyarca corpulenta is currently known from a single Juan Fernández specimen, an original syntype, obtained at H.M.S. Challenger Expedition station 300 (33°42′ S, 78°18′ W, off Robinson Crusoe Island], in 1,375 fathoms (2,515 m) depth (Smith, 1885), which is here figured for the first time (NHMUK 1889.11.11.131, Figs. 6N–6P). This syntype, although severely damaged, shows some differences in general shell outline, inflation, sculpture, and form of the lunule from the other (eastern Pacific and Indonesian) syntypes. The significance of these differences could not be determined.

Tetrarca fernandezensis (Hertlein & Strong, 1943)

Figs. 6Q–6EE

Arca angulata King, 1832: 336. (Non Bruguière, 1789).

Arca (Arca) fernandezensis Hertlein & Strong, 1943: 154 (replacement name for Arca angulata King, 1832).

Arca angulata King, – Reeve, 1843–1844: Arca plate XIII, species 84. Stempell, 1899: 219, pl. 12, figs. 1–9. Dall, 1909: 251 [listed only]. Odhner, 1922: 222.

Arca (Arca) fernandezensis, – Soot-Ryen, 1959: 20 [listed only]. Osorio & Bahamonde, 1970: 189 [listed only]. Bernard, 1983: 14 [listed only]. Rozbaczylo & Castilla, 1987: 176 [listed only]. Ramírez & Osorio, 2000: 6 [listed only].

Arca fernandezensis, – Bernard, 1983: 14 [listed only]. Bernard, McKinnell & Jamieson, 1991: 36 c.

“Arca” fernandezensis, – Valentich-Scott, Coan & Zelaya, 2020: 123, pl. 40.

“Arca” species A Valentich-Scott, Coan & Zelaya, 2020: 123, pl. 41.

Tetrarca fernandezensis, – Vermeij & Amano, 2021: 44.

? Arca cf. fernandezensis, – Tapia-Guerra et al., 2021: Table S2 [listed only].

Type locality: Off Cumberland Bay, [Robinson Crusoe Island], Juan Fernández archipelago, 80 fathoms [146.3 m].

Type material: Two syntypes (NHMUK 1969.202; with Figs. 6Q–6U).

Other material studied: Desventuradas: San Félix: IOC97-26 (FMNH 322280: 12 vs, juvenile), MNHN-CL MOL 101613 ex FMNH 327895: 11 vs, juvenile); IOC97-29 (FMNH 322282: 2 vs); IOC97-30 (FMNH 322281: 1 v); IOC97-32 (FMNH 322283: 6 vs); LACM 1965-95.3 (1 av, 15 vs); LACM 1966-98.3 (3 vs, juvenile). Juan Fernández: (LACM 10496: 2 avs, with Figs. 6Y, 6Z). Alejandro Selkirk: IOC97-48 (FMNH 322285: 3 specs; FMNH 322287: 4 specs); IOC97-48A (FMNH 322278: 12 vs, juvenile); IOC97-50 (FMNH 322279: 9 vs); IOC97-50A (FMNH 322272: 12 vs, juvenile, with Figs. 6AA–6CC); Eltanin Station 21-203 (USNM 1548474: 1 av, juvenile). Robinson Crusoe: IOC97-44 (FMNH 322271: 6 avs, 5 vs, with Figs. 6W, 6X); IOC97-44A (FMNH 322276: 46 vs, juvenile); IOC97-57 (FMNH 322274: 1 av, Fig. 6V; FMNH 322286: 4 specs); IOC97-57A (MNHN-CL MOL 101614 ex FMNH 322275: 12 vs, juvenile); IOC97-64 (FMNH 322284: 1 spec); IOC97-66A (MNHN-CL MOL 101615 ex FMNH 322267: 2 avs, 2 vs); IOC97-66A (FMNH 322273: 1 av, 6 vs, juvenile, with Figs. 6DD, 6EE); IOC97-68A (FMNH 322277: 17 vs, juvenile); LACM 1965-96.2 (2 vs); LACM 1965-97.2 (4 vs); LACM 1965-99.2 (3 vs, juvenile); LACM 1965-101.3 (1 v); LACM 1965-103.1 (1 v); LACM 1966-100.4 (14 vs, juvenile); “Masatierra”, 60–70 m (SMNH 1225: 2 specs, reported by Odhner, 1922).

Other published records: Juan Fernández, intertidal and in 20–40 fathoms (37–73 m) (Stempell, 1899). Playa “El Palillo”, Robinson Crusoe, Juan Fernández (Ramírez & Osorio, 2000). West Bay, Isla Santa Clara, Juan Fernández, 25 fathoms, USNM 368920; Bahía Cumberland, Robinson Crusoe, Juan Fernández, USNM 368930 (Valentich-Scott, Coan & Zelaya, 2020).

Distribution: Only known from Juan Fernández and Desventuradas archipelagos.

Description: Shell to 41 mm L, subquadrate, trapezoidal or triangular, longer than high, inflated, inequilateral, thick. Posterior end obliquely truncated; anterior end pointed, usually lower than posterior end. Umbo small and pointed in small-sized specimens, wide and rounded in larger specimens; anteriorly displaced, sometimes terminal; usually well-outstanding from dorsal shell margin, frequently eroded in larger specimens; prosogyrate. Antero-dorsal and postero-dorsal margins straight, the former sometimes indistinct. Posterior margin sloping obliquely, straight or sinuous, forming well-marked angulations at the junction with dorsal and ventral margins. Ventral margin sometimes straight and parallel to dorsal margin, sometimes forming a gentle curve; sinuated by a usually broad byssal embayment. Anterior margin parallel to posterior margin, straight to slightly arched; distinctly separated from ventral margin or clearly differentiated and forming a weak angulation with it. Posterior area of shell depressed, delimited by prominent, rounded umbonal carina. Dissoconch sculptured with scaly radial ribs, which crenulate shell margins. Posterior area with 5–7 strong, wide and low ribs; central and anterior areas with numerous, narrow, densely packed ribs, the anteriormost sometimes broader than the others, particularly in smaller specimens. Largest specimens usually with sculpture eroded. Shell surface whitish with brownish blotches or lines. Periostracum light brown, forming foliated distally-branched periostracal bristles, with irregular margins; longer bristles along umbonal carina; periostracum usually lost in larger specimens. Hinge plate narrow, usually uniformly in width, sometimes widening anterior and posteriorly; ventral margin angled to almost straight. Teeth in two series, at weak angle in the smaller specimens, but forming a continuous, straight line in larger specimens. Teeth small, increasing in size from the umbo to the anterior and posterior ends; obliquely oriented in smaller specimens, perpendicular to the hinge line in larger ones; microstriated. Exterior cardinal area wide. Ligament external, amphidetic. Inner shell surface also showing brownish blotches or color lines. Inner margin crenulated. Anterior and ventral margins with fine crenulations, posterior margin with strong crenulations; crenulations restricted to the outside of pallial line. Posterior adductor muscle scar with myophoric ridge extending into the umbonal cavity; anterior adductor muscle sometimes also with myophoric ridge.

Comments: Hertlein & Strong (1943) introduced Arca (Arca) fernandezensis as a replacement name for Arca angulata King, 1832 “non Arca angulata Meuschen, 1787”. Coan & Petit (2006) did not find any available usage of Arca angulata prior to 1832, consequently considering Hertlein & Strong’s name as an unnecessary replacement name. However, Arca angulata King, 1832 is a junior homonym of Arca angulata Bruguière, 1789 (=Glycymeris decussata (Linnaeus, 1758)), a fact that makes Hertlein & Strong’s name valid.

The studied material reveals that Tetrarca fernandezensis is greatly variable in shell outline, morphology of the umbo, and shell sculpture. This is consistent with the variation previously described and figured by Stempell (1899: figs. 1–9). The material reported by Valentich-Scott, Coan & Zelaya (2020) as “Arca” species A fits within this range of variability and consequently this name is included in the synonymy.

Tapia-Guerra et al. (2021) identified as Arca cf. fernandezensis specimens collected in Desventuradas, at 150–180 m. These specimens most probably correspond to Tetrarca fernandezensis.

GLYCYMERIDIDAE

Tucetona sanfelixensis n. sp.

Fig. 7

Figure 7 Family Glycymerididae.

(A–F) Tucetona sanfelixensis n. sp.; (A, B) Holotype, MNHN-CL MOL 101616; (C, D) Paratype, MNHN-CL MOL 101617; (E, F) Paratype, FMNH 32250. Scale bar: (A–F) = 2 cm.

? Tucetona kauaia, – Tapia-Guerra et al., 2021: Table S2 (listed only). (Non Dall, Bartsch & Rehder, 1938.

Type locality: 26°16′52.86″ S, 80° 06′ 48″ W, NW side of San Félix Island, Desventuradas, 40–110 m (by bottom trawling; 28 February 1997; IOC97-39).

Type material: Holotype (MNHN-CL MOL 101616 ex FMNH 327892: 1 v, Figs. 7A, 7B) and 4 paratypes (MNHN-CL MOL 101617 ex FMNH 327893: 1 v, Figs. 7C, 7D; FMNH 322250: 3 v; with Figs. 7E, 7F; LACM 3824 ex FMNH 327894: 1 v) from the type locality.

Distribution: Only known from Desventuradas archipelago.

Diagnosis: Shell subovate, with low and wide umbo, strong hinge plate and prominent crenulations along inner margin. Anterior radial rib flat-topped; posterior radial ribs triangular in outline.

Description: Shell to 36 mm L, subovate, slightly longer than high, somewhat inflated, inequilateral, solid. Umbo subcentrally located, low, wide, orthogyrate. Anterior margin forming a continuous curve with anterior part of ventral margin. Posterior part of ventral margin flattened. Posterior margin straight to slightly arcuated, forming a prominent angulation at the junction with ventral margin. Posterior shell area flattened. Dissoconch sculptured with 21–23 strong radial ribs, crossed by fine, regularly distributed, widely spaced comarginal threads, which forms crossbars over radial ribs. Radial sculpture profile differing along shell: anterior ribs flat-topped, divided longitudinally by a thin furrow; posterior ribs triangular in cross-section. Ribs undulating shell margin. Interspaces as wide as ribs. Hinge plate moderately strong, with relatively long hinge line; 8–12 anterior teeth and 8–13 posterior teeth; both series of teeth separated by a short, edentulous space. Ligament subsymmetrical, slightly longer anteriorly, with 4 chevron grooves. Inner shell surface white to cream, with brown-yellowish mottling. Inner shell margins strongly crenulated. Adductor muscle scars well marked, subovate, subequal.

Etymology: Named for the type locality, San Félix Island, Desventuradas; adjective.

Comments: Tucetona sanfelixensis n. sp. closely resembles the Hawaiian Tucetona kauaia (Dall, Bartsch & Rehder, 1938) (holotype: USNM 173043), from which it differs by having a smaller posterior area of shell and a shorter anterior end. In addition, Tucetona sanfelixensis n. sp. has wider and higher umbos than T. kauaia, and a stronger hinge plate. The radial ribs in Tucetona sanfelixensis are less numerous than in the holotype of T. kauaia (21–23 vs. 29), and contrary to those found in that species, they differ in outline along the shell. Raines & Huber (2012) identified as T. kauaia two valves collected at Eastern Island. However, these valves strikingly differ from the holotype of T. kauaia by having full umbos, less projected anterior and posterior ends, and fewer radial ribs. The specimen figured by Raines & Huber (2012: figs. 9A–9E) appears similar to that described herein as Tucetona sanfelixensis n. sp. The question of conspecificity of these Easter Island specimens deserves further study. Tapia-Guerra et al. (2021) also listed as Tucetona kauaia specimens collected in Desventuradas and the adjacent seamounts, in a bathymetric range of 130 to 215 m depth. These specimens most probably also correspond to the new species described herein.

In general shell outline, Tucetona sanfelixensis n. sp. also resembles Tucetona bicolor (Reeve, 1843) (syntype figured in Valentich-Scott & Garfinkle, 2011: figs. 1H–1J), from which it differs by having a greater number of coarser radial ribs, wider interspaces between the radial ribs, and stronger internal crenulations.

PARALLELODONTIDAE

Kamanevus platei (Stempell, 1899)

Fig. 8

Figure 8 Family Paralellodontidae.

(A–F) Kamanevus platei (Stempell, 1899); (A) Original labels; (B–F) Lectotype, ZMB 51988. Scale bar: (A–F) = 5 mm.

Arca (Barbatia) platei Stempell, 1899: 220, pl. 12, figs. 10–12 (in part).

Arca (Cucullaria) platei, – Dall, 1909: 252 [listed only].

Barbatia platei, – Soot-Ryen, 1959: 20 [listed only]. Osorio & Bahamonde, 1970: 189 [listed only]. Rozbaczylo & Castillo, 1987: 176 [listed only].

Barbatia magellanica, – Bernard, 1983: 15 [listed only]. Bernard, McKinnell & Jamieson, 1991: 36 [listed only]. (Non Gmelin, 1791).

Barbatia s.l. platei, – Huber, 2010: 564.

Kamanevus platei, – Valentich-Scott, Coan & Zelaya, 2020: 135, pl. 45 [syntype].

Type locality: Juan Fernández [archipelago], intertidal.

Type material: The syntype figured by Stempell (1899) was studied herein and is designated as lectotype (ZMB 51988; Figs. 8A–8F).

Distribution: Only known from Juan Fernández archipelago.

Description: Shell to 23 mm L, elongated, longer than high, flat, inequilateral, moderately solid. Posterior end higher than anterior end. Umbo broad, low, only slightly raised from dorsal margin, prosogyrate. Dorsal margin straight. Posterior margin sloping obliquely, straight to slightly convex. Ventral margin not clearly separated from anterior margin, with well-marked byssal embayment. Posterior area of shell slightly depressed. Umbonal carina ill defined. Dissoconch sculptured with numerous, thin radial ribs and comarginal threads. Periostracum dark brown, slightly projected from shell margin, and forming relatively long, thin periostracal bristles, uniform in morphology along the shell. Hinge plate extremely narrow below the umbos, widening anteriorly and posteriorly. Hinge line straight; ventral margin of hinge plate arcuated. Two series of crenulated teeth, parallel to hinge line, comprising four anterior and five posterior teeth; both series separated by an edentulous space. Anterior teeth short, increasing in size from the umbo forwards. Posterior teeth long, straight, reducing in size from the umbo backwards. Exterior cardinal area extremely narrow. Ligament amphidetic, more extended posteriorly than anteriorly. Inner shell surface porcellaneous. Inner shell margin smooth.

Comments: The material on which Stempell (1899) based the description of this species is housed at the ZMB. The syntype series currently consists of one valve pair and a single valve (ZMB 51988). Stempell (1899) mentioned additional specimens not currently in the ZMB collection (C. Zorn in litt., July 2022). The individuals in the available type series are not conspecific. One valve of a different species was separated and described (above) as a new species: Anadara stempelli. To fix the species concept of Arca (Barbatia) platei, we here designate a lectotype for this species (illustrated in Figs. 8B–8F; ZMB 51988). The selected specimen is the specimen originally figured by Stempell (1899: figs. 10–12) and was marked by his label stating “Dieses Expl. liegt meiner Zeichnung zu Grunde” (=this specimen is the basis of my drawing).

Bernard, McKinnell & Jamieson (1991) listed Barbatia magellanica for Juan Fernández archipelago, “from Rozbaczylo & Castilla (1987)”. However, these authors did not report that species, but “Barbatia platei”, a species missing in Bernard, McKinnell & Jamieson’s (1991) list.

Bernard (1983) placed Arca (Barbatia) platei as a junior synonym of Barbatia magellanica Bruguière, 1789 (incorrectly as of Gmelin, 1791), a species originally described from the “Magellan Strait”. This was followed by Bernard, McKinnell & Jamieson (1991) when reporting Barbatia magellanica from Juan Fernández archipelago. Bernard did not explain the proposed synonymy, which disagrees with earlier opinions by Philippi (1849), Kobelt (1888–1891), and Lamy (1907), who considered the stated type locality of B. magellanica as erroneous and viewed it as a junior synonym of Mediterranean Barbatia barbata (Linnaeus, 1758). This was also followed by recent authors (e.g., Huber, 2010).

Kamanevus platei is the first unequivocal extant member of the family Parallelodontidae (Hickman, 2021).

PHILOBRYIDAE

Philobrya aequivalvis (Odhner, 1922)

Figs. 9A–9L

Figure 9 Family Philobryidae.

(A–L) Philobrya aequivalvis (Odhner, 1922); (A, B) Holotype, GNM 11968; (C) LACM 1965-97.3; (D) LACM 1965-101.4; (E, F) USNM 870044; (G) FMNH 322237; (H–J) USNM 886931; (K, L) (SEM) FMNH 322239. (M–S) Verticipronus denticulatus n. sp.; (M, N) Holotype, MNHN-CL MOL 101619; (O–S) (Q–S, SEM) Paratypes, FMNH 322238. Scale bars: (A, B, L) = 2 mm; (C–J, M–P, Q, R) = 1 mm; (K) = 500 µm; (S) = 200 µm.

Avicula (Stempellia n. subgen.) aequivalvis Odhner, 1922: 221, pl. 8, figs. 3, 4. (Non Stempellia Léger & Hesse, 1910).

Philobrya aeqvivalvis [sic], – Soot-Ryen, 1959: 22 (listed only; transfer to Philobryidae).

Philobrya aequivalvis, – Osorio & Bahamonde, 1970: 190 [listed only]. Rozbaczylo & Castilla, 1987: 176 [listed only]. Valentich-Scott, Coan & Zelaya, 2020: 142, pl. 47 [holotype].

Avicula (Stempellia) aequivalvis, – Sandberg & Warén, 1993: 121 [list of Odhner’s taxa].

Philobrya antarctica, – Bernard, 1983: 16 [listed only]. Bernard, McKinnell & Jamieson, 1991: 36 [listed only]. (Non Philippi, 1868).

Philobrya brattstroemi [sic], – Bernard, McKinnell & Jamieson, 1991: 36 [listed only]. (Non P. brattstromi Soot-Ryen, 1957).

Type locality: Masatierra (=Robinson Crusoe Island, Juan Fernández archipelago), 20–35 m.

Type material: Holotype (GNM 11968: 1 av, Figs. 9A, 9B).

Other material studied: Juan Fernández: Alejandro Selkirk: IOC97-48A (FMNH 322235: 2 vs); IOC97-50A (FMNH 322237: 12 vs, with Fig. 9G); Eltanin station 21-203 (USNM 886931: > 500 specs, with Figs. 9H–9J; USNM 904366: > 200 specs, mostly juvenile); Eltanin station 21-205 (USNM 870044: 2 avs, with Figs. 9E, 9F). Robinson Crusoe: IOC97-44A (FMNH 322236: 21 vs); IOC97-57A (FMNH 322234: 1 v); IOC97-66A (FMNH 322239: 53 vs, 2 fragments, with Figs. 9K, 9L), MNHN-CL MOL 101618 ex FMNH 327896: 40 vs); LACM 1965-97.3 (1 av, Fig. 9C); LACM 1965-98.2 (1 av); LACM 1965-101.4 (2 avs, with Fig. 9D); LACM 1965-103.2 (1 av).

Distribution: Only known from Juan Fernández archipelago.

Description: Shell to 5.6 mm L, trigonal to ovoidal, longer than high, somewhat inflated, markedly inequilateral, moderately solid. Posterior end much higher than anterior end. Umbo at anterior end, wide, well projected, particularly in larger specimens. Dorsal margin comprising a short, obliquely straight anterior part, and a widely curved posterior part, the latter clearly separated from posterior margin. Ventral margin straight or with a small byssus embayment; forming a right angle with the anterior part of dorsal margin. Dorsal area of shell weakly differentiated, flat to slightly depressed. Prodissoconch about 1.9 mm long, bounded by a prominent, rounded cord; surface with moderately solid, regularly distributed, zigzag cords, comarginally arranged. Dissoconch sculptured with low but wide comarginal folds and 6–7 low, widely separated radial ribs. Shell surface whitish, shiny. Periostracum thin, pale brownish, slightly projected beyond shell margin; forming radial folds over dissoconch sculpture, and relatively long, strout setae. Hinge with two series of elongate, irregular G1b teeth, perpendicular to hinge line: one anterior, the other posterior to the umbo; the anterior series shorter than the posterior series. Resilifer elongate, triangular, asymmetric. Inner shell surface whitish. Inner margin smooth. Only one (the posterior) adductor muscle scar, present.

Comments: When describing this species, Odhner (1922) thought he was dealing with an unusual pteriid species and consequently introduced a new subgenus. His proposed name Stempellia is preoccupied by a genus of Sporozoa (Stempellia Léger & Hesse, 1910). A replacement name for Odhner’s name is unnecessary, as the species is a member of Philobrya, closely related to P. setosa (Carpenter, 1864) [type species of the genus] and other (sub-) Antarctic species of this genus described by Urcola & Zelaya (2021).

Huber (2010) synonymized Philobrya aequivalvis with P. brattstromi Soot-Ryen, 1957. However, the study of the holotype of the latter (SMNH 3894) reveals that these species are clearly different: P. aequivalvis has a much shorter anterior part of the dorsal margin, more outstanding umbos, a dissoconch with fewer radial ribs (6–7 vs 9–13), a periostracum that is less projected from the shell margin, and an inner shell surface that is completely smooth, lacking the strong posterior radial folds of P. brattstromi. In addition, P. brattstromi has a nearly smooth prodissoconch, whereas that of P. aequivalvis is strongly sculptured with zigzag lines.

Bernard (1983) included P. aequivalvis and Avicula (Meleagrina) magellanica Stempell, 1899 in the synonymy of P. antarctica (Philippi, 1868) (non Thiele, 1931). Philippi’s description of P. antarctica, based on a Magellanic specimen, provides few details, and the nominal species was never illustrated. Its type material could not be located at MNHN-CL (Oscar Gálvez Herrera, in litt., October 2019) or ZMB (D. Zelaya August 2008, personal observations), where other species studied by Philippi are housed. Philobrya antarctica (Philippi, 1868) is here regarded as a nomen dubium. The study of a syntype of Avicula (Meleagrina) magellanica (ZMB-Moll 51991) revealed that this species strikingly differs from P. aequivalvis by having a more rounded shell outline and a periostracum not projected into setae. SEM-aided study of conspecific (Magellanic) specimens of P. magellanica showed that this species bears microscopic pits, a sculpture that clearly differs from the zigzag lines present in P. aequivalvis. Accordingly, Bernard’s (1983) synonymy is not accepted herein. Bernard, McKinnell & Jamieson (1991) reported Philobrya antarctica and Philobrya brattstroemi [sic] Soot-Ryen, 1957 from Juan Fernández archipelago, based on material collected by the Anton Bruun Cruise 17, 1966. The study of the Philobrya specimens collected at this locality by the Anton Bruun proved they actually correspond to P. aequivalvis.

Odhner (1922) reported a size of 6 mm for the holotype of P. aequivalvis. A new measurement of this specimen (herein) reveals that the maximum antero-posterior distance actually is 5.6 mm.

Verticipronus denticulatus n. sp.

Figs. 9M–9S

Type locality: 33°45′37.14″ S, 80°45′08.13″ W, E coast of Alejandro Selkirk Island, Juan Fernández, 25 m (sediment sample collected from sand pockets by R. Bieler; 6 March 1997; IOC97-50A).

Type material: Holotype (MNHN-CL MOL 101619 ex FMNH 327900: 1 v, Figs. 9M, 9N) and seven paratypes (MNHN-CL MOL 101620 ex FMNH 327901: 2 v; FMNH 322238: 5 vs, with Figs. 9O–9S) from the type locality.

Distribution: Only known from Juan Fernández archipelago.

Diagnosis: Shell modioliform, extremely high, with a prominent angulation at the junction of dorsal and posterior margins. Shell surface glossy, with orange-brownish color bands. G1b teeth persistent in larger specimens.

Description: Shell to 2.3 mm L, trigonal, longer than high, markedly inequilateral, inflated, thin. Posterior end much higher than anterior end. Umbo near anterior end, low, not outstanding from shell margin. Dorsal margin long, straight, oblique; forming a prominent angulation at the junction with the dorsal part of posterior margin. Ventral part of posterior margin widely curved. Ventral margin straight to slightly concave. Anterior margin extremely short, curved. Dorsal area of shell wide, depressed. Prodissoconch of about 340 µm in diameter, sculptured with strong radial ribs, increasing in number distally by intercalation. Dissoconch only sculptured with irregular growth marks, some of them forming folds. Shell surface glossy, with wide, orange-brownish color bands. Hinge with two series of minute G1b teeth, perpendicular to hinge line: one anterior and the other posterior to the ligament, the latter twice longer than the former. G1b teeth persistent in larger specimens. G2 teeth represented by three elements in each valve: a minute, tubercular anterior (“cardinal”) tooth, located just behind the umbo, and two larger posterior (“lateral”) teeth, well distanced from umbo. Posterior G2 teeth elongate. The most ventral posterior tooth of the right valve distally bifurcated. Resilifer elongated, widening posteriorly. Inner shell surface shiny, also showing external color. Inner shell margin smooth. Posterior adductor muscle scar large, subcircular, dorsally displaced.

Etymology: The name of the species refers to the numerous (G1b) teeth present in the larger specimens.

Comments: To date, Verticipronus is the only philobryid genus having small but stout, anterior G2 “cardinal” and two elongated, posterior G2 “lateral” teeth. These conditions are present in the Juan Fernández material studied herein. However, specimens of the new species show two anterior G2 teeth, instead of only one as described for the genus by Urcola & Zelaya (2021); and the G1 teeth remain well-developed in larger specimens (instead of fading during the ontogeny). In view of our imperfect knowledge of philobryids, the significance of these morphological differences remains unclear and we tentatively place the new species in Verticipronus.

To date, only three living species of Verticipronus are known worldwide: V. mytilus Hedley, 1899 (from New Zealand), V. tristanensis Soot-Ryen, 1952 (from Tristan da Cunha in the central South Atlantic Ocean), and V. cowuti Urcola & Zelaya, 2021 (from southern Patagonia) (MolluscaBase, 2024). The modioliform shape of Verticipronus denticulatus n. sp. makes this species most similar to V. tristanensis, from which it differs by a much higher shell with a wider posterior area and a prominent angulation at the junction of dorsal and posterior margins. Verticipronus mytilus and V. cowuti have the umbo at the anterior shell end, resulting in a mytiliform shape. This is the first record of the genus in the Eastern Pacific.

ANOMIIDAE

Monia sp. B

Fig. 10

Figure 10 Family Anomiidae.

(A–F) Monia sp. B; (A, B) FMNH 322337; (C, D) FMNH 322338; (E, F) FMNH 322336. Scale bar: (A–F) = 2 mm.

Material studied: Desventuradas: San Félix: IOC97-22 (FMNH 322337: 1 v, Figs. 10A, 10B); IOC97-29 (FMNH 322336: 1 v, Figs. 10E, 10F); IOC97-32 (FMNH 322338: 1 v, Figs. 10C, 10D).

Description: Shell to 5.5 mm H, subcircular to ovate, irregular in outline, higher than long, flat, thin, translucent. Upper valve only slightly inflated. Umbo not outstanding from shell margin. Shell sculptured with numerous, wart-like, rounded or somewhat elongated, protuberances that show some radial arrangement; additional growth striae are evident towards the margins. Prodissoconch I orange, of about 240 µm in diameter; prodissoconch II amber, well discernible from dissoconch. Prodissoconch I + Prodissoconch II = 1,100 µm in diameter. Dissoconch whitish or brownish. Inner surface smooth, glossy, showing the outer shell color. Resilifer wide, arched, deep. Hinge teeth absent. Central area with two elongate, confluent, finely striated muscle scars.

Comments: To date, only one anomiid species is known from Chilean waters: Monia sp. (fide Raines & Huber, 2012). The material studied herein likewise shows two elongated and striated muscles scars. However, our material strikingly differs by lacking the radial sculpture described and figured by Raines & Huber (2012: fig. 15) for Monia sp.

Monia sp. B appears to be a new, undescribed, species. However, the limited material we have available (three upper valves) and its imperfect state of preservation preclude us from formally naming it herein.

PECTINIDAE

Argopecten purpuratus (Lamarck, 1819)

Figs. 11A–11D

Figure 11 Family Pectinidae.

(A–D) Argopecten Purpuratus (Lamarck, 1819), LACM 1965-95.4. (E–P) Zygochlamys phalara (Roth, 1975), USNM 764199. Scale bars: (A–P) = 2 cm.

Pecten purpuratus Lamarck, 1819: 166.

Pecten rudis Sowerby, 1846: 254, pl. 3, fig. 32.

Pecten purpuratus, – Hupé, 1854–1858: 289. Philippi, 1860: 178. Stempell, 1899: 228. Dall, 1909: 256 [listed only]. Dijkstra, 1994: 473, pl. 4, figs. 9–11 [lectotype].

Chlamys (Argopecten) purpurata, – Grau, 1959: 103, pl. 34. Herm, 1969: 107–109, pl. 4, figs. 1–5. Osorio & Bahamonde, 1970: 193 [listed only]. Osorio, Atria Cifuentes & Mann Fischer, 1979: 27, fig. 29.

Plagioctenium purpuratum, – Soot-Ryen, 1959: 31.

Aequipecten (Plagioctenium) purpuratus, – Olsson, 1961: 162–163, pl. 19, figs. 1, 1b. Peña Gonzáles, 1971: 130.

Argopecten purpuratus, – Marincovich, 1973: 10, fig. 8. Wolff, 1988: 213–217. Bernard, McKinnell & Jamieson, 1991: 36 [listed only]. Rombouts, 1991: 8, pl. 4, fig. 5. Bellolio, Lohrmann & Dupré, 1993: 332–342; Bellolio, Toledo & Campos, 1994: 229–237. Bernard, 1983: 24 [listed only]. Tapia, Dupré & Bellolio, 1993: 75–84. Alamo & Valdivieso, 1997: 106, fig. 255. Avendaño & Le Pennec, 1997: 175–182; 1998: 13–16. Guzmán, Saá & Ortlieb, 1998: 65. González et al., 1999: 307. Aguilar & Stotz, 2000: 749–755. Moraga et al., 2001: 51. Osorio, 2002: 132. Raines & Poppe, 2006: 310, 311, pls. 269, figs. 1–3, pl. 270, figs. 1–3. Thébault et al., 2008: 45. Pérez et al., 2009: 1585–1593. Huber, 2010: 203. Coan & Valentich-Scott, 2012: 282, pl. 92. Uribe et al., 2013: 217. Paredes et al., 2016: 136 [listed only]. Valentich-Scott, Coan & Zelaya, 2020: 170, pl. 55.

Pecten rudis, – Griffin & Nielsen, 2008: 285, pl. 15, figs. 7, 8 [holotype].

Type localities: Callao, Perú (Pecten purpuratus, see Grau, 1959). Coquimbo, Chile (Pecten rudis).

Type material: Lectotype (MHNG 1088/20/2), one paralectotype (MNHG 1088/20/1) and two paralectotypes (MNHN-IM 2000-29951) of Pecten purpuratus.

Material studied: Desventuradas: San Félix: LACM 1965-95.4 (1 av, Figs. 11A–11D).

Other published records: Juan Fernández (Bernard, McKinnell & Jamieson, 1991).

Distribution: Paita, Perú (Peña Gonzáles, 1971) to Valparaíso, Chile (Osorio, 2002); Desventuradas and Juan Fernández archipelagos.

Description: Shell to 159 mm L, subcircular, slightly longer than high, somewhat inflated, thick. Umbo minute, pointed. Shell disc sculptured with 22–30 wide and flat radial ribs, square in section, crossed by fine comarginal threads. Interspaces narrower than ribs. Anterior auricle slightly longer than posterior auricle. Base of posterior auricle completely attached to the disc; base of anterior auricle detached from the disc, more evident in the right valve. Anterior auricle sculptured with 3–5 scaly radial riblets. Posterior auricle with 7–10 scaly riblets. Byssal notch moderately deep. Exterior color evenly purple, brownish, violet, violet or orange, sometimes mottling or radially banded; right valve usually paler. Interior shell relief reflecting outer shell surface, strongly serrated at the margin. Inner surface shiny, whitish, usually variably stained in brownish or reddish, and with a dark color band along hinge line. Hinge edentulous. Resilifer small, triangular, symmetric with respect to the umbo.

Comments: Lamarck (1819) reported this species as coming from “Japon.” Grau (1959) considered this locality in error, emending it to “Callao, Peru.” Argopecten purpuratus was reported northward to Nicaragua (e.g., Grau, 1959) but these records are based on misidentifications of Argopecten ventricosus (Sowerby, 1842) (see Coan & Valentich-Scott, 2012).

Zygochlamys phalara (Roth, 1975)

Figs. 11E–11P

Chlamys phalara Roth, 1975: 81–84, pl. 6, figs. 1–4.

Chlamys phalara, – Bernard, 1983: 25 [listed only]. Bernard, McKinnell & Jamieson, 1991: 36 [listed only]. Waller, 1991: 30.

Chlamys (Chlamys) phalara, – Rombouts, 1991: 17, pl. 7 fig. 3.

Psychrochlamys phalara, – Jonkers, 2003: 52–53, pls. 1, 8. Raines & Poppe, 2006: 226–228, pl. 175, 8 figs.

Zygochlamys phalara, – Huber, 2010: 205. Valentich-Scott, Coan & Zelaya, 2020: 174, pl. 56.

Type locality: 33°29–42′ S, 78°55′ W, Isla Más a Tierra [=Robinson Crusoe Island], Juan Fernández archipelago, 80–200 m.

Type material: Holotype (CAS 54752); two paratypes (CAS 54753, 54754); three paratypes (F277692); four paratypes (4 avs, USNM 880644; here studied); four paratypes (LACM 2653).

Other material studied: Juan Fernández: 200 m (USNM 846081: 1 av; USNM 888364: 1 av). Alejandro Selkirk: Eltanin Station 21-203 (USNM 887914: 1 spec, 1 v); Eltanin Station 21-205 (USNM 870045: 1 av, 3 vs; USNM 897774: 12 specs. juvenile). Robinson Crusoe: LACM 1965-98.3 (62 vs, juvenile); LACM 1965-101.5 (8 avs, 48 vs); LACM 1966-102.2 (5 vs); Anton Bruun Cr. 12, [33°40′27″ S, 78°56′3″ W], W Carvajal Bay, 9–12 m (USNM 764206: 1 v); Anton Bruun Cr. 12, R-206, 80–200 m (USNM 701662: 16 vs); 80–200 m, by fish trawler (USNM 846082: 2 avs); Anton Bruun Cr. 12, 33°41.2′ S, 78°57′ W to 33°40.7′ S, 78°51.8′ W, 130–170 m (USNM 764199: 10 avs, 72 vs, with Figs. 11E–11P); Anton Bruun Cr. 12, station 255, 33°40′35″ S, 78°54′40″ W, 130–170 m (USNM 679522: 5 avs, 10 vs); station MV65-IV-47, 33°37′52″ S 78°49′07″ W, 150 m (USNM 764215: 275 vs); station MV65-IV-54, 33°37′54″ S, 78°46′03″ W (USNM 764195: 20 vs).

Distribution: Only known from Juan Fernández archipelago.

Description: Shell to 39 mm L, subcircular, as high as long, somewhat obliquely projected posteriorly, compressed, thin. Umbo minute, pointed. Shell disc sculptured with 30–65, thin, flat radial ribs, crossed by fine comarginal threads. Interspaces as wide as or wider than ribs. Anterior auricle larger than posterior auricle. Base of anterior and posterior left valve, and posterior right valve auricles completely attached to the disc; base of right valve anterior auricle detached from the disc for about a half of its length. Anterior auricle sculptured with 5–6 scaly to nodulose radial riblets. Posterior auricle with five spiny riblets. Byssal notch moderately deep. Exterior color of left valve ochraceous, orange or reddish, sometimes with color bands or blotches; right valve usually paler, uniform in color or with color bands. Interior shell surface shiny, yellowish or whitish, sometimes stained reddish near the umbo. Hinge edentulous. Resilifer small, triangular, symmetric with respect to the umbo.

Comments: When describing this species, Roth (1975) examined 34 specimens, all of them considered as part of the type series, with the originally illustrated holotype and two paratypes deposited in numbered lots of the CAS collection, and additional, unspecified number of paratypes distributed to LACM, MNHN-CL, and USNM. Of these type specimens, we could trace the holotype and 13 paratypes. The paratypes intended for MNHN-CL were not received (O. Galvez Herrera, in. litt., June 2022). The USNM collection contains about 1,000 topotypic specimens from the Anton Bruun’s Cruise 12, but these could not be unambiguously linked to the type series. The exact type locality is unknown. As noted by Roth (1975), the Anton Bruun’s Cruise 12 performed eight hauls off Juan Fernández Island on 15 December 1965. Roth (1975) deemed specimens cited by Soot-Ryen (1959) from south-central Chile (as C. amandi) as “almost certainly conspecific” with Zygochlamys phalara, and thus gave the distributional range of his new species as “Calbuco to Chonos Archipelago, in 5–300 m.” However, Soot-Ryen’s records most probably correspond to Zygochlamys patagonica (King, 1832), a common species in this area (Valentich-Scott, Coan & Zelaya, 2020).

PROPEAMUSSIIDAE

Propeamussium/Parvamussium sp. A

Fig. 12

Figure 12 Family Propeamussiidae.

(A, B) Propeamussium/Parvamussium sp. A, LACM 1966-98.4. Scale bar: (A, B) = 1 mm.

Propeamussium cf. malpelonium, – Bernard, McKinnell & Jamieson, 1991: 36 [listed only]. (Non Dall, 1908).

Material examined: Desventuradas: San Félix: LACM 1966-98.4 (1 v, Figs. 12A, 12B).

Description: Shell of 4.5 mm L (partially broken), ovate, as high as long, somewhat obliquely projected anteriorly, fragile. Shell disc sculptured with weak, regularly distributed comarginal lirae. Auricles not present in the available material. Exterior color white, opaque. Inner shell surface showing 11 radial riblets.

Comments: The material reported herein was previously identified by Bernard, McKinnell & Jamieson (1991) as Propeamussium cf. malpelonium (Dall, 1908). However, the syntypes of that species (USNM 122871, figured by Coan & Valentich-Scott, 2012: pl. 100), from the north-eastern Pacific, show a more orbicular shell outline, and that species is considerably larger than the material studied herein (up to 20 mm H, according to Coan, 2000). The material studied herein appears to be undescribed, but the poor state of preservation of the single (right) valve precludes a formal description herein. The valve is internally corroded near the umbo and broken in its ventral margin, thus not allowing to determine the total extension of the internal radial riblets, and consequently the exact generic placement for this entity.

LIMIDAE

Limaria crusoensis n. sp.

Figs. 13A–13H

Figure 13 Family Limidae.

(A–H) Limaria crusoensis n. sp.; (A, B) Holotype, MNHN-CL MOL 101621; (C–F) LACM 3835; (G, H) USNM 887913. (I–L) Lima nasca (Bernard, 1988), LACM 1966-99.2. (M–P) Limatula sanfelixensis n. sp.; (M, N) Holotype, LACM 3825; (O, P) Paratype, LACM 3826. Scale bars: (A–F, I–L) = 5 mm; (G, H, M–P) = 1 mm.

Lima angulata, – Stempell, 1899: 229. Dall, 1909: 256 [listed only]. Odhner, 1922: 221. (Non Sowerby, 1843, non Münster, 1841).

Limaria orbignyi, – Soot-Ryen, 1959: 33 [listed only]. Stuardo, 1968: 172 [listed only]. (Non Lamy, 1930–1931).

Limaria orbigny [sic], – Osorio & Bahamonde, 1970: 193 [listed only]. Rozbaczylo & Castilla, 1987: 176 [listed only]. (Non Lamy, 1930–1931).

Promantellum orbignyi, – Bernard, McKinnell & Jamieson, 1991: 36 [listed only]. (Non Lamy, 1930–1931).

Limaria hemphilli, – Valentich-Scott, Coan & Zelaya, 2020: 187, pl. 60. (Non Hertlein & Strong, 1946).

Type locality: 33°36′01″–33°36′12″ S, 78°53′42″–78°53′08″ W, W side of N tip of Robinson Crusoe Island, Juan Fernández, 50–80 m (by bottom trawling, IOC-97 team; 8 March 1997; IOC97-62).

Type material: Holotype (MNHN-CL MOL 101621 ex FMNH 322331: 1 av (1 v broken), Figs. 13A, 13B). Paratypes: Juan Fernández: Alejandro Selkirk: Eltanin Cruise 21, station 203 (USNM 887913: 2 vs, with Figs. 13G, 13H). Robinson Crusoe: IOC97-44 (FMNH 322332: 2 vs, damaged); IOC97-67 (FMNH 322330: 1 v); LACM 3835 (66-100; 1 av, 6 vs, with Figs. 13C–13F).

Other material examined: Juan Fernández, “Masatierra” (Robinson Crusoe), 30–40 m (SMNH 1226: 1 spec., reported by Odhner, 1922 as L. angulata).

Other published records: Juan Fernández, 20 and 40 fathoms (Stempell, 1899: as L. angulata).

Distribution: Only known from Juan Fernández archipelago.

Diagnosis: Shell obliquely ovate, with flat posterior margin. Umbo narrow and pointed. Radial sculpture prominent, separated by wide interspaces.

Description: Shell to 12.4 mm H, obliquely ovate, higher than long, weakly inflated, inequilateral, thin. Anterior end somewhat produced. Umbo small, narrow, pointed; only slightly projected from shell margin. Dorsal margin straight. Anterior margin long, sloping obliquely, only slightly arched. Anterior part of ventral margin markedly curved. Posterior part of ventral margin and posterior margin flattened, forming an obscure angulation among them. Auricles small, subequal; margin of anterior auricle continuous with anterior margin of disc; auricular sinus indistinct; posterior auricle minutely projected, with a barely developed sinus. Dissoconch sculptured with fine radial ribs and regularly distributed comarginal threads. Radial ribs restricted to median and posterior areas. Median area with 20–26 ribs, separated by interspace wider than ribs width, ventrally developing into narrow lamellae, originating serration in shell margin; posterior area with 5–6 ribs, lower, weaker and more spread than those of central area. Shell surface whitish, translucent. Anterior and posterior gaps absent. Hinge plate solid, edentulous. Resilifer broadly triangular, symmetric with respect to the umbo, shallow. Inner shell surface reflecting outer shell sculpture.

Etymology: Named for the type locality, Robinson Crusoe Island; adjective.

Comments: Three nominal species of Limaria have been indicated as occurring in the Juan Fernández Archipelago: Stempell (1899) and Odhner (1922) reported L. angulata Sowerby, 1843, Osorio & Bahamonde (1970), Rozbaczylo & Castilla (1987), and Bernard, McKinnell & Jamieson (1991) reported Promantellum orbignyi, and Valentich-Scott, Coan & Zelaya (2020) reported L. hemphilli (Hertlein & Strong, 1946). Study of the specimens mentioned by Stempell (1899), Bernard, McKinnell & Jamieson (1991) and Valentich-Scott, Coan & Zelaya (2020) showed the presence of only a single species of Limaria at Juan Fernández archipelago. These mentioned taxa do resemble the Juan Fernanández specimens but a comparison of their type material revealed that it does not match Limaria crusoensis n. sp. herein.

The holotype of L. hemphilli (figured in Coan & Valentich-Scott, 2012: pl. 105) clearly differs from the Juan Fernández specimens by having shells with a broad posterior area, more projected and sharply pointed posterior auricles, and the more numerous (about 45), wider, and lower radial ribs. We thus consider the record of L. hemphilli from Juan Fernández Islands by Valentich-Scott, Coan & Zelaya (2020) as a misidentification.

Lamy (1930–1931) noticed that Lima angulata G.B. Sowerby II, 1843 is a homonym of Lima angulata Münster, 1841, consequently proposing Lima (Mantellum) orbignyi as a replacement name for the former. This species was later considered a member of Limaria (e.g., Hertlein & Strong, 1946; Stuardo, 1968; Coan & Valentich-Scott, 2012). Within this species, Lamy (1930–1931) recognized the “form basilanica”, including the nominal species Limaria basilanica (Adams & Reeve, 1848–1850) and its presumed synonym, Limaria orientalis (Adams & Reeve, 1848–1850). However, recent studies suggest not only that L. basilanica and L. orientalis are different species, but also that these taxa are different from L. orbignyi, with the first two species restricted to the western Pacific and Indo-Pacific Oceans (Stuardo, 1968; Beu, 1977, 2004; Kilburn, 1998; Higo, Callomon & Goto, 1999; Beu & Raine, 2009; Marshall & Spencer, 2013), and the latter to the eastern Pacific (Stuardo, 1968; Coan & Valentich-Scott, 2012). Kilburn (1998) figured the holotype of L. basilanica which differs from Limaria crusoensis n. sp. by having longer shell, with narrower and more numerous radial sculpture and markedly inflated umbo. Several specimens of L. orientalis were figured by Beu (1977: pl. 1; 2004: figs. 11B, 11D) and Kilburn (1998: fig. 5). All these specimens differ from the Juan Fernández specimens studied herein by having more numerous and irregular radial sculptural elements and a markedly convex posterior shell margin. Limaria crusoensis n. sp. also differs from the syntypes of Limaria orbignyi (NHMUK 968879) by having a flattened (instead convex) posterior margin, a less arched anterior margin, a smaller number of more widely separated radial ribs, and a narrower (not inflated) umbo.

Bernard (1988) described Limaria valdiviesae from 6°21′ S, 80°56′ W, “northern Chile” [actually Perú]. The holotype of this species (LACM 2334) has a longer and straighter anterior margin and a more convex posterior margin, originating a more markedly oblique shell outline than in the Juan Fernández specimens. In addition, L. valdiviesae shows a rounded umbo, an indistinct posterior auricle, and obscure radial striae. Coan & Valentich-Scott (2012) considered L. valdiviesae as a synonym of L. hemphilli, which was followed by Valentich-Scott, Coan & Zelaya (2020), but not herein.

In general shell outline, the Juan Fernández specimens also resemble the Eastern Island specimens mentioned by Raines & Huber (2012) as Limaria (Limatulella) sp. However, these specimens differ by having radial ribs all along the shell surface.

Bernard, McKinnell & Jamieson (1991) listed a new species of Limaria from Desventuradas archipelago. The material could not be located at the LACM (L. Groves in litt., February 2022).

Lima nasca (Bernard, 1988)

Figs. 13I–13L

Plicacesta nasca Bernard, 1988: 228, 230, fig. 4.

Acesta diomedia {sic}, – Bernard, McKinnell & Jamieson, 1991: 36 [listed only]. (Non Lima (Acesta) diomedae Dall, 1908).

Lima (“Allolima”) nasca, – Huber, 2010: 632.

Type locality: 25°44′ S, 82°25′ W, off Shoal Guyot, Nasca Ridge, 228 m.

Type material: Holotype (LACM 2332) and 1 paratype (NSMT 64674).

Material studied: Desventuradas: San Félix: LACM 1966-99.2 (1 av, with dried body, Figs. 13I–13L).

Distribution: Off Chile, between 25°44°S and 26°20′ W, including Desventuradas archipelago, 160–228 m.

Description: Shell up to 127 mm H, obliquely ovate, higher than long, slightly inflated, inequilateral, solid. Umbo small, narrow and pointed, only slightly projected from shell margin. Postero-dorsal margin short, straight, horizontal. Anterior margin long, sloping obliquely straight. Ventral and posterior margins forming a wide, continuous curve. Posterior auricle small, separated by a shallow auricular sinus from the disc. Dissoconch sculptured with about 40 strong radial ribs, separated by narrow interspaces. Interspaces with delicate comarginal threads. Anteriormost radial rib with a series of small but stout spines; posteriormost radial ribs with few, sparse spines, particularly towards shell margin. Posterior auricle with prominent, sharply pointed spines. Anterior auricle ‘hidden’ from dorsal view below strongly demarcated, oblique anterodorsal ridge adorned with 6-8 finer radial ribs. Shell surface whitish. Anterior gap a narrow slit; posterior gap absent. Hinge plate solid, edentulous. Resilifer broad, posterior to the umbo. Inner shell surface furrowed by outer radial sculpture; margin strongly crenate.

Comments: When describing this species, Bernard (1983) considered Lima nasca as closely related to Acesta sphoni (Hertlein, 1963), a species occurring in the northeast Pacific Ocean at California, USA (Coan, 2000). However, L. nasca differs strikingly from that species by having a thicker shell with coarser radial sculpture and a more clearly separated posterior auricle. In addition, L. nasca bears spines, which are absent in A. sphoni.

Lima nasca closely resembles Atlantic L. marioni P. Fischer, 1882 (redescribed and figured by Mikkelsen & Bieler, 2003), from which it differs by having a longer and straighter anterior margin, a more clearly differentiated posterior auricle, and narrower radial ribs. In an unpublished dissertation, Stuardo (1968) proposed “Allolima” as a subgenus of Lima, for a group of species related to Lima tomlini Prashad, 1932. Within this group, Stuardo (1968) also included Lima marioni and Lima perfecta Smith, 1904. Thirty years later, Kilburn (1998) introduced the genus Fukama to include his new species F. messura (type species) and L. perfecta. Kilburn (1998: 236) recognized that “it would appear probable that… [Stuardo’s unpublished “Allolima”] group is equivalent to the genus here proposed.” Mikkelsen & Bieler (2003) and Huber (2010), the latter mistakenly considering Stuardo’s manuscript name a senior synonym of Kilburn’s Fukama, did not recognize the grouping as distinct from Lima at the genus level. In view of the still imperfect knowledge of worldwide limid taxonomy, we follow the placement in Lima sensu lato.

Lima nasca was only known from the type locality (Nasca Ridge, off Chile), from where it was described based on disarticulated valves. A specimen from San Félix in the Desventuradas Archipelago, reported as “Acesta diomedia [sic]” by Bernard, McKinnell & Jamieson (1991) belongs to the same species and extends the known range. Acesta diomedae (Dall, 1908) differs from Lima nasca in having an evenly convex posterior margin (i.e., without the sinuation that appears in L. nasca) and consequently lacking a differentiated posterior auricle. In addition, A. diomedae has a greater number of radial ribs than L. nasca (50 vs. 40, respectively).

Limatula sanfelixensis n. sp.

Figs. 13M–13P

Limatula cf. pygmaea, – Bernard, McKinnell & Jamieson (1991): 36 [listed only]. (Non Philippi, 1845).

Type locality: SE off San Félix Island, Desventuradas, 170–160 m; station 676B, by Menzies trawl, 26°20′ S, 80°02′ W; 12 July 1966 (LACM 66-99).

Type material: Holotype (LACM 3825: 1 v, Figs. 13M, 13N) and 2 paratypes (LACM 3826: 1 v, Figs. 13O, 13P, plus 1 fragment; FMNH 312473: 1 v ex LACM 3826) from the type locality.

Distribution: Only known from Desventuradas archipelago.

Diagnosis: Shell somewhat inequilateral, with posterior margin more convex than the anterior one. Auricles small, ill defined. Radial ribs strong, with scales.

Description: Shell of up to 5.1 mm H, ovate, higher than long, inflated, somewhat inequilateral, solid. Umbo small but well outstanding from shell margin, rounded. Antero-dorsal and postero-dorsal margins straight, similar in length. Anterior, ventral and posterior margins forming a continuous curve, the posterior one somewhat more markedly convex than the anterior one. Anterior and posterior auricles small, similar in size, weakly differentiated; auricular margins continuous with anterior and posterior shell margin, lacking auricular sinus. Dissoconch sculptured with strong, scaly radial ribs and delicate comarginal threads. Radial sculpture evenly distributed along all shell surface, sometimes with a stronger, median rib. Comarginal sculpture widely separated near the umbo, more closely spaced towards ventral margin. Shell surface whitish. Anterior and posterior gaps absent. Hinge plate solid, edentulous. Resilifer small, triangular, just below the umbo. Inner shell surface furrowed by outer radial sculpture, with a median channeled groove; margins crenulated.

Etymology: Named for the type locality, San Félix Island, Desventuradas; adjective.

Comments: Limatula sanfelixensis n. sp. clearly belongs to the “Limatula japonica species complex” as defined by Kilburn (1998). In particular, the new species described herein closely resembles Limatula japonica Adams, 1864 from Japan and Limatula spinulosa Fleming, 1978 from the Kermadec Islands, from which it differs by having a more broadly curved anterior shell margin leading to a more inequilateral outline, and a smaller umbo. In addition, Limatula sanfelixensis n. sp. has a considerably smaller resilifer than the holotype of Limatula spinulosa (NZNM 226931), and less differentiated auricles than L. japonica. Fleming (1978) introduced Limatuletta as a subgenus of Limatula, based on Lima japonica. Within this subgenus, he also included L. spinulosa as a subspecies of L. japonica. The validity of Limatuletta, and its relationships to other proposed subgenera of Limatula could not be determined in the context of the still limited understanding of limid phylogeny.

The Desventuradas specimens studied herein were previously identified by Bernard, McKinnell & Jamieson (1991) as Limatula cf. pygmaea. The original description of Lima pygmaea (Philippi, 1845) lacks detail and illustration; its type material is considered lost and was never figured. Consequently, the identity of this species is an enigma. However, Lamy (1930–1931), Carcelles (1947), and Dell (1964) regarded Limatula falklandica A. Adams, 1864 as a synonym of L. pygmaea; and Carcelles (1947) added Limea martiali Mabille & Rochebrune in Rochebrune & Mabille, 1889 to that synonymy. Most recent researchers (e.g., Fleming, 1978; Dell, 1990; Allen, 2004; Engl, 2012) followed this treatment. Huber (2010) distinguished more than one species under the concept of “Limatula pygmaea” as used by various authors. The Desventuradas specimens studied here strikingly differ from the holotype of Limatula falklandica (NHMUK without number) and the syntype of Limea martiali (MNHN-IM-2000-31627) by having radial sculpture distributed all along the dissoconch (instead of restricted to the central part) and with prominent scales, stronger and more separated comarginal sculpture, and smaller resilifer. The above-mentioned set of characters also allows the distinction of Limatula sanfelixensis n. sp. from L. ceciliaosorioae Gálvez & Wacquez, 2018, a species described from central Chile.

Limatula chilensis Campusano, Ruz & Oliva, 2012, described from northern Chile, differs from Limatula sanfelixensis n. sp. by having more elongated and arched shell outline and narrower radial ribs.

LUCINIDAE

Cavilinga taylorgloverorum n. sp.

Fig. 14

Figure 14 Family Lucinidae.

(A–K) Cavilinga taylorgloverorum n. sp.; (A–D) Holotype, LACM 3827; (E–J) Paratypes, FMNH 322315; (K) Paratype, FMNH 322313. Scale bars: (A–J) = 5 mm; (K) = 2 mm.

Cavilinga species A Valentich-Scott, Coan & Zelaya, 2020: 195, pl. 62.

Type locality: LACM 1965-95: “26°17.5′ S, 80°05.6′ W”, NW side of San Félix Island, Desventuradas, 10–45 ft [3–13.7 m]; leg. Sylvia E. Taylor, Alan Chapman, 5–6 December 1965 [stated coordinates too far inland].

Type material: Holotype (LACM 3827: 1 av, Figs. 14A–14D). Paratypes: Desventuradas: San Félix: IOC97-22 (FMNH 322315: 1 av, 23 vs, with Figs. 14E–14J; MNHN-CL MOL 101622 ex FMNH 327902: 10 vs); IOC97-26 (FMNH 322313: 9 avs juv, 28 vs, with Fig. 14K).

Other material studied: Desventuradas: San Félix: IOC97-29 (FMNH 322317: 5 vs); IOC97-32 (FMNH 322316: 4 vs); LACM 1965-94.3 (3 vs); LACM 1965-98.4 (1 v); “San Félix” (MZUC, without number/10005: 35 v). Juan Fernández: Robinson Crusoe: IOC97-68A (FMNH 322318: 1 v).

Distribution: Only known from Desventuradas and Juan Fernández archipelagos.

Diagnosis: Shell subovate, with short antero-dorsal margin, incised lunule, wide posterior area of shell, and weak comarginal sculpture.

Description: Shell up to 14 mm L, subovate, longer than high, somewhat inflated, inequilateral, moderately solid. Anterior end projected; posterior end truncated. Umbo small, narrow, markedly prosogyrous. Postero-dorsal margin steeply sloping, straight to slightly convex. Posterior margin short, straight, slightly oblique, forming angulations at the junction with ventral and dorsal margins. Anterior margin gently curved, not distinctly separated from ventral margin. Antero-dorsal margin short, markedly concave, forming a well-marked angulation at the junction with anterior margin. Posterior area of shell flat, set off by weak radial fold. Lunule broad, moderately excavated. Dissoconch sculptured with fine, regular comarginal lines, originating deeply impressed growth disruptions. Shell surface creamy white, with anterior and posterior rusty incrustations. Periostracum yellowish. Hinge plate narrow. Left valve with two divergent cardinal teeth: the anterior strong, triangular, high; the posterior narrowly elongate, a half the length of the anterior one. Right valve with a strong, triangular, relatively high cardinal tooth. Both valves with small, short, similar in morphology anterior and posterior lateral teeth. Nymph long, narrow, shallow. Inner shell surface whitish, glossy outside pallial line. Inner margin finely crenulate. Anterior adductor muscle scar long, broad, detached from pallial line for a half of its length; posterior adductor muscle scar ovate, smaller than the anterior one.

Etymology: Named for our colleagues John D. Taylor and Emily A. Glover in recognition of their remarkable contributions to our understanding of Lucinidae phylogeny and systematics.

Comments: To date, only seven living species of Cavilinga are known worldwide (MolluscaBase, 2024, with three of them occurring in the Eastern Pacific Ocean (C. lampra (Dall, 1901), C. lingualis (Carpenter, 1864), and C. prolongata (Carpenter, 1857)), three in the Atlantic Ocean (C. compacta (Smith, 1890), C. inconspicua (Smith, 1890) and C. blanda (Dall & Simpson, 1901)), and one in the Indian Ocean (C. fieldingi (Adams, 1871)). The presence of a short and horizontal antero-dorsal margin, and a broadly rounded anterior margin, make Cavilinga taylorgloverorum n. sp. closely similar to C. prolongata. However, the latter differs from C. taylorgloverorum n. sp. by having the anterior end less projected, the shell higher than long, more prominent umbos, and the dissoconch with stronger comarginal sculpture. Cavilinga lampra, C. lingualis, C. blanda, C. inconspicua, and C. fieldingi differ strikingly from C. taylorgloverorum n. sp. by having the antero-dorsal margin longer and steeply sloping, resulting in shell outlines that are higher than long or as high as long. Cavilinga inconspicua differs by a more rounded shell outline, with wider, subcentrally located umbos.

THYASIRIDAE

Thyasira fernandezensis n. sp.

Fig. 15

Figure 15 Family Thyasiridae.

(A–F) Thyasira fernandezensis n. sp.; (A, B) Holotype, LACM 3828; (C) Paratype, LACM 3829; (D–F) (SEM): USNM 1548477. Scale bars: (A–D) = 1 mm; (E, F) = 500 µm.

Type locality: 33°38′ S, 78°50′ W, off Robinson Crusoe Island, Juan Fernández, 62 m (R/V Anton Bruun Cruise 17, station 680E, by Campbell grab, 18 July 1966; LACM 1966-100).

Type material: Holotype (LACM 3828: 1 v, Figs. 15A, 15B) and 35 paratypes (LACM 3829: 31 vs [and fragments], with Fig. 15C; FMNH 312474 ex LACM 3829: 2 vs; MNHN-CL MOL 101622 ex LACM 3829: 2 vs) from the type locality.

Other material studied: Juan Fernández: Alejandro Selkirk: Eltanin Station 21-203 (USNM 1548477: 10 vs, with Figs. 15D–15F).

Distribution: Only known from Juan Fernández archipelago.

Diagnosis: Shell subovate, with projected anterior end, short antero-dorsal margin and markedly flattened postero-ventral margin.

Description: Shell up to 2.2 mm L, subovate, longer than high, only slightly inflated, markedly inequilateral, thin. Anterior end projecting; posterior end short, truncated. Umbos small, pointed, posteriorly located, prosogyrate. Antero-dorsal margin short, nearly straight, horizontal. Anterior margin widely curved, forming weak angulation at the junction with dorsal margin; not clearly separated from ventral margin, which is evenly arched. Postero-ventral margin flat, forming a prominent angulation at the junction with posterior margin. Posterior margin slightly sinuated. Postero-dorsal margin short, straight, steeply sloping. Posterior area of shell flat. First posterior fold strong. Second posterior fold weak. Submarginal sulcus well marked, but narrow. Auricle extending along the entire length of submarginal sulcus. Lunule ill defined. Prodissoconch ovate, of about 150 μm in diameter, smooth. Dissoconch with low growth lines. Shell surface whitish, glossy. Anterior and posterior ends encrusted with ferruginous material. Hinge plate narrow, with prominent pseudocardinal tubercle in left valve, and a smaller tubercle in the right valve. Resilifer elongate, short.

Etymology: Named for the type locality in the Juan Fernández archipelago; adjective.

Comments: Thyasira fernandezensis n. sp. closely resembles the Antarctic and sub-Antarctic T. debilis (Thiele, 1912) (figured by Zelaya, 2009) from which it however differs by having a more projected anterior shell end, and a more markedly flattened postero-ventral margin. In addition, the posterior shell auricle in T. fernandezensis n. sp. is more developed than in T. debilis, and the hinge tubercle appears over the left (instead of the right) valve. Thyasira fernandezensis n. sp. also resembles smaller specimens of Thyasira succisa (Jeffreys, 1876) as illustrated by Oliver & Killeen (2002: pl. 19), although the latter has a much longer antero-dorsal shell margin, resulting in a more projected anterior end, shorter auricle, and stronger pseudocardinal tubercle. In addition, T. succisa and T. fernandezensis n. sp. show allopatric distribution, with the former occurring in the northeastern Atlantic Ocean and the Mediterranean (MolluscaBase, 2024).

LASAEIDAE

Kellia tumbesiana (Stempell, 1899)

Figs. 16A–16D

Figure 16 Family Lasaeidae.

(A–D) Kellia tumbesiana (Stempell, 1899), LACM 165-100.2. (E–K) Lasaea macrodon Stempell, 1899; (E, F) Syntype of Lasaea macrodon Stempell, 1899, ZMB 51987; (G) LACM 1965-102.1; (H, I) LACM 1965-94.4; (J, K) (SEM): FMNH 322339. (L–R) Malvinasia selkirkensis n. sp.; (L, M) Holotype, MNHN-CL MOL 101625; (N–Q) FMNH 322335; (R) FMNH 322334. (S–Y): Melliteryx platei (Stempell, 1899); (S, T) Syntype of Lepton platei Stempell, 1899, ZMB 51986; (U–X, Y) (SEM): FMNH 322255. (Z–EE) Tellimya crusoensis n. sp.; (Z, AA) Holotype, MNHN-CL MOL 101627; Paratypes, (BB, CC) FMNH 327364; (DD, EE) FMNH 327366. Scale bars: (A–D) = 5 mm; (E–I, S-X, Y, Z–EE) = 2 mm; (J–R) = 1 mm.

Diplodontina tumbesiana Stempell, 1899: 232, pl. 12, figs. 18, 19, 19a.

Kellia tumbesiana, – Dall, 1909: 264 (listed only). Soot-Ryen, 1959: 50, text fig. 4. Osorio & Bahamonde, 1970: 200 (listed only). Bernard, McKinnell & Jamieson, 1991: 36 (listed only). Guzmán, Saá & Ortlieb, 1998: 67–68. Valentich-Scott, Coan & Zelaya, 2020: 233, pl. 70 (holotype).

Telimya (Diplodontina) tumbeziana (sic), – Carcelles & Williamson, 1951: 340 (listed only).

Kellia cf. tumbesiana, – Marincovich, 1973: 11–12, figs. 12–14.

Diplodontina tumbesiana, – Bernard, 1983: 19 [listed only]. Paredes & Cardoso, 2004: 210–212, fig. 2.

Type locality: Halbinsel Tumbes bei Talcahuano [36°36′ S, 73°00′ W, Caleta Tumbes, Talcahuano, Chile].

Type material: Holotype (ZMB-Moll 51985).

Other material studied: Juan Fernández: Robinson Crusoe: LACM 1965-100.2 (1 av, 7 vs, with Figs. 16A–16D).

Distribution: Callao, Lima, Perú (12°06′ S) to Península de Tumbes, Bío Bío, Chile (36°36′ S) (Valentich-Scott, Coan & Zelaya, 2020); and Juan Fernández.

Description: Shell up to 10 mm L, ovate, longer than high, somewhat inflated, subequilateral, thin. Anterior end somewhat projected. Umbo small, low, subcentral, prosogyrate. Antero-dorsal, anterior, ventral, and posterior margins forming a continuous, ovoidal curve. Postero-dorsal margin rapidly sloping, straight to slightly arched. Dissoconch only sculptured with low growth lines. Shell surface shiny, whitish. Periostracum yellowish. Hinge plate extremely weak; hinge teeth and resilifer appearing as hanging from dorsal shell margin. Left valve with two divergent cardinal teeth and an elongated posterior lateral tooth. Anterior cardinal antero-ventrally directed, posterior cardinal ventrally directed, half the size of anterior cardinal; both cardinals in contact at their bases. Right valve with a strong, triangular cardinal tooth, and a thin, elongated, posterior lateral tooth. Resilifer elongated, posterior to the umbo. Inner shell surface whitish. Inner margin smooth.

Lasaea macrodon Stempell, 1899

Figs. 16E–16K

Lasaea macrodon Stempell, 1899: 231, pl. 12, figs. 16, 17.

Lasaea macrodon, – Keen, 1938: 22 [listed only]. Soot-Ryen, 1959: 51 [listed only]. Osorio & Bahamonde, 1970: 201 [listed only]. Ponder, 1971: 133 [listed only]. Bernard, 1983: 31 [listed only]. Rozbaczylo & Castilla, 1987: 176 [listed only]. Bernard, McKinnell & Jamieson, 1991: 36 [listed only]. Ramírez & Osorio, 2000: 6 [listed only].

Lasaea petitiana, – Dall, 1909: 264 [listed only]. Bernard, McKinnell & Jamieson, 1991: 36 [listed only]. Valentich-Scott, Coan & Zelaya, 2020: 227, pl. 72 [syntype of Lasaea macrodon]. (Non Récluz, 1842).

Type locality: Bahía Padres, [33°40′45″ S, 78°56′45″ W, Robinson Crusoe Island], Juan Fernández [archipelago].

Type material: three syntypes (fide Stempell, 1899), although only one of them (the syntype figured by Stempell, 1899) is currently preserved (C. Zorn in litt., July 2022; ZMB 51987: 1 v, Figs. 16E, 16F).

Other material studied: Desventuradas: San Félix: LACM 1965-94.4 (1 av, 1 v, with Figs. 16H, 16I). Juan Fernández: Alejandro Selkirk: IOC97-50A (FMNH 322339: 2 vs, Figs. 16J, 16K). Robinson Crusoe: IOC97-57A (FMNH 322342: 1 v); IOC97-66A (FMNH 322340: 1 v; MNHN-CL MOL 101624 ex FMNH 327903: 2 vs); LACM 1965-102.1 (6 avs, 1 v, with Fig. 15G).

Other published records: Playa “El Palillo”, Robinson Crusoe, Juan Fernández (Ramírez & Osorio, 2000).

Distribution: Only known from Juan Fernández and Desventuradas archipelagos.

Description: Shell to 5.5 mm L, ovate, longer than high, inflated, markedly inequilateral, thick. Anterior end widely projected. Umbo low but wide, posteriorly located, prosogyrate. Antero-dorsal margin not clearly separated from anterior margin, forming a wide curve. Ventral margin gently arched. Posterior margin relatively short, curved. Postero-dorsal margin steeply sloping, straight to somewhat arched, forming an obscure angulation at the junction with posterior margin. Dissoconch sculptured with conspicuous comarginal lines and microscopic pits. Shell surface shiny, whitish to reddish. Hinge plate short but stout, deeply excavated below the umbo; with one anterior and one posterior lateral tooth, similar in length. Lateral teeth short, elongate, strong; the anterior one fused to a small cardinal tooth. Resilifer long and wide. Inner shell surface withish. Inner margin smooth.

Comments: Stempell (1899) introduced the name Lasaea macrodon for specimens collected in the Juan Fernández archipelago. Stempell’s name was subsequently accepted as valid (e.g., Soot-Ryen, 1959; Osorio & Bahamonde, 1970; Rozbaczylo & Castilla, 1987; Bernard, McKinnell & Jamieson, 1991; Ramírez & Osorio, 2000; MolluscaBase, 2024). Dall (1909), in a checklist of the mollusks of the Peruvian province, included the Juan Fernández archipelago in the distributional range of L. petitiana, likely resulting from considering L. macrodon as a synonym of L. petitiana, a synonymy also followed by Valentich-Scott, Coan & Zelaya (2020). Bernard, McKinnell & Jamieson (1991) listed Lasaea macrodon from Juan Fernández but Lasaea petitiana from Desventuradas. The study of these specimens (housed at LACM) revealed that they correspond to the same species. In view of the scarcity of available material of Lasaea from these archipelagoes Islands, consisting of empty shells only, and the general difficulties for properly delineating Lasaea species (which are known for polyploidy), we conservatively retain as valid the local name provided by Stempell (1899). Ponder (1971) suggested that Lasaea helenae Soot-Ryen, 1957, described from Iquique, Chile, is possibly a synonym of L. macrodon.

Stempell (1899) referred to three specimens in his original description, which qualify as syntypes. The mention of a “holotype” by Valentich-Scott, Coan & Zelaya (2020: 228) was erroneous.

Malvinasia selkirkensis n. sp.

Figs. 16L–16R

Type locality: 33°45′37.14″ S, 80°45′8.13″ W, E side of Alejandro Selkirk Island, Juan Fernández, 25 m (sediment sample collected from sand pockets by R. Bieler, 6 March 1997; IOC97-50A).

Type material: Holotype (MNHN-CL MOL 101625 ex FMNH 322333: 1 v, Figs. 16L, 16M), and three paratypes, Juan Fernández: Robinson Crusoe: IOC97-66A (FMNH 322334: 1 av, Fig. 16R); IOC97-68A (FMNH 322335: 2 vs, Figs. 16N–16Q).

Distribution: Only known from Juan Fernández archipelago.

Diagnosis: Shell minute, low, elongate. Anterior end markedly projected, pointed. Umbos low, strongly displaced posteriorly.

Description: Shell to 1.8 mm L, trigonal-ovate, longer than high, somewhat inflated, markedly inequilateral, thin. Anterior end widely projected, pointed. Umbo extremely low, not outstanding from shell margin, markedly posteriorly displaced, opisthogyrate. Antero-dorsal margin long, slightly to markedly convex. Anterior, ventral and posterior margins not distinctly separated, forming a uniform curve. Postero-dorsal margin short, sloping straight to slightly concave, sloping slightly more steeply than antero-dorsal margin. Dissoconch sculptured with low, irregularly spaced comarginal ridges; usually with 3–4 stronger growth folds. Shell surface whitish. Periostracum thick, straw yellowish. Hinge plate relatively weak, appearing as cleft below umbo. Right valve with long, strong peg-like tooth anterior to ligament, overhanging from hinge plate; dorsal margin anterior and posterior to ligament forming shallow, narrow grooves to receive left dorsal margin. Left valve with deep triangular socket where right tooth fits; antero-dorsal and postero-dorsal margins enlarged forming strong, long but narrow ridges. Inner shell surface whitish. Inner margin smooth.

Etymology: Named for the type locality, Alejandro Selkirk Island; adjective.

Comments: Malvinasia selkirkensis n. sp. is most similar to Malvinasia piccola Ituarte & Zelaya, 2015, from which it differs by having a lower and more elongated shell outline, an anterior end that is more projected and pointed, and much lower umbos. This set of characters also allows the distinction of Malvinasia selkirkensis n. sp. from M. arthuri Cooper & Preston, 1910 and Malvinasia cf. arthuri of Ituarte & Zelaya (2015). The low and elongate shell outline of Malvinasia selkirkensis n. sp. also resembles that figured by Ituarte & Zelaya (2015) as Malvinasia sp. 1, from which it differs by having a shorter posterior end with faster-sloping postero-dorsal margin. Another low-umboned species is Malvinasia molinae (Ramorino, 1968), although that species is much higher-shelled and has a shorter and more rounded anterior end than M. selkirkensis n. sp.

Melliteryx platei (Stempell, 1899)

Figs. 16S–16Y

Lepton platei Stempell, 1899: 233, pl. 12, figs. 20, 21. Kaspar, 1913: 545–625 [anatomy].

Bornia platei, – Dall, 1909: 263 [listed only]. Soot-Ryen, 1959: 49 [listed only]. Osorio & Bahamonde, 1970: 200 [listed only]. Rozbaczylo & Castilla, 1987: 176 [listed only].

Erycina platei, – Bernard, 1983: 31 [listed only]. Bernard, McKinnell & Jamieson, 1991: 36 [listed only].

? Melliteryx ? n. sp., Bernard, McKinnell & Jamieson, 1991: 36 [listed only].

Litigiella platei, – Huber, 2010: 550.

Melliteryx platei, – Valentich-Scott, Coan & Zelaya, 2020: 229, pl. 73 (syntype).

Type locality: Bahía Padres, (33°40′45″ S, 78°56′45″ W, Robinson Crusoe Island), Juan Fernández (archipelago).

Type material: 9 syntypes (fide Stempell, 1899), although only one of them is currently preserved (C. Zorn, in litt, July 2022; ZMB 51986: 1 av, Figs. 16S, 16T).

Other material studied: Desventuradas: San Félix: IOC97-26 (FMNH 322310: 1 v); IOC97-29 (FMNH 322309: 1 v); IOC97-32 (FMNH 322307: 1 v). San Ambrosio: IOC97-13 (FMNH 322308: 3 vs); IOC97-18 (FMNH 322256: 10 vs). Juan Fernández: Alejandro Selkirk: IOC97-48A (FMNH 322260: 35 vs); IOC97-50A (FMNH 322257: 4 avs, 41 vs). Robinson Crusoe: IOC97-44A (FMNH 322255: 5 avs, 43 vs, with Figs. 16U–16Y; MNHN-CL MOL 101626 ex FMNH 327904: 25 vs); IOC97-57A (FMNH 322258: 18 vs); IOC97-66A (FMNH 322259: 2 avs, 72 vs); IOC97-68A (FMNH 322261: 1 av, 17 vs).

Other published records: Bahía Padres, Juan Fernández (Kaspar, 1913); Juan Fernández (Bernard, McKinnell & Jamieson, 1991).

Distribution: Juan Fernández and Desventuradas archipelagos, and Caldera, Atacama, Chile (Valentich-Scott, Coan & Zelaya, 2020).

Description: Shell to 4 mm L, subquadrate, longer than high, inflated, subequilateral, moderately solid. Anterior end as high as posterior end. Umbo low, wide, subcentrally located to slightly posteriorly displaced, orthogyrate. Antero-dorsal and postero-dorsal margins sloping at similar angle, straight to slightly arched. Posterior margin height, obliquely straight or forming a continuous curve with postero-dorsal margin, in the first case originating a weak angulation at the junction with dorsal margin. Ventral margin only slightly curved. Anterior margin widely rounded, not clearly separated from dorsal and ventral margins. Dissoconch sculptured with shallow, microscopic punctae, forming a honeycomb pattern, and densely packed, regularly distributed comarginal cords. Comarginal sculpture more evident ventrally. Shell surface withish, brilliant. Periostracum yellowish. Hinge plate markedly excavated below the umbo, stout anteriorly, narrower posteriorly; with a small cardinal and two lateral teeth in each valve. Cardinal tooth small, triangular, ventrally directed. Anterior and posterior lateral teeth straight, extending all along hinge plate height, similar in length; the anterior stronger than the posterior one. Resilifer elongate, on the margin of posterior lateral tooth, supporting an internal ligament. External ligament present. Inner shell surface whitish, reflecting outer comarginal sculpture.

Tellimya crusoensis n. sp.

Figs. 16Z–16EE

Type locality: 33°38′27.6″ S, 78°49′22.8″ W, Cumberland Bay, Robinson Crusoe, 14 m (sediment sample collected from sand pockets by R. Bieler, 4 March 1997; IOC97-44A).

Type material: Holotype (MNHN-CL MOL 101627 ex FMNH 327365: 1 v, Figs. 16Z, 16AA), and two paratypes, Juan Fernández: Alejandro Selkirk: IOC97-50 (FMNH 327366: 1 v, Figs. 16DD, 16EE); IOC97-50A (FMNH 327364: 1 v, Figs. 16BB, 16CC).

Distribution: Only known from Juan Fernández archipelago.

Diagnosis: Shell ovate-elongate, with long posterior flap. Umbos wide, orthogyrate. Left valve with short, postero-ventrally directed anterior tooth.

Description: Shell to 9.1 mm L, ovate, longer than high, somewhat inflated, markedly inequilateral, thin. Anterior end greatly produced, as high as posterior end. Umbo low but wide, posteriorly displaced, orthogyrate. Antero-dorsal margin long, straight to slightly arched, sloping slowly. Anterior, ventral and postero-ventral margins uniformly curved. Postero-dorsal margin obliquely straight. Postero-dorsal margin short, obliquely sloping, forming a prominent angulation at the junction with posterior margin. Posterior flap low but long. Prodissoconch I brownish, only preserved in smaller specimens; worn away by erosion and producing an umbonal cleft in larger specimens. Dissoconch sculptured with regularly distributed comarginal threads, and numerous, discontinuous radial lines. Shell surface withish, shiny, translucent, sometimes crusted with ferruginous material. Periostracum yellowish. Hinge plate extremely narrow. Left valve with a single, short, stout, postero-ventrally directed anterior tooth, overhanging from hinge plate. Resilifer long, broad, spoon-shaped. Inner shell surface whitish. Inner shell margin smooth.

Etymology: Named for the type locality, Robinson Crusoe Island; adjective.

Comments: Tellimya crusoensis n. sp. is most similar to T. pauciradiata Raines & Huber, 2012, from which it differs by having wider umbos, longer posterior flap, and a greater number of radial lines. In addition, the anterior tooth is ventrally directed in T. pauciradiata and postero-ventrally directed in Tellimya crusoensis n. sp.

In general shell outline, Tellimya crusoensis n. sp. also resembles Tellimya ferruginosa, from which it differs by having lower and orthogyrate (instead of opisthogyrate) umbos. Furthermore, in Tellimya crusoensis n. sp. the left valve anterior tooth is short, postero-ventrally directed and overhanging from hinge plate, while in T. ferruginosa, this tooth is longer, antero-ventrally directed and completely supported by the hinge plate (see Gofas, 2000: fig. 6; Kamenev, 2008: fig. 5). Another similar species is Tellimya auporia (Ponder, 1968), which differs by having a less projected anterior end than Tellimya crusoensis n. sp., resulting in higher shell outlines. Furthermore, T. auporia has fewer radial lines than Tellimya crusoensis n. sp.

CONDYLOCARDIIDAE

Condylocardia angusticostata n. sp.

Figs. 17A–17H

Figure 17 Family Condylocardiidae.

(A–H) Condylocardia angusticostata n. sp.; (A, C) (SEM) Holotype, MNHN-CL MOL 101628; (B) (SEM) Paratype, MNHN-CL MOL 101629; (D–G) Paratypes, FMNH 322251; (H) FMNH 322252. (I–N) Condylocardiidae sp. A (I, K, L) (K, L, SEM) FMNH 322312; (J, M, N) (M, N, SEM) FMNH 322253. Scale bars: (A, B) = 2 mm; (C–J) = 1 mm; (K–M) = 500 µm; (N) = 100 µm.

Condylocardia n. sp., Bernard, McKinnell & Jamieson, 1991: 36 [listed only].

Type locality: 33°40′20″ S, 78°56′27″ W, SW of Robinson Crusoe Island, Juan Fernández, 17–18 m (sediment sample collected from sand patches by R. Bieler, 8 March 1997; IOC97-66A).

Type material: Holotype (MNHN-CL MOL 101628 ex FMNH 327992: 1 v, Figs. 17A, 17C), and 17 paratypes; 11 from the type locality (MNHN-CL MOL 101629 ex FMNH 327993: 1 v, Fig. 17B; FMNH 322251: 10 vs, with Figs. 17D–17G) and 6 from 66-100 (LACM 3838: 6 vs).

Other material studied: Juan Fernández: Alejandro Selkirk: Eltanin Station 21-203 (USNM 1548482: 1 av). Robinson Crusoe: IOC97-44A (FMNH 322254: 2 avs, 1v); IOC97-68A (FMNH 322252: 1 av, 1 v, with Fig. 17H).

Distribution: Only known from Juan Fernández archipelago.

Diagnosis: Shell minute, triangular, sculptured with 9–13 radial ribs, separated by interspaces wider than ribs width.

Description: Shell up to 1.7 mm H, triangular, as high as long, compressed to slightly inflated, inequilateral, moderately solid. Umbo prominent, pointed, anteriorly located. Antero-dorsal and postero-dorsal margins sloping at similar angle; the former straight to somewhat concave, the latter markedly straight, longer than the anterior one. Ventral margin widely convex, producing weak angulations in the junction with antero-dorsal and postero-dorsal margins. Lunule and escutcheon elongated, relatively wide, deep. Prodissoconch of about 400 µm, mamillated; P-1 with a central depression. P-2 sculptured with weak radial sculpture; separated from teleoconch by a strong rim, which is higher posteriorly than anteriorly. Dissoconch sculptured with 9–13 narrow radial ribs, separated by wider interspaces. In addition, with fine comarginal lamellae, some forming folds. Shell surface whitish, glossy. Periostracum thin, ambarine. Hinge plate narrow. Right valve with a minute, trigonal posterior cardinal tooth and a larger, arched anterior cardinal; the latter with the posterior branch stronger than the anterior branch; both anterior and posterior cardinals dorsally attached to hinge line. Anterior lateral tooth narrow, elongate, well-discernible from shell margin. Posterior lateral tooth indistinct. Left valve with a free, short but stout, obliquely directed and well separated from hinge line anterior cardinal tooth, a massive posterior cardinal tooth, and an elongated posterior lateral tooth. Anterior lateral tooth indistinct. Internal ligament small, on a small resilifer, between anterior and posterior cardinal teeth. External ligament absent. Inner shell surface reflecting outer sculpture. Inner margin undulated by radial sculpture.

Etymology: The name of the species refers to the narrow radial sculpture; adjective.

Comments: In general shell outline, Condylocardia angusticostata n. sp. closely resembles the Australian Condylocardia limnaeformis Cotton, 1930 (figured by Middelfart, 2002: fig. 4) and some specimens of Condylocardia digueti Lamy, 1917, particularly those from Baja California, US (figured by Coan, 2003: figs. 1–3, 5–7). However, the new species described herein strikingly differs from these two species by having much narrower radial ribs that are separated by wider interspaces. In addition, the radial ribs in C. digueti are crossed by strong comarginal bars, which are virtually absent in Condylocardia angusticostata.

Recent studies have questioned the status of Condylocardiidae (see Passos, Batistão & Bieler, 2021) after multilocus molecular studies (González & Giribet, 2015; Combosch et al., 2017) found members of Carditopsis E. A. Smith, 1881 and Carditella E. A. Smith, 1881–two genera historically considered as condylocardiids–nested within the family Carditidae. However, the name-bearing genus Condylocardia has yet to be analyzed.

Condylocardiidae sp. A

Figs. 17I–17N

Material studied: Desventuradas: San Félix: IOC97-26 (FMNH 322312: 1 v, Figs. 17I, 17K, 17L); IOC97-29 (FMNH 322253: 1 v, Figs. 17J, 17M, 17N).

Distribution: Only known from Desventuradas archipelago.

Description: Shell up to 1.48 mm L, ovate, longer than high, inequilateral, solid. Umbo prominent, widely rounded, posteriorly located. Antero-dorsal and postero-dorsal margins sloping at similar angles, straight to somewhat concave, the anterior longer than the posterior one. Ventral margin widely convex, not clearly separated from anterior and posterior margins in smaller specimens, producing weak angulations in the junction with antero-dorsal and postero-dorsal margins. Posterior margin flattened in larger specimens. Prodissoconch of about 300 µm in diameter, mamillated; P-1 with two radially elongate depressions. P-2 separated from teleoconch by a prominent rim, which is higher posteriorly than anteriorly. Dissoconch sculptured with 10 broad and low radial ribs, separated by interspaces narrower than ribs width. In addition, fine comarginal lamellae, forming projected flat scales on the radial ribs. Shell surface whitish. Hinge plate narrow. Left valve with elongated anterior and posterior lateral teeth. Cardinal teeth imperfectly preserved in the available material. Internal ligament small, on a small resilifer. Inner shell surface and inner margin smooth.

Comments: Condylocardiidae sp. A appears to be a new species. The limited material, two isolated shell valves in poor condition, precludes us from naming it herein. Condylocardiidae sp. A resembles some species of Carditella considered by Güller & Zelaya (2013) and some species assigned to Condylocardia by Coan (2003). The imperfect state of preservation of the hinge plate of the available material (namely, the number and arrangement of cardinal teeth) precludes us from confirming the generic placement of this material. In general shell outline, it is most similar to Condylocardia koolsae Coan, 2003, from Galapagos Islands, Equator. However, Condylocardiidae sp. A clearly differs from that species by having a smaller number of broader radial ribs on the dissoconch (10 vs. 15–16) and displaying a smooth inner shell surface and margin.

PSAMMOBIIDAE

Gari sp. B

Fig. 18

Figure 18 Family Psammobiidae.

(A, B) Gari sp. B LACM 1965-101.6. Scale bar: (A, B) = 1 cm.

Gari solida (Gray, 1828–1830), – Bernard, McKinnell & Jamieson, 1991: 36 [listed only]. Valentich-Scott, Coan & Zelaya, 2020: 260 (in part: Juan Fernández record). (Non Gray, 1828–1830).

Material studied: Juan Fernández: Robinson Crusoe: LACM 1965-101.6 (1 v, Figs. 18A, 18B).

Distribution: Only known from Juan Fernández archipelago.

Description: Shell of 29.6 mm L, ovate-elongate, longer than high, compressed, subequilateral, thin. Anterior end as high as posterior end. Umbo minute, pointed, slightly anteriorly displaced. Antero-dorsal margin sloping straight. Anterior margin rounded. Ventral margin evenly arcuated. Posterior margin obliquely straight. Postero-dorsal margin straight, almost horizontal. Posterior area of the shell ill defined. Dissoconch sculptured with weak comarginal lines. Shell surface whitish, shiny; stained with reddish. Hinge plate narrow. Right valve with two small, divergent cardinal teeth, the anterior one stronger, bifid, and ventrally directed; the posterior one postero-ventrally directed. Nymph short. Inner shell surface whitish, stained in reddish. Inner margin smooth. Pallial sinus broad, deep, completely separated from pallial line, anteriorly directed. Anterior adductor muscle scar ovoid, ventrally directed; posterior adductor muscle scar subcircular, dorsally displaced.

Comments: The specimen of Gari sp. B studied herein was previously identified by Bernard, McKinnell & Jamieson (1991) as Gari solida, listing an incorrect locality of “Isla San Félix, Desventuradas”. Based on the same material, Valentich-Scott, Coan & Zelaya (2020) also reported Gari solida, but from Juan Fernández (provenance confirmed herein by the specimen label). The nominal species Gari solida represented thus far the only species of that genus from Chilean waters. However, Gari solida strikingly differs from Gari sp. B by its more rapidly sloping antero-dorsal and postero-dorsal margins and higher shell, resulting in a more triangular shell outline. Moreover, in Gari solida the ventral part of the pallial sinus is convergent with the pallial line for about one-third of its total length, and the nymph is narrower and longer.

Coan (2000) revised the species of Gari occurring in the Eastern Pacific. In general shell outline, the Juan Fernández specimen is most similar to the Galapagean specimen identified by Coan (2000) as Gari new species A and small specimens of Gari helenae Olsson, 1961 (both of them also figured by Coan & Valentich-Scott, 2012). These two species however differ from the material studied herein by having a longer antero-dorsal margin, which consequently results in a more posteriorly placed umbo, and by having the ventral part of the pallial sinus confluent for more than a half of its length with the pallial line. Furthermore, Gari new species A of Coan (2000) has a higher posterior end and more arcuated ventral margin than the specimen referred herein as Gari sp. B.

Gari sp. B appears to be a new, undescribed, species. However, the limited material we have available (a single valve) and its poor state of preservation, preclude us from naming it herein.

SEMELIDAE

Ervilia producta Odhner, 1922

Fig. 19

Figure 19 Family Semelidae.

(A–I) Ervilia producta Odhner, 1922; (A–E) Syntypes, GNM Moll. 7066; (F–I) FMNH 322240. Scale bar: (A–I) = 2 mm.

Ervilia producta Odhner, 1922: 222, pl. 8, figs. 11, 12.

Ervilia producta, – Soot-Ryen, 1959: 65 [listed only]. Osorio & Bahamonde, 1970: 205 [listed only]. Bernard, 1983: 41 [listed only]. Rozbaczylo & Castilla, 1987: 176 [listed only]. Bernard, McKinnell & Jamieson, 1991: 36 [listed only]. Sandberg & Warén, 1993: 128 [list of Odhner’s taxa]. Huber, 2010: 703. ? Valentich-Scott, Coan & Zelaya, 2020: 266, pl. 85.

Ervilia galapagana, – Bernard, McKinnell & Jamieson, 1991: 36 [listed only]. (Non Dall & Ochsner, 1928).

Type localities: Masatierra [=Robinson Crusoe Island, Juan Fernández archipelago], 20–35 m and 30–45 m.

Type material: Syntypes at SMNH and GNM (Sandberg & Warén, 1993); six syntypes studied herein (GNM Moll. 7066: 2 avs, 2 vs, with Figs. 19A–19E).

Other material studied: Desventuradas: San Félix: IOC97-26 (FMNH 322311: 1 av, 2 vs); IOC97-29 (FMNH 322300: 7 vs); LACM 1966-98.5 (1 v). Juan Fernández: Alejandro Selkirk: IOC97-50A (FMNH 322243: 4 avs, 1 v); 33°45′ S, 80°40′48″ W, 79–91 m (USNM 887915: 1 av, 38 vs; USNM 904373: 15 avs, 61 vs). Robinson Crusoe: IOC97-66A (FMNH 322240: 3 avs, 3 vs, with Figs. 19F–19I; MNHN-CL MOL 101630 ex FMNH 327994: 2 avs, 7 vs); IOC97-66A (FMNH 322241: 2 avs); IOC97-68A (FMNH 322242: 2 avs, 7 vs); LACM 1965-100.3 (4 vs); LACM 1965-101.7 (1 v); LACM 1966-100.5 (>100 vs); LACM 1966-102.3 (4 vs); LACM 1966-103.1 (35 vs).

Distribution: Only known from Juan Fernández and Desventuradas archipelagos.

Description: Shell to 8.4 mm L, ovate, longer than high, compressed, inequilateral, thin. Anterior end broadly rounded, posterior end bluntly pointed, lower than anterior one. Umbo small, narrow, pointed; subcentral to slightly anteriorly displaced, opisthogyrate. Antero-dorsal margin sloping straight and forming a weak angulation at the junction with the rounded anterior margin, or markedly arcuate, connected in a broad curve with anterior margin. Ventral margin evenly arched or flattened in the posterior half, with the maximum projection in the medial line. Posterior margin short and rounded, forming weak angulation at the junction with postero-dorsal margin, which may be either straight and obliquely sloping, or convex. Postero-dorsal margin longer than antero-dorsal margin. Escutcheon wide and deep. Prodissoconch pale cream. Dissoconch sculptured with numerous, low, regularly distributed comarginal cords, separated by interspaces as wide as the cords; posterior third of the shell also bearing prominent radial lines. Lines extending from umbo to shell margin, forming granules at the intersection with comarginal sculpture. Shell surface translucent, whitish, shiny; sometimes stained with violet, brownish or yellowish. Hinge plate narrow at anterior and posterior ends, but widened below the umbos. Right valve with a triangular, high and stout median cardinal, anterior to the resilifer. Anterior and posterior cardinal teeth merged to shell margin. Anterior and posterior lateral teeth similar in size, narrow, elongate, separated by a shallow depression from shell margin. Left valve with three straight cardinal teeth: posterior cardinal short and stout; anterior and median cardinals narrow, longer than the posterior cardinal. Anterior tooth antero-ventrally directed, median tooth ventrally directed. Anterior and ventral teeth delimiting a socket where the median tooth of the opposite valve articulates. Median and posterior teeth delimiting the resilifer. Resilifer large, triangular. Internal ligament massive. External ligament narrow, amphidetic, longer anteriorly than posteriorly. Inner shell surface whitish or stained as the outer shell surface. Inner margin smooth. Pallial sinus broad, deep, reaching the half of shell length; base fused to pallial line for three-fourth of its total length.

Comments: Odhner (1922) mentioned “many specimens” in the original diagnosis, without selecting a holotype. Four of these syntypes were studied herein.

Bernard, McKinnell & Jamieson (1991: 36) listed Ervilia producta and Ervilia galapagana Dall & Ochsner, 1928 from Juan Fernández. However, the specimens they identified as E. galapagana strikingly differ from the holotype of that species (CAS 2971) by being higher and by having the umbos narrower and pointed. These specimens fit within the variability of the syntypes of E. producta, and we interpret (Bernard, McKinnell & Jamieson’s (1991) records of E. galapagana in Juan Fernández as based on a misidentification.

Valentich-Scott, Coan & Zelaya (2020) identified as Ervilia producta an eroded specimen from 12°05′ S, 77°08′ W, off Callao, Lima, Perú, 88 m. The provided SEM micrographs (Valentich-Scott, Coan & Zelaya, 2020: pl. 85) show straighter antero-dorsal and postero-dorsal shell margins than in any of the Juan Fernández specimens studied herein. The comarginal and radial sculpture is not visible in the corroded figured specimen. An assessment of the conspecificity of the Peruvian shell, and the associated wider geographic range, needs access to further material.

CHAMIDAE

Chama pellucida Broderip, 1835

Fig. 20

Figure 20 Family Chamidae.

(A–N) Chama pellucida Broderip, 1835; (A–D) Syntypes, NHMUK 1950.11.1.63-65; (E–G) LACM 1965-99.3, (H–K) LACM 1965-103.3; (L–N) Holotype of Chama chilensis Philippi, 1887, SGO.PI.632; (N) Original label. Scale bars: (A–J, L, M) = 1 cm; (K) = 5 mm.

Chama pellucida Broderip, 1835a: 149; 1835b: 302–303, pl. 38, fig. 3.

Chama pellucida, – Hanley, 1842–1846: 228, pl. 24, fig. 19. d’Orbigny, 1834–1848: 670–671. Chenu, 1846: pl. 6, fig. 12. Reeve, 1846–1847: Chama species 32, pl. 6, fig. 32 [topotypic specimen; Mus. Cuming]. Philippi, 1860: 177. Clessin, 1888–1889: 18–19, pl. 8, figs. 3, 4. Stempell, 1899: 238. Dall, 1909: 156, 262 [listed only]. Grieser, 1913 [anatomy]. Lamy, 1928: 343 [review of genus]. Pilsbry & Lowe, 1934: 82. Soot-Ryen, 1959: 40 [listed only; coastal Chile]. Olsson, 1961 [in part: Perú record]: 225, pl. 33, figs. 2, 2a, pl. 34, fig. 5. Herm, 1969: 115, pl. 5, fig. 6. Osorio & Bahamonde, 1970: 198 [listed only; in part: Juan Fernández records]. Marincovich, 1973: 11, fig. 10. Bernard, 1976: 20, figs. 4C [topotypic specimen], 4d [holotype of Chama chilensis]; 1983: 35 [listed only]. Rozbaczylo & Castilla, 1987: 176 [listed only]. Bernard, McKinnell & Jamieson, 1991: 36 [listed only]. Alamo & Valdivieso, 1997: 115 [listed only]. Guzmán, Saá & Ortlieb, 1998: 69–70. Ramírez & Osorio, 2000: 6 [listed only]. Huber, 2010: 284. Coan & Valentich-Scott, 2012: 437, pl. 144. Uribe et al., 2013: 219. Nielsen, 2013: 56, figs. 11g–11j. Cardoso et al., 2016: 18, fig. 6. Paredes et al., 2016: 139 [listed only]. Valentich-Scott, Coan & Zelaya, 2020: 276, pl. 88.

Chama chilensis Philippi, 1887: 173, pl. 37, fig. 9.

Chama imbricata, – Odhner, 1922: 222. Soot-Ryen, 1959: 40 [listed only]. Osorio & Bahamonde, 1970: 198 [listed only]. Rozbaczylo & Castilla, 1987: 176 [listed only]. (Non Broderip, 1835).

Pseudochama janus, – Bernard, McKinnell & Jamieson, 1991: 36 [listed only]. (Non Chama janus Reeve, 1847).

Type localities: ad Peruviam. (Iquiqui) [=Iquique, Chile], “dredged up attached to stones, Mytili, and turbinated shells, at a depth varying from nine to eleven fathoms, from a bottom of coarse sand, and also found under stones at low water mark” (Chama pellucida). Cahuil, about 1 h far from the sea and 25 to 30 m above sea level” [=Pliocene of Laguna Cahuil, Chile] (Chama chilensis).

Type material: 3 syntypes of Chama pellucida (NHMUK 1950.11.1.63-65, Figs. 20A–20D). Holotype of Chama chilensis (SGO.PI.632: 1 av, Figs. 20L–20N).

Other material studied: Juan Fernández: Alejandro Selkirk: IOC97-48A (FMNH 322322: 6 vs, juvenile); IOC97-50 (FMNH 322319: 7 vs). Robinson Crusoe: IOC97-44 (FMNH 322326: 7 vs; MNHN-CL MOL 101631 ex FMNH 327995: 7 vs); IOC97-44A (FMNH 322323: 4 vs, juvenile; MNHN-CL MOL 101632 ex FMNH 327996: 7 vs, juvenile); IOC97-57A (FMNH 322321: 3 vs, juvenile); IOC97-62 (FMNH 322320: 1 v); IOC97-66A (FMNH 322325: 3 vs); IOC97-68A (FMNH 322324: 11 vs); LACM 1965-97.4 (2 vs); LACM 1965-99.3 (5 avs, 2 vs, with Figs. 20E–20G); LACM 1965-103.3 (4 avs, 1 v, with Figs. 20H–20K); LACM 1966-100.6 (1 v, juvenile).

Other published records: Bahía Padres, Juan Fernández [33°40′45″ S 78°56′45″ W], lower intertidal to 40 fathoms (73 m) (Stempell, 1899). Masatierra [=Robinson Crusoe] [33°38′27″ S 78°52′10″ W], Juan Fernández, 40–100 m (Odhner, 1922: as Chama imbricata). Juan Fernández (Bernard, 1976; Valentich-Scott, Coan & Zelaya, 2020). Playa “El Palillo”, Robinson Crusoe, Juan Fernández (Ramírez & Osorio, 2000).

Distribution: Puerto Pizarro, Tumbes, Perú [5°18′ S] (Cardoso et al., 2016) to Tocopilla, Antofagasta, Chile [22°06′ S], Juan Fernández (Valentich-Scott, Coan & Zelaya, 2020 and herein).

Description: Shell to 76.5 mm H, irregular in outline, usually subcircular, sometimes ovoid, higher than long; inflated, thick, attached to substratum by left valve. Right valve smaller and flatter than left valve. Umbo wide, well outstanding from shell margin. Shell sculptured with comarginal lamellae, with short, flat, blunt projections. Outer sculpture usually eroded in larger specimens. Exterior color white to pink. Inner surface whitish. Margins finely crenulated. Hinge plate wide; with a prominent tubercle in the left valve and the corresponding depression in the right valve.

Comments: Odhner (1922)’s reference to C. imbricata Broderip, from Juan Fernández (Masatierra, 40–100 m) [repeated by Osorio & Bahamonde, 1970] was assigned to C. pellucida by Bernard (1976). Bernard, McKinnell & Jamieson (1991) listed two chamid species from Juan Fernández: Chama pellucida and Chama janus Reeve, 1847 (the latter under Pseudochama). However, only one species was recognized in the material studied herein. The records of C. pellucida from the northern hemisphere (e.g., Adams & Adams, 1853–1858; Carpenter, 1857, Olsson, 1961) actually correspond to C. arcana Bernard, 1976.

Bernard (1976) included Chama chilensis Philippi, 1887 in the synonymy of C. pellucida. However, the specimen that he (Bernard, 1976: fig. 4d) figured as holotype of C. chilensis does not agree with the original figure of this species (Philippi, 1887: pl. 37, fig. 9). A photograph of the holotype is reproduced herein (Figs. 20L–20N). This specimen, as well as the Pliocene specimens from Mejillones (northern Chile) figured by Nielsen (2013: figs. 11g–11j), match the syntypes of C. pellucida.

VENERIDAE

Paphonotia fernandesiana (Stempell, 1899)

Figs. 21A–21H

Figure 21 Family Veneridae.

(A–H) Paphonotia fernandesiana (Stempell, 1899); (A) Syntype of Venerupis fernandesiana Stempell, 1899, ZMB 51989; (B–H) FMNH 322244. (I–K) Timoclea sanfelixensis n. sp.; Holotype, LACM 3830. Scale bars: (A–H, I, J) = 2 mm; (K) = 1 mm.

Venerupis fernandesiana Stempell, 1899: 237, pl. 12, figs. 22, 23.

Venerupis fernandesiana, – Lamy, 1923: 299 [taxonomic note]. Hertlein & Strong, 1948: 193.

Venerupis fernandeziana [sic], – Dall, 1909: 269 [listed only], 292 [synonymy]. Odhner, 1922: 222. Riveros Zuñiga & González Reyes, 1950: 153, fig. 44.

Irus fernandeziana [sic], – Soot-Ryen, 1959: 56 [listed only]. Osorio & Bahamonde, 1970: 202 [listed only]. Rozbaczylo & Castilla, 1987: 176 [listed only].

Irus fernandezianus [sic], – Fischer-Piette & Métivier, 1971: 91 [taxonomic review]. Bernard, 1983: 55 [listed only].

Irusella fernandiziana [sic], – Bernard, McKinnell & Jamieson, 1991: 36 [listed only].

Paphonotia fernandesiana, – Valentich-Scott, Coan & Zelaya, 2020: 325, pl. 104 [“holotype”].

Type locality: Bahía Cumberland, [Robinson Crusoe Island], Juan Fernández [archipelago].

Type material: “Numerous syntypes” (fide Stempell, 1899), although only one of them (the syntype figured herein) is currently preserved (C. Zorn in litt., July 2022; ZMB 51989: 1 v, Fig. 21A).

Other material studied: Juan Fernández: Alejandro Selkirk: IOC97-48A (FMNH 322248: 1 v); IOC97-50 (FMNH 322244: 2 avs, 6 vs, with Figs. 21B–21H; MNHN-CL MOL 101633 ex FMNH 327997: 5 vs); 33°45′S 80°40′48″ W, 79–91 m (USNM 1548480: 1 av). Robinson Crusoe: IOC97-44A (FMNH 322247: 2 vs); IOC97-59 (FMNH 322249: 6 vs, fragment); IOC97-66A (FMNH 322246: 4 vs); IOC97-68A (FMNH 322245: 7 vs); LACM 1965-100.7 (6 vs); LACM 1966-100.4 (>200 avs + vs); Masatierra [=Robinson Crusoe], 20–35 m (SMNH 1228: 4 spec; specimens mentioned by Odhner, 1922).

Distribution: Only known from Juan Fernández archipelago.

Description: Shell to 12.5 mm L, subovate to subquadrate, longer than high, somewhat inflated, inequilateral, moderately solid. Posterior end higher than anterior end. Umbo small, pointed, anteriorly located, prosogyrate. Antero-dorsal margin short, steeply sloping, straight to slightly arched, not distinctly separated from anterior margin. Anterior margin short, rounded. Ventral margin forming a wide curve. Posterior margin obliquely straight to slightly curve, higher than anterior margin. Postero-dorsal margin long, almost straight, sloping slowly; forming a well-marked angulation at the junction with posterior margin. Posterior area of the shell flatter than the other parts. Lunule not defined. Escutcheon raised. Dissoconch sculptured with numerous, low and narrow radial riblets, closely spaced and delicate comarginal threads, and wrinkled, widely separated comarginal lamellae. Shell surface whitish, yellowish, reddish or brownish, sometimes with color bands or blotches. Hinge plate moderately solid, with 3 cardinal teeth in each valve: an elongate anterior tooth, antero-ventrally directed; a markedly triangular median tooth, ventrally directed; an elongate posterior tooth postero-ventrally directed. Median tooth stout in both valves. Anterior cardinal moderately solid and longer than median tooth in the left valve, small and narrow in the right valve. Posterior cardinal narrow, fused to shell margin in the left valve, almost as strong as median cardinal in the right valve. Median cardinal and posterior cardinal of right valve grooved. Nymph short. Inner shell surface whitish, brownish or pinkish. Inner shell margin smooth. Pallial sinus wide, deep, bluntly pointed.

Comments: There is no doubt about the close affinities of Paphonotia fernandesiana with P. elliptica (Sowerby, 1834), the type species of the genus. However, the relationship of these two species with Irus Schmidt, 1818 is less clear. Paphonotia was originally created as a subgenus of Irus, from which it was differentiated by the presence of a crenulated (instead smooth) inner shell margin, ascending pallial sinus, an incised line bounding the lunule (absent in Irus), and more divergent teeth (Hertlein & Strong, 1948: 192–193). The present study reveals that in P. fernandesiana the smooth inner margin may appear crenulated as a consequence of erosion and that the pallial sinus is ascending in some specimens but horizontal in some others. In addition, the comparison of the material of P. fernandesiana here studied with photographs of Irus irus (Linnaeus, 1758) (type species of the genus; figured by Oliver et al., 2016) shows no clear difference in the degree of divergence of hinge teeth between these two species. The only recognizable differences between these taxa are: 1) the three cardinal teeth of I. irus are narrow and similar in width, contrary to the wider median and the right valve posterior cardinal of P. fernandesiana; 2) the anterior cardinal of the right valve appears completely separated from the shell margin in Irus irus, whereas in Paphonotia fernandesiana it is fused; and 3) the posterior cardinal of the left valve appears somewhat more separated from the base of the median cardinal in Irus irus than in Paphonotia fernandesiana. The significance of these differences cannot be resolved in the context of this study and we retain both nominal genera as distinct.

Stempell (1899) based the original description on several syntypes. The label of “holotype” of the specimen figured by Valentich-Scott, Coan & Zelaya (2020: pl. 104) is erroneous, as is the associated repository number given by the authors for this syntype.

Stempell (1899) spelled the species name Venerupis fernandesiana, although the species was described from Juan Fernández. His original spelling is here considered not an inadvertent error (in the sense of International Commission on Zoological Nomenclature [ICZN] (1999), Art. 32.5.1.) but instead Stempell’s choice of latinization (Latin does not properly have a letter “z”). Subsequent citations with “z” by some authors (see synonymy list above) are here interpreted as incorrect subsequent spellings and not emendations in the sense of International Commission on Zoological Nomenclature [ICZN] (1999) Art. 33.2.

Timoclea sanfelixensis n. sp.

Figs. 21I–21K

Type locality: 26°20′ S, 80°03′ W, SE off San Félix Island, Desventuradas archipelago, 415 m (R/V Anton Bruun Cruise 17, station 675H, by Campbell grab, 12 July 1966, LACM 1966-98).

Type material: Holotype (LACM 3830: 1 v, Figs. 21I–21K).

Distribution: Only known from Desventuradas archipelago.

Diagnosis: Shell subovate, higher posterior than anteriorly. Umbo small, pointed, markedly recurved. Dissoconch sculptured with thin radial ribs and comarginal cords.

Description: Shell of 4.0 mm L, subovate, longer than high, inequilateral, solid. Posterior end higher than anterior end. Umbo markedly recurved, low, narrow, pointed, anteriorly located, prosogyrate. Antero-dorsal margin steeply sloping, not distinctly separated from anterior margin. Anterior, ventral and posterior margins forming a continuous curve. Postero-dorsal margin long, almost straight. Lunule well-marked, sculptured with densely placed lamellar. Dissoconch cancellate. Radial sculpture consisting of about 30 radial ribs, separated by interspaces wider than ribs width. Comarginal sculpture consisting of regularly, widely separated cords, narrower than radial elements; originating small granules at the intersection with radial sculpture. Shell surface whitish. Hinge plate solid, with three cardinal teeth in the right valve: a narrow, elongate anterior tooth, anteriorly directed; a stout median tooth, antero-ventrally directed; a solid, elongate posterior tooth postero-ventrally directed. Median and posterior teeth dorsally connected, forming a right angle. Posterior cardinal grooved. Nymph relatively short and wide. Inner shell surface whitish. Inner shell margin finely crenulate. Pallial sinus hardly discernible.

Etymology: Named for the type locality, San Félix Island, Desventuradas; adjective.

Comments: Timoclea sanfelixensis n. sp. is most similar to Eastern Island T. keegani Raines & Huber, 2012, the only species of the genus thus far known from the southeastern Pacific Ocean. However, T. keegani differs from Timoclea sanfelixensis n. sp. by having higher and wider umbos and colorful shells. In addition, T. keegani has secondary radial ribs between the primary ribs (absent in Timoclea sanfelixensis n. sp.), small scales projected in the intersection of radial and comarginal sculpture, and a stronger anterior cardinal tooth than T. sanfelixensis n. sp.

In general shell outline, Timoclea sanfelixensis n. sp. also resembles T. infans (Smith, 1885) and T. scabra (Hanley, 1845), from which it differs by having a more broadly rounded and higher posterior margin, a smaller and more recurved umbo, as well as narrower radial ribs that are separated by wider interspaces.

NEOLEPTONIDAE

Neolepton sanfelixensis n. sp.

Figs. 22A–22I

Figure 22 Family Neoleptonidae.

(A–I) Neolepton sanfelixensis n. sp., (A, B) Holotype, MNHN-CL MOL 101634; (C, D) Paratype, MNHN-CL MOL 101635; (E, F) (SEM) Paratype, FMNH 322301; (G) (SEM): Paratype, LACM 3831; (H–I) (SEM) FMNH 322302. (J–L) Neolepton sp. A, LACM 1966-98.6. Scale bars: (A–F, I, L) = 1 mm; (G, J, K) = 2 mm; (H) = 500 µm.

Type locality: 26°17′24.14″ S, 80°6′36.22″ W, San Félix Island, Desventuradas, 12.2 m (collected by scuba by Rüdiger Bieler, 26 February 1997; IOC97-30A).

Type material: Holotype (MNHN-CL MOL 101634 ex FMNH 327998: 1 v, Figs. 22A, 22B) and 14 paratypes from the type locality (MNHN-CL MOL 101635 ex FMNH 327999: 5 vs, with Figs. 22C, 22D; FMNH 322301: 7 vs, with Figs. 22E, 22F; LACM 3831 ex FMNH 328000: 2 vs, with Fig. 22G).

Other material studied: Desventuradas: San Félix: IOC97-26 (FMNH 322314: 18 vs, 2 fragments); IOC97-29 (FMNH 322302: 2 avs subadult, 19 vs, with Figs. 22H, 22I); IOC97-32 (FMNH 322306: 7 vs). San Ambrosio: IOC97-12 (FMNH 322303: 1 av); IOC97-13 (FMNH 322305: 5 vs); IOC97-18 (FMNH 322304: 2 vs).

Distribution: Only known from Desventuradas archipelago.

Diagnosis: Shell ovate, with the anterior end bluntly pointed. Umbos low. Dissoconch sculptured with regularly distributed comarginal cords. Hinge plate completely supporting cardinal teeth. Shell surface usually stained brown in the posterior area.

Description: Shell to 3.7 mm L, ovate, longer than high, not inflated, somewhat inequilateral, solid. Anterior end bluntly pointed, posterior end evenly rounded, higher than anterior end. Umbo small, low, subcentral. Antero-dorsal and postero-dorsal margins of about the same length, sloping at similar angle, the anterior one almost straight, the posterior one markedly convex. Anterior, ventral, and posterior margins forming a wide, even curve. Prodissoconch of 975 µm in diameter. Dissoconch sculptured with low, coarse, closely spaced, regularly distributed comarginal cords. Shell surface shiny, whitish, usually widely stained in brown at the posterior area, sometimes with small brownish stains behind the umbo. Hinge plate relatively solid. Left valve with two solid cardinal teeth behind the resilifer; cardinal dorsally in contact, forming a hook. In addition, an elongate posterior lateral tooth. Posterior cardinal one-third the length of the anterior one. Right valve: with three cardinal teeth behind the resilifer, and two posterior lateral teeth. Median cardinal strong, sharply triangular, with relatively short base and posteriorly displaced cusp. Anterior and posterior cardinals short, narrowly elongate, located at right angle, the posterior about a half the length of the anterior one. Posterior laterals elongate, massive. Resilifer moderate in size, strout, supporting a strong internal ligament. External ligament narrow, amphidetic, with the posterior part longer than the anterior one. Inner shell surface reflecting outer shell sculpture; withish or stained in brown in externally colored specimens. Inner margin smooth.

Etymology: Named for the type locality, San Félix Island, Desventuradas; adjective.

Comments: In general shell outline, Neolepton sanfelixensis n. sp. closely resembles Magellanic N. hupei Soot-Ryen, 1957 (figured and described in Zelaya & Ituarte, 2004). However, N. sanfelixensis n. sp. has coarser and evenly distributed comarginal sculpture (cords), whereas N. hupei only has irregularly spaced comarginal lines. Another difference lies in the width of the hinge plate: in N. sanfelixensis n. sp., the hinge is evenly wide and completely supports the cardinal teeth. In N. hupei, the hinge plate is narrowed below the umbo, producing an overhanging of the distal part of cardinal teeth. In addition, the posterior area of the shell is usually stained brown in N. sanfelixensis n. sp., whereas the shell of N. hupei is uniformly white. A similarly speckled posterior part of the shell appears in Neolepton chaneyi Coan & Valentich-Scott, 2012, but that species has the antero-dorsal and postero-dorsal margins sloping more steeply, resulting in a markedly triangular shell outline. That species also has wider umbos than N. sanfelixensis n. sp.

Neolepton sp. A

Figs. 22J–22L

Material studied: 26°20′ S, 80°03′ W, SE off San Félix Island, Desventuradas archipelago, 415 m (R/V Anton Bruun Cruise 17, station 675H, by Campbell grab, 12 July 1966, LACM 1966-98.6 (1 v, Figs. 22J–22L).

Distribution: Only known from Desventuradas archipelago.

Description: Shell of 3.8 mm L, triangular, longer than high, not inflated, somewhat inequilateral, thin. Anterior end somewhat projected, posterior end higher than anterior end. Umbo small, low, subcentral. Antero-dorsal and postero-dorsal margins of about the same length, the anterior one almost straight, the posterior slightly convex. Anterior and ventral margins forming an even curve. Dissoconch sculptured with fine growth lines. Shell surface shiny, whitish. Hinge plate solid. Right valve: with three cardinal teeth behind the resilifer, and two posterior lateral teeth. Median cardinal massive, sharply triangular, with short base and posteriorly displaced cusp. Anterior and posterior cardinals short, narrowly elongate, located at right angle, the posterior weaker and shorter than the anterior one. Posterior laterals elongate, massive. Resilifer triangular, deep. Inner shell surface withish. Inner margin smooth.

Comments: In general shell outline, Neolepton sp. A resembles Magellanic Neolepton amatoi Zelaya & Ituarte, 2004, from which it differs by having a less projected anterior end that leads to a more subcentrally located umbo. In addition, the right valve median cardinal tooth has a shorter base in Neolepton sp. A. It appears to be a new, undescribed, species. However, the limited material (a single valve) and its poor state of preservation preclude us from naming it herein.

XYLOPHAGAIDAE

Xylophaga sp. A

Fig. 23

Figure 23 Family Xylophagaidae.

(A–D) Xylophaga sp. A, LACM 1965-101.8. Scale bar: (A–D) = 2 mm.

Xylophaga globosa, – Rozbaczylo & Castilla, 1987: 176 [listed only]. Bernard, McKinnell & Jamieson, 1991: 36 [listed only]. (Non Sowerby, 1835).

Material studied: Juan Fernández; Robinson Crusoe: LACM 1965-101.8 (15 avs with dried tissue, with Figs. 23A–23D).

Description: Shell of 4.1 mm L, ovate, higher than long, flatter posteriorly than anteriorly, equivalve, thin. Umbo broad and low. Umbonal-ventral sulcus wide and deep; flanked by two low but strong radial ribs, originating ventral sinuation. Posterior rib more ventrally projected than anterior rib. Postumbonal area of shell subcircular in outline, sculptured with low growth lines. Anterior area of shell shorter than posterior area, inflate, sculptured with few, widely spaced comarginal ridges. Anterior incision relatively small. Umbonal reflection small. Prodissoconch orange. Dissoconch surface whitish. Hinge plate edentate. Umbonal-ventral ridge gradually increasing in width, with semicircular condyle at ventral end.

Comments: Stempell (1899) reported Xylophaga dorsalis (Turton, 1819) from the Juan Fernández archipelago. This is a European species (Turner, 1955; Romano et al., 2014), morphologically similar to X. globosa (Turner, 1955). Dall (1909: 292) considered Stempell’s record as belonging to X. globosa, an opinion followed by Valentich-Scott, Coan & Zelaya (2020). No additional records of Xylophaga species appear in the literature from this archipelago, although X. globosa was listed from Juan Fernández in several checklists (Osorio & Bahamonde, 1970; Rozbaczylo & Castilla, 1987; Bernard, McKinnell & Jamieson, 1991). The specimen studied herein has a shorter and more evenly rounded posterior end of the shell, and the anterior area sculptured with fewer and more widely separated ridges than the specimens of X. globosa figured by Sowerby (1849: pl. 108, figs. 101–102) [type material] and Turner (1955: pl. 89). Recently, Marcel Velásquez (pers. com., December 2021; ongoing research) studied freshly collected material from Juan Fernández, resulting in the recognition of two Xylophaga species: one of them morphologically similar to X. globosa (but possibly new to science) and another indubitably new. The specimens identified as X. dorsalis by Stempell (1899) could not be located at the ZMB (C. Zorn in litt., July 2022), where other specimens reported by this author are housed.

TEREDINIDAE

Bankia martensi (Stempell, 1899)

Fig. 24

Figure 24 Family Teredinidae.

(A, B) Bankia martensi (Stempell, 1899), LACM 1966-100.8. Scale bar: (A, B) = 1 mm.

Teredo (Xylotrya) martensi Stempell, 1899: 240–242, pl. 12, figs. 24–27.

Xylotrya martensi, – Dall, 1909: 278 [listed only].

Bankia (Bankia) chiloensis Bartsch, 1923: 147–149.

Bankia odhneri Roch, 1931: 20–21, pl. 4, fig. 10. Carcelles & Williamson, 1951: 348 [listed only].

Bankia valparaisensis Moll in Roch & Moll, 1935: 273, pl. 2, fig. 3.

Bankia argentinica Moll in Roch & Moll, 1935: 274, pl. 2, fig. 5.

Bankia chiloensis, – Carcelles & Williamson, 1951: 348. [listed only].

Bankia martensi, – Turner, 1966: 88, 93, 109, 114, 128, pl. 61 A-D. Campos & Ramorino, 1990: 19–20, 23–25, pl. 1, figs. 1–4, pl. 3, figs. 1–4, pl. 5, figs. 1–6, pl. 6, figs. 1–6. Bernard, McKinnell & Jamieson, 1991: 36 [listed only]. Sporman, López & González, 2006: 105–109. Zelaya, 2009: 447. Velásquez, Gallardo Silva & Lira, 2011: 33–36; 2014: 211–220. Valentich-Scott, Coan & Zelaya, 2020: 376, pl. 121.

Bankia (Bankia) martensi, – Soot-Ryen, 1959: 70. Stuardo, Saelzer & Rosende, 1970: 153–166. Osorio & Bahamonde, 1970: 207 [listed only]. Osorio, Atria Cifuentes & Mann Fischer, 1979: 36, fig. 44. Bernard, 1983: 62 [listed only]. Reid & Osorio Ruiz, 2000: 140, figs. 5N–5P. Osorio, 2002: 166–167.

Type localities: Punta Arenas, Chile (Teredo (Xylotrya) martensi). Chiloé Island (Bankia (Bankia) chiloensis). Valparaiso, Chile (Bankia valparaisensis). Buenos Aires, [Argentina] (Bankia argentinica). Port Williams, [Malvinas]/Falklands Inseln (Bankia odhneri).

Type material: Types of Bankia martensi originally deposited in ZMB, but not currently found there (Turner, 1966; C Zorn, October 2023, personal communication). Holotype of Bankia (Bankia) chiloensis (USNM 348498). Holotype of Bankia odhneri (SMNH 5094). Holotype of Bankia argentinica (ZMB 108920). Holotype of Bankia valparaisensis (ZMB 108923).

Material studied: Juan Fernández: Robinson Crusoe: LACM 1966-100.8 (1 v, Figs. 24A, 24B).

Distribution: Valparaiso (33° S), Chile to Tierra del Fuego (54° S) (Turner, 1966), and Juan Fernández (herein), extending in the Atlantic to Malvinas / Falkland Islands (51° S) (Roch, 1931) and “Buenos Aires” Province, Argentina (Moll in Roch & Moll, 1935). Nair (1975) identified as Bankia martensi specimens from Gulf of Cariaco, Venezuela. However, Velásquez, Valentich-Scott & Capelo (2017) considered this record as a species-level misidentification.

Description: Shell of 3 mm H, higher than long, equilateral, thin; composed of two clearly discernible parts: an ovate, globose anterior part and an auricular and flat posterior part (“posterior slope”). Umbo low and wide. A narrow, relatively shallow sulcus running from the umbo to the ventral margin, dividing the anterior part of the shell in two areas: an anterior slope and a posterior disc. Ventral margin of anterior slope forming a deep, right angle indentation; ventral margin of disc evenly rounded. Anterior slope sculptured with numerous, serrated comarginal ridges; disc and auricle only sculptured with weak growth lines. Shell surface whitish. Hinge plate edentate. Umbonal area projected forming an apophysis. A rounded ventral condyle present.

Comments: Two teredinid species were mentioned previously from Juan Férnández: Lyrodus pedicellatus (Quatrefages, 1849) and Bankia martensi (Stempell, 1899). The former was reported by Stuardo, Saelzer & Rosende (1970), who however did not detail the collection site, the number of specimens found, or the repository of the studied material. The only other record of L. pedicellatus in Chilean waters stems from Coquimbo (Valentich-Scott, Coan & Zelaya, 2020). By contrast, Bankia martensi is a common species in Chilean waters. The species was reported from Juan Fernández by Bernard, McKinnell & Jamieson (1991) based on the same Anton Bruun specimens studied herein. This material consists of empty valves, which indeed are morphologically indistinguishable from those of B. martensi. We tentatively assign this material to this species, although the pallet morphology, crucial for confirming species identity (Turner, 1966), remains unknown.

CUSPIDARIIDAE

Cuspidaria fernandezensis n. sp.

Figs. 25A–25H

Figure 25 Family Cuspidariidae.

(A–H) Cuspidaria fernandezensis n. sp.; (A–D) Holotype, USNM 898738; Paratypes, (E–H) LACM 3832. (I–P) Cuspidaria sanfelixensis n. sp.; (I, J) Holotype, LACM 3833; (K–N) Paratypes, LACM 3834; (O, P) Paratype, FMNH 312476. Scale bar (A–P) = 2 mm.

Cuspidaria cf. patagonica, – Bernard, McKinnell & Jamieson, 1991: 36 [listed only]. (Non Smith, 1885).

Type locality: 33°45′00″ S, 80°40′48″ W, off Alejandro Selkirk Island, Juan Fernández archipelago, 79–91 m (R/V Eltanin Cruise 21, station 203, by Blake trawl, 26 November 1965).

Type material: Holotype (USNM 898738: 1 av, Figs. 25A–25D) and four paratypes from 65-98 (LACM 3832: 3 vs, with Figs. 25E–25H; FMNH 312475 ex LACM 3832: 1 av, damaged).

Distribution: Only known from Juan Fernández archipelago.

Diagnosis: Disc ovate-elongate, sculptured with irregular comarginal striae. Rostrum long, dorsally recurved, widely connected with the disc.

Description: Shell up to 11 mm L, ovate-elongate, low, somewhat inflated, markedly inequilateral, moderately solid. Anterior end wide, roundly projected; posterior end narrowly projected into rostrum. Umbo extremely low, only slightly outstanding from shell margin; somewhat anteriorly displaced, opisthogyrate. Antero-dorsal, anterior and anterior half of ventral margins forming a round arch; posterior half of ventral margin sinuous. Postero-dorsal margin longer than antero-dorsal margin; sloping, evenly straight in smaller specimens, distally recurved in larger specimens. Rostrum long, massive, widely connected with disk. Transition between disk and rostrum moderately demarcated by a depression of shell surface. Rostral ridge strong. Dissoconch sculptured with irregular comarginal striae. Shell surface white, dull. Periostracum thin, yellowish. Hinge: right valve with a narrow, elongated posterior lateral tooth with subcentral cusp; forming a depression with the dorsal margin, where the dorsal margin of the opposite valve fits. Left valve with no distinct teeth, just a postero-dorsal thickened margin. Resilifer large, elongate-ovate, posterior to umbo, postero-ventrally directed. Inner shell surface whitish. Inner margin smooth.

Etymology: Named for the type locality in the Juan Fernández archipelago; adjective.

Comments: Cuspidaria fernandezensis n. sp. is most similar to Cuspidaria hawaiensis Dall, Bartsch & Rehder, 1938, from which it differs by having a shorter and wider rostrum, and a smaller and less recurved umbo. In general shell outline, Cuspidaria fernandezensis n. sp. also resembles the material referred to by Raines & Huber (2012) as “Myonera sp.”. However, the latter strikingly differs by having comarginal lamellae. Among the Eastern Pacific Cuspidaria species, Cuspidaria fernandezensis n. sp. is also similar to C. parapodema Bernard, 1969 and C. parkeri Knudsen, 1970. Cuspidaria fernandezensis n. sp. differs from these species by having a distally concave postero-dorsal shell margin, which results in a dorsally recurved rostrum. By comparison, the postero-dorsal margin of C. parapodema is completely straight, resulting in an anteriorly directed rostrum. Cuspidaria fernandezensis n. sp. also resembles the specimen from southern Chile identified by Cárdenas, Aldea & Valdovinos (2008: fig. 7.104) as Cuspidaria cf. infelix, although the latter has a more steeply sloping antero-dorsal shell margin and less projected anterior end and rostrum.

Cuspidaria sanfelixensis n. sp.

Figs. 25I–25P

Type locality: 26°20′ S, 80°03′ W, SE off San Félix Island, Desventuradas archipelago, 415 m (R/V Anton Bruun Cruise 17, station 675H, by Campbell grab, 12 July 1966, LACM 1966-98).

Type material: Holotype (LACM 3833: 1 v, Figs. 25I, 25J) and 3 paratypes from the type locality (LACM 3834: 2 vs, Figs. 25K–25N [plus 7 fragments]; FMNH 312476 ex LACM 3834: 1 v, Figs. 25O, 25P).

Distribution: Only known from Desventuradas archipelago.

Diagnosis: Antero-dorsal margin straight, steeply sloping. Rostrum wide, short to medium in size. Right valve with short posterior lateral tooth.

Description: Shell to 9.8 mm L, subovate, high, inflated, markedly inequilateral, moderately solid. Anterior end wide and rounded; posterior end projected into rostrum. Umbo prominent, high and wide, slightly anteriorly displaced, markedly opisthogyrate. Antero-dorsal margin straight, steeply sloping. Anterior and anterior half of ventral margins forming a wide, continuous curve; posterior half of ventral margin sinuous. Postero-dorsal margin concave, sloping at a similar angle than antero-dorsal margin. Rostrum short to medium in size, massive, widely connected with disk. Transition between disk and rostrum moderately demarcated by a depression of shell surface. Rostral ridge weak. Dissoconch only sculptured with low growth lines. Shell surface whitish, shiny, translucent. Hinge: right valve with short but high posterior lateral tooth, with subcentral cusp; forming a depression with the dorsal margin, where the dorsal margin of the opposite valve fits. Left valve with a postero-dorsal thickened margin. Resilifer small, posterior to umbo. Inner shell surface whitish. Inner margin smooth.

Etymology: Named for the type locality, San Félix Island, Desventuradas; adjective.

Comments: Among Eastern Pacific Cuspidaria species, Cuspidaria sanfelixensis n. sp. is most similar to Cuspidaria parapodema Bernard, 1969, from which it differs by having a more concave postero-dorsal margin, a shorter and wider rostrum, and a smaller posterior lateral tooth in the right valve. The markedly triangular disc of Cuspidaria sanfelixensis n. sp. resembles that of Luzonia chilensis (Dall, 1890), a name applied by Bernard, McKinnell & Jamieson (1991) to the Juan Fernández specimens studied herein. However, Cuspidaria sanfelixensis n. sp. clearly differs from that species by having a narrower and more markedly set-off and projected rostrum. In general shell outline, Cuspidaria sanfelixensis n. sp. also resembles Plectodon scaber Carpenter, 1884, although the latter bears elongate anterior and posterior teeth in the right valve, and has a granulate shell surface (both characters considered as distinctive for that genus by Coan & Valentich-Scott, 2012).

Cuspidaria sanfelixensis n. sp. strikingly differs from C. fernandezensis n. sp. by having a higher and shorter shell outline, with a straight (instead of widely curved) antero-dorsal margin, a shorter rostrum, and a weaker rostral ridge. Moreover, the dissoconch of C. sanfelixensis is sculptured only by low growth lines whereas irregular comarginal striae appear in C. fernandezensis. Furthermore, the right valve of C. fernandezensis bears a higher posterior lateral tooth than C. sanfelixensis.

HALONYMPHIDAE

Halonympha recurvirostris n. sp.

Fig. 26

Figure 26 Family Halonymphidae.

(A, B) Halonympha recurvirostris n. sp., holotype, LACM 3836. Scale bar: (A, B) = 2 mm.

Type locality: 26°20′ S, 80°03′ W, SE off San Félix Island, Desventuradas archipelago, 415 m (R/V Anton Bruun Cruise 17, station 675H, by Campbell grab, 12 July 1966, LACM 1966-98).

Type material: Holotype (LACM 3836: 1 v, Figs. 26A, 26B).

Distribution: Only known from Desventuradas archipelago.

Diagnosis: Disc ovate, widely rounded anteriorly. Posteriorly projected in long, dorsally arched rostrum. Shell surface with dense comarginal ribs. Internal surface with a wide posterior shelf.

Description: Shell to 7.5 mm L, ovate, high, inflated, markedly inequilateral, moderately solid. Anterior end widely rounded; posterior end narrow, projected into rostrum. Umbo broad but low, slightly posteriorly displaced, markedly opisthogyrate. Antero-dorsal, anterior and antero-ventral margins forming a continuous curve; posterior half of ventral margin sinuous. Postero-dorsal margin markedly concave. Rostrum short and wide. Transition between disk and rostrum moderately demarcated by a depression of shell surface. Dissoconch sculptured with numerous, low, densely packed comarginal cords. Shell surface whitish. Right valve edentulous. Chondrophore narrowly triangular, elongate, oblique. Inner shell surface whitish, with an elongate, oblique posterior ridge (“clavicular rib”), delimiting a wide shelf. Inner margin smooth.

Etymology: The name of the species refers to the markedly curved rostrum (used as Latin adjective; curved-beaked).

Comments: The new species fits within the current concept of Halonympha (as defined for instance by Allen & Morgan, 1981; Hayami & Kase, 1993; Poutiers & Bernard, 1995), Within this genus, Halonympha recurvirostris n. sp. closely resembles Halonympha claviculata (Dall, 1881) and Halonympha inflata Jeffreys, 1882, from which it differs by having a longer rostrum, a narrower umbo, more markedly arched postero-dorsal margins, and a wider posterior shelf.

Previously, the genus Halonympha was reported from the Atlantic, western Pacific, and Indian Oceans (Poutiers & Bernard, 1995). This is the first record from the eastern Pacific.

PANDORIDAE

Pandora pyxis n. sp.

Fig. 27

Figure 27 Family Pandoridae.

(A–D) Pandora pyxis n. sp., holotype, LACM 3837. Scale bar: (A–D) = 2 mm.

Pandora cistula, – Bernard, McKinnell & Jamieson, 1991: 36 [listed only]. (Non Gould, 1850).

Type locality: 33°34–41′ S, 78°45–55′ W, off W side of Robinson Crusoe Island, Juan Fernández archipelago, 130–180 m (R/V Anton Bruun Cruise 12, trawled, 13–15 December 1965, LACM 65-101).

Type material: Holotype (LACM 3837: 1 av, Figs. 27A–27D).

Distribution: Only known from Juan Fernández archipelago.

Diagnosis: Shell elongate, low, with vertical posterior margin. Posterior area of shell wide, not projected into rostrum.

Description: Shell of 6.3 mm L, elongate-cuneiform, longer than high, relatively low, markedly inequilateral, delicate. Right valve flat; left valve somewhat inflated, with a shallow sulcus, running from the umbo to the ventral margin, delimiting a projected anterior area of shell. Anterior end pointed, posterior end evenly flattened, higher than anterior end. Umbo small, low, anteriorly displaced. Antero-dorsal margin short, sloping nearly straight, forming a prominent angulation at the junction with anterior margin; postero-dorsal margins long, nearly horizontal. Anterior and ventral margins evenly arched, only slightly indented by radial sulcus. Posterior margin straight, vertical. In the left valve, a solid radial rib running from the umbo to the junction of ventral and posterior margins delimits a wide, flat, posterior area. In the right valve, an obscure ridge running from the umbo to the dorsal fourth part of the posterior margin. Dissoconch sculptured with irregularly growth folds, and faint radial grooves only in right valve. Shell surface whitish. Right valve with two straight, posteroventrally directed crura diverging from umbo. Posterior crus long and narrow; anterior crus short and stout. In addition, with a prominent thickening bordering dorsally the anterior muscle scar. Left valve with a short, straight and solid anterior crus, posteroventrally directed.

Etymology: The Latin noun pyxis (small box), a play on the phrase “Pandora’s Box”, based on an artifact in Greek mythology.

Comments: The material studied herein was previously identified as Pandora cistula by Bernard, McKinnell & Jamieson (1991). This name was used by several authors (e.g., Osorio & Reid, 2004; Cárdenas, Aldea & Valdovinos, 2008) to refer to Magellanic pandorids. However, Güller & Zelaya (2016) demonstrated that P. cistula remains at present only known from its type material, and that other records previously attributed to that species actually correspond to Pandora braziliensis Sowerby, 1874. However, this is not the case of the Juan Fernández material, which clearly differs from P. braziliensis by having a considerably lower and more elongated shell outline, a vertical posterior margin, a wider posterior area of shell, and narrower crura. The above-mentioned characters resulted in the distinction of the Juan Fernández specimen from the Peruvian specimen reported by Valentich-Scott, Coan & Zelaya (2020) as Pandora cf. braziliensis. Pandora pyxis n. sp. resembles the southwestern Atlantic Pandora brevirostris Güller & Zelaya (2016), from which it differs by having an even lower shell outline and a wider posterior area of shell, which in addition is not projected into a rostrum as in that species.

LYONSIIDAE

Entodesma sp.

Fig. 28

Figure 28 Family Lyonsiidae.

(A, B) Entodesma sp., LACM 1966-98.7. Scale bar: (A, B) = 2 mm.

Entodesma cuneatum, – Bernard, McKinnell & Jamieson, 1991: 36 [listed only]. (Non Gray, 1828–1830).

Material examined: Desventuradas: San Félix: LACM 1966-98.7 (1 v, Figs. 28A, 28B).

Description: Shell of 5.8 mm L, ovate, longer than high, inflated, markedly inequilateral, solid. Posterior end much higher than anterior end. Umbo broad, low, anteriorly displaced. Antero-dorsal margin short, sloping straight. Ventral margin truncated at anterior half; posterior half of ventral margin forming a continuous curve with posterior margin. Postero-dorsal margins longer than antero-dorsal margin. Dissoconch sculptured with irregular comarginal lamellae. Shell surface whitish. Inner margin smooth.

Comments: The Juan Fernández material studied herein consists in a single, eroded, partially broken and small valve, which closely resembles the specimen figured by Prezant (1981: fig. 10) as Entodesma chilensis Philippi, 1845, the latter an objective synonym of E. cuneatum (see Valentich-Scott, Coan & Zelaya, 2020). However, the scarce material available from Juan Fernández, and its poor state of preservation, preclude us from confirming conspecificity.

PARALIMYIDAE

Panacca chilensis Coan, 2000

Fig. 29

Figure 29 Family Paralimyidae.

(A, B) Panacca chilensis Coan, 2000, holotype, LACM 2876. Scale bar: (A, B) = 1 cm.

Pholadomya cf. darwini, – Bernard, McKinnell & Jamieson, 1991: 36 [listed only]. (Non Dall & Ochsner, 1928).

Panacca chilensis Coan, 2000: 165, figs. 1, 2.

Panacca chilensis, – Valentich-Scott, Coan & Zelaya, 2020: 406, pl. 131 [holotype].

Type locality: 33°34–41′ S, 78°45–55′ W, [off Robinson Crusoe Island], Juan Fernández archipelago, 130–180 m.

Type material: Holotype (LACM 2876: 1 v, figs. 29A, 29B).

Distribution: Only known from Juan Fernández archipelago.

Description: Shell of 21 mm L, triangular, longer than high, markedly inequilateral, thin. Anterior end short, posterior end projected. Umbo wide, anteriorly located. Antero-dorsal and postero-dorsal margins sloping at similar angle, the latter much longer than the former. Anterior and ventral margins forming a continuous wide curve. Dissoconch sculptured with granulate, comarginal striae and 11 strong radial rib on central area; radial sculpture separated by wide interspaces. Shell surface dull, whitish. Hinge edentulous. Ligament external, seated on a weak nymph. Inner shell surface reflecting outer shell sculpture. Inner margins crenulated by radial sculpture.

Comments: The species is currently known only from the original description, which was based on a single right valve (Coan, 2000). The holotype was refigured by Valentich-Scott, Coan & Zelaya (2020).

Diversity and distribution of species

Resulting from this study, a total of 48 taxa are recognized from the Juan Fernández and Desventuradas archipelagos. Three of them (Bathyarca corpulenta, Entodesma sp. and Amygdalum sp.) could not be determined with certainty. Of the 45 remaining taxa, only 18 species had been previously reported from the area. Nineteen species proved to be new to science (and are described herein), and six species are probably new, but not formally named herein due to limited material (Fig. 30). Most of the taxa recognized in this study correspond to endemic species from Juan Fernández (16 species), Desventuradas (13 species), or shared between these two areas and not present anywhere else (six species). Ten species are shared with the Perú-Chile Province (Fig. 30). Two additional species were reported as regionally occurring but lack concrete records.

Figure 30 Diversity and distribution of species.

Distribution and status of the species studied from desventuradas (DES) and juan fernández (JF) archipelagos. Number of species is indicated between brackets. The two species with unconfirmed literature records are excluded from this graph.

Discussion

Although the Juan Fernández and Desventuradas archipelagos have been globally recognized as important biodiverse ecosystems (Grandi-Nagashiro, González & Fernández, 2010; Pompa, Ehrlich & Ceballos, 2011; Friedlander et al., 2016; Tapia-Guerra et al., 2021), the knowledge of their marine invertebrate fauna remains poor, and bivalves are no exception. The remoteness and the difficulty of accessing these islands have undoubtedly limited the historical study of these areas. In fact, most of the bivalve species reported from Juan Fernández come from occasional findings, in contributions usually covering single or few species each. Bernard, McKinnell & Jamieson (1991) reported 31 preliminarily identified species from this archipelago. However, the published information was limited to a species listing, without descriptions or illustrations, and including provisional names. The knowledge of the Desventuradas bivalve fauna is even more limited: the only available information comes from the specimens collected by the Anton Bruun cruise 17, on the basis of which Bernard, McKinnell & Jamieson (1991) recognized 13 species (with the same limitations of the species list as that from Juan Fernández), and two additional species listed by Tapia-Guerra et al. (2021). The present study is the most comprehensive contribution to the Juan Fernández and Desventuradas bivalve fauna available to date. For the Desventuradas, the number of recorded species nearly doubled the previously known. In the case of the Juan Fernández archipelago, the raw number of recorded nominal species recognized herein almost matches the number compiled by Bernard, McKinnell & Jamieson (1991) (30 and 31 species, respectively), although only one third of the species listed by those authors are here recognized as valid for this archipelago. In fact, we found that 21 of the names applied by Bernard, McKinnell & Jamieson (1991) were based on misidentifications. Furthermore, three nominal species reported by these authors from Juan Fernández (as “Nucula grayi”, “Cuspidaria chilensis”, and “Cuspidaria n. sp.”) actually originated in the Desventuradas, whereas another species (“Limaria n. sp.”) reported from Desventuradas came from Juan Fernández. Most of the new findings arising from this study are species new to science, with 19 species obtained from the intertidal to 415 m depth. In addition, six other – and probably new – species are recognized: two from Juan Fernández (Gari sp. B; Xylophaga sp. A), the other four from Desventuradas (Propeamussium/Parvamussium sp. A; Neolepton sp. A; Monia sp. B; Condylocardiidae sp. A). Most of the new species described in this study are micro-mollusks, with 79% of the species having maximum shell sizes smaller than 10 mm, of which 60% are smaller than 5 mm. The high number of previously unrecognized species with small body size clearly point to the scarce attention that this fraction of regional mollusks has received in the past. Several of the previously undescribed species found in this study correspond to families that are known to be highly diverse in deeper waters (e.g., Cuspidariidae, Nuculidae, Propeamussiidae) and were obtained at a single station sampled by the R/V Anton Bruun at 415 m off Desventuradas. Additional deeper-water sampling will undoubtedly lead to a substantive increase of bivalve species richness in this region.

In addition to Juan Fernández and Desventuradas, two other oceanic archipelagos are positioned in the southeastern Pacific off Chile: Easter Island and Salas & Gómez. These archipelagos are located at about the same latitude as Desventuradas, but nearly 2,900 km westward. The bivalves from these archipelagos were studied by Raines & Huber (2012). Salas & Gómez (with 24 species), shows a species richness similar to that recognized in the present study for Juan Fernández (31 species) and Desventuradas (25 species). However, Easter Island has a considerably higher reported diversity (69 species). This fact is most probably based on the more intensive historic sampling efforts in that archipelago compared to the still unsatisfactory sampling effort in the other areas, as evidenced by the species accumulation curve provided by Tapia-Guerra et al. (2021) for Desventuradas. There is a clear bias in the distribution of sampling sites both at Desventuradas and Juan Fernández archipelagos, with most concentrated on the northern side of the islands and in sheltered bays (see Fig. 1), with a predominance of shallow-water collections. A similar difference in bivalve richness is evident when comparing Juan Fernández and Desventuradas with that of the Chilean continental coast. A compilation (data from Güller & Zelaya, 2015; Valentich-Scott, Coan & Zelaya, 2020) accounts for a total of about 50 and 60 bivalve species at the Chilean coast, respectively, at the corresponding latitudes of Desventuradas (26°S) and Juan Fernández (33.5°S).

Faunistic affinities of Juan Fernández and Desventuradas archipelagos

Several recent studies highlighted the great affinities between Juan Fernández and Desventuradas marine biota (Meneses & Hoffmann, 1994; Dyer & Westneat, 2010; Pequeño & Lamilla, 2000; Pequeño & Sáez, 2000; Moyano, 2005; Silva & Chacana, 2005; Retamal & Moyano, 2010; Rodríguez-Ruiz et al., 2014). However, depending on the target taxa, different affinities of these archipelagos with other geographic areas were proposed. Based on algae, sessile and mobile invertebrates, and fishes, sampled between 10 and 20 m depth, Friedlander et al. (2016) reported for Juan Fernández and Desventuradas the presence of a mixture of tropical, subtropical, and temperate species, with strong affinities to the Indo-west Pacific. Such affinities were also mentioned by other authors, when considering bryozoans (Moyano, 2005), fishes (Springer, 1982; Parin, 1991; Pequeño & Lamilla, 2000; Pequeño & Sáez, 2000, 2004; Dyer & Westneat, 2010), and algae (Meneses & Hoffmann, 1994), a fact that led to considering Juan Fernández and Desventuradas (or at least the latter) as the easternmost extension of the Indo Pacific Region (e.g., Pequeño & Lamilla, 2000; Pequeño & Sáez, 2000, 2004; Moyano, 2005; Dyer & Westneat, 2010). Within this framework, an Indo-west Pacific faunal origin, followed by subsequent dispersal and arrival event(s) to Desventuradas and Juan Fernández, was proposed (Briggs, 1999; Dyer & Westneat, 2010). Santelices & Meneses (2000) found in Juan Fernández and Desventuradas a group of algal species shared with several sub-Antarctic islands and the southern tip of South America (i.e., the Magellan Province). The occurrence of such cold-water species in Juan Fernández and Desventuradas is not surprising considering that these archipelagos lie at a confluence of subtropical and sub-Antarctic waters (Silva, 1985; Meneses & Hoffmann, 1994; Parin, Mironov & Nesis, 1997; National Geographic Society, 2013). On the other hand, Andrade (1985) and Bernard, McKinnell & Jamieson (1991) reported (for decapods and bivalves, respectively) high similarities of Juan Fernández/ Desventuradas with the adjacent continental coast of Perú and Chile, which led Bernard, McKinnell & Jamieson (1991) to consider these archipelagos as part of the Perú-Chile Province.

Most of the species shared between Juan Fernández and Desventuradas archipelagos are either byssally attached (mytilids and arcids) or nestling species (Lasaea). Likewise, most of the species shared between Juan Fernández/Desventuradas and the Perú-Chile Province are byssally attached (mytilids and arcids) or cementing (chamids). Thus, a faunal exchange might be explained by natural means (e.g., drifting wood) and/or through human mediation (e.g., fishing vessels). In this context, it is noteworthy that the cementing species Chama pellucida has reached Juan Fernández, which has seen long-distance vessel traffic for centuries, but not the traffic-isolated Desventuradas archipelago, which has no harbor.

Conclusions

The present study comes to different conclusions, for bivalves, than the prior studies cited above: We found low similarity between the Juan Fernández and Desventuradas bivalve faunas: of the 48 species recognized in this study, only nine species (19%) are present in both archipelagos. Most of this faunal overlap might be explained by natural or human-mediated rafting.

Juan Fernández and Desventuradas have no bivalve species in common with the Indo-west Pacific. The only exception is “Bathyarca corpulenta”, a taxon excluded from our biogeographic analysis because conspecificity of the specimens reported from Juan Fernández with those from the Indo-west Pacific is unresolved (see comments under that species). Indeed, the results arising from our study agree with the biogeographic scheme proposed by Parin, Mironov & Nesis (1997), as part of which Juan Fernández and Desventuradas remain outside the Indo-Pacific Region (in contrast to Easter Island and Salas & Gómez, which fit within that region based on the data provided by Raines & Huber, 2012). The present study shows no bivalve species in common between Juan Fernández/Desventuradas and Eastern Island/Salas & Gómez, much in contrast to the reports for algae by Santelices & Meneses (2000) and Silva & Chacana (2005).

No sub-Antarctic or Magellanic bivalve species are found in Juan Fernández and Desventuradas. These data do not seem to be biased by limited knowledge; in fact, our (DGZ and MG) extensive prior work on the southern fauna (e.g., Zelaya, 2005, 2015; Güller & Zelaya, 2015) recorded none of the species here recognized for Juan Fernández and Desventuradas.

There is low similarity of the bivalve fauna of Juan Fernández and Desventuradas archipelagos with that of the Perú-Chile Province: only 10 bivalve species occurring in Desventuradas and Juan Fernández (20%) also occur at the South America mainland, a number further reduced if the material here tentatively identified as Perumytilus purpuratus and Bankia martensi turns out to be different species. This contrasts with the considerably higher similarity inferred by Bernard, McKinnell & Jamieson (1991): 95 and 68%, respectively, which arose from misidentifications as discussed above. The presence of a different fauna in Juan Fernández and Desventuradas compared to that of the Perú-Chile Province has been previously explained as a consequence of the biogeographic barrier imposed by the Humbolt current system (e.g., Rodríguez-Ruiz et al., 2014). However, the present study, as well as prior ones (e.g., Meneses & Hoffmann, 1994; Pequeño & Lamilla, 2000; Santelices & Meneses, 2000; Silva & Chacana, 2005; Retamal & Moyano, 2010) show that some Perú-Chile species have been able to surpass this current system and occur in the Juan Fernández/Desventuradas archipelagos. This may be explained by the frequent mesoscale eddies and meanders occurring in the area. These elements, which originate in central Chile, move westwards, from the coast to 600–800 km offshore, with a coherent spatial structure that could extend for several months (Hormazabal, Shaffer & Leth, 2004a; Hormazabal et al., 2004b). In the case of bivalves, these eddies and meanders may be contributing (facilitating or determining) the dispersal of certain larvae from the continent to Juan Fernández and Desventuradas. Also, at least some of this faunal exchange might be explained by human-mediated rafting.

Instead of having obvious affinities with other (close or distant) geographic areas, the bivalve fauna of Juan Fernández and Desventuradas is extremely peculiar, characterized by a high number of species only known from these archipelagos. This involves a total of 35 species (77% of the species studied herein), including 13 species only known from Desventuradas and 16 species only known from Juan Fernández. Bernard, McKinnell & Jamieson (1991) suggested that the high endemicity in the bivalve fauna of these archipelagos could be “merely a result of poor knowledge of the fauna of the adjacent mainland coast of South America”. However, the study of extant museum collections (Valentich-Scott, Coan & Zelaya, 2020) suggests that the lack of records of numerous Juan Fernández and Desventuradas species in the Perú-Chile Province is not an artifact. High percentages of endemic species in Juan Fernández and Desventuradas was also mentioned by some previous authors for algae and other invertebrate groups (e.g., Rozbaczylo & Castilla, 1987; Santelices & Meneses, 2000; Friedlander et al., 2016), with values reaching 80% in the case of echinoderms (Rodríguez-Ruiz et al., 2014).

The current findings give weight for considering Juan Fernández and Desventuradas as two distinctive biogeographic units, of either Province (e.g., Moyano, 1983; Parin, Mironov & Nesis, 1997; Spalding et al., 2007; Retamal & Moyano, 2010) or Ecoregion (Friedlander et al., 2016) rank.

The following new marine bivalve species are described herein, all by Zelaya, Güller & Bieler:

Tindaria sanfelixensis n. sp. (Tindariidae) urn:lsid:zoobank.org:act:F28DBF89-B0E9-408C-9450-4B6215F6CB2A.

Ledella costulata n. sp. (Nuculanidae).

urn:lsid:zoobank.org:act:BD240F8C-4278-4972-B8E4-8E724CC12E52.

Acar bernardi n. sp. (Arcidae) urn:lsid:zoobank.org:act:0B006933-BCDE-4838-AAE1-85BFB91EC6F6.

Anadara stempelli n. sp. (Arcidae) urn:lsid:zoobank.org:act:127CE134-89DA-43CD-9988-C1AEA962F679.

Tucetona sanfelixensis n. sp. (Glycymerididae) urn:lsid:zoobank.org:act:90A05C1E-C102-46B8-9684-22A3F9CB8540.

Verticipronus denticulatus n. sp. (Philobryidae) urn:lsid:zoobank.org:act:62D5FBC1-147C-4922-8813-0BC6F7CB9ECC.

Limaria crusoensis n. sp. (Limidae) urn:lsid:zoobank.org:act:C0E6AB15-FA72-458F-88B3-956064537288.

Limatula sanfelixensis n. sp. (Limidae) urn:lsid:zoobank.org:act:06000C16-F13A-46DE-94F0-4E2F7D58B545.

Cavilinga taylorgloverorum n. sp. (Lucinidae) urn:lsid:zoobank.org:act:EC3756F7-33D3-432C-A9E2-684D93729258.

Thyasira fernandezensis n. sp. (Thyasiridae) urn:lsid:zoobank.org:act:A2352CAC-BC12-41A1-BD40-1EABC335BF51.

Malvinasia selkirkensis n. sp. (Lasaeidae) urn:lsid:zoobank.org:act:D31B6960-44BB-4882-9484-C0D470D778C1.

Tellimya crusoensis n. sp. (Lasaeidae) urn:lsid:zoobank.org:act:52CEFE8D-2DF7-4BD9-8A30-72AA1ABE0271.

Condylocardia angusticostata n. sp. (Condylocardiidae) urn:lsid:zoobank.org:act:18F6E941-6BCA-4845-862A-024A81B03589.

Timoclea sanfelixensis n. sp. (Veneridae) urn:lsid:zoobank.org:act:BC6F6B71-CB46-446B-AFFF-EBA6918FBF81.

Neolepton sanfelixensis n. sp. (Neoleptonidae) urn:lsid:zoobank.org:act:06845FDE-3770-4538-82DA-83066A01A76E.

Cuspidaria fernandezensis n. sp. (Cuspidariidae) urn:lsid:zoobank.org:act:3468E275-DE8E-4DD0-AB65-A364959B81AB.

Cuspidaria sanfelixensis n. sp. (Cuspidariidae) urn:lsid:zoobank.org:act:76F959F8-1BFA-41D8-A2C4-EF30EB8892B9.

Halonympha recurvirostris n. sp. (Halonymphidae) urn:lsid:zoobank.org:act:FDBBB75B-992A-448A-86C2-73706A7607E1.

Pandora pyxis n. sp. (Pandoridae) urn:lsid:zoobank.org:act:65ABA641-3F86-46C3-B9E0-380ED8B0CA34.

Supplemental Information

Supplemental Information 1 Location and museum accession data for deposited expedition specimens.

RB thanks the organizers and team members of the IOC97 expedition, especially the captain and crew of the M/V Carlos Porter (who had us on board for 25 days in occasionally rough seas), Brian Dyer, Mark Westneat, and Melina Hale (who also acted as scuba team partners), and Sergio Letelier (for help with shore collecting and with sorting on board, and for hospitality extended during the post-cruise visit to the MNHN-CL in Santiago). Numerous colleagues kindly provided specimen loans or photographs, answered queries, and hosted our visits to their collections, including Gary Rosenberg (ANSP), Ted von Proschwitz (GNM), Tom White and Andreia Salvador (NHMUK), Lindsey Groves and Jann Vendetti (LACM), Jennifer Trimble (MCZ), Philippe Bouchet and Virginie Héros (MNHN), Jorge Artigas Coch (MZUC), Paul Valentich-Scott (Santa Barbara Museum of Natural History), Anders Warén (SMNH), Ellen Strong and Kathryn Ahlfeld (USNM), Matthias Glaubrecht, Frank Köhler, Thomas von Rintelen, and Christine Zorn (ZMB). Lisa Kanellos (FMNH) assisted with specimen imaging, Janeen Jones, Kalina Griffin-Jakymec, and John d’Angelo (FMNH) with specimen and data management, and Stephanie Ware (FMNH) with SEM support. Photographs of MCZ specimens are credited to the Museum of Comparative Zoology and Harvard University, those of NHMUK type specimens to Kevin Webb of the NHMUK Imaging Unit and the Trustees of the Natural History Museum, London. We greatly appreciate the constructive input by the four reviewers.

Additional Information and Declarations

Competing Interests

Author Contributions

Field Study Permissions

Data Availability

New Species Registration

Rüdiger Bieler is an Academic Editor for PeerJ.

Diego Gabriel Zelaya conceived and designed the experiments, performed the experiments, analyzed the data, prepared figures and/or tables, authored or reviewed drafts of the article, and approved the final draft.

Marina Güller conceived and designed the experiments, performed the experiments, analyzed the data, prepared figures and/or tables, authored or reviewed drafts of the article, and approved the final draft.

Rüdiger Bieler conceived and designed the experiments, performed the experiments, analyzed the data, prepared figures and/or tables, authored or reviewed drafts of the article, and approved the final draft.

The following information was supplied relating to field study approvals (i.e., approving body and any reference numbers):

The IOC97 expedition utilized a vessel of the Chilean Fisheries Department Instituto Fomento Pesquero (IFOP). The Chilean Navy (Armada de Chile) provided the permit to collect in the vicinity of, and disembark on, San Félix island. Chile’s Servicio Nacional de Pesca (SERNAPESCA) provided the associated permits for both island groups.

The following information was supplied regarding data availability:

This study is based on direct observations of morphological features, with the findings documented by textual descriptions and photographs. Types of new species and other specimens are deposited in the permanent collections of the Field Museum of Natural History in Chicago (FMNH), the Museo Nacional de Historia Natural de Chile (MNHN-CL) in Santiago, the Natural History Museum of Los Angeles County, and the National Museum of Natural History-Smithsonian Institution (USNM) in Washington, DC, with individual registration numbers provided in the text). Core taxonomic and collection-event data for IOC97 mollusks mentioned in this study are available online through FMNH’s institutional Invertebrate Zoology collections database at http://collections-zoology.fieldmuseum.org/.

The detailed location and accession data for the expedition material studied herein are available in the Supplemental File.

The following information was supplied regarding the registration of a newly described species:

Publication LSID: urn:lsid:zoobank.org:pub:571610DE-8F2D-4CB4-B527-8B999F6CB098.

Acar bernardi n. sp. (Arcidae): urn:lsid:zoobank.org:act:0B006933-BCDE-4838-AAE1-85BFB91EC6F6;

Anadara stempelli n. sp. (Arcidae): urn:lsid:zoobank.org:act:127CE134-89DA-43CD-9988-C1AEA962F679;

Cavilinga taylorgloverorum n. sp. (Lucinidae): urn:lsid:zoobank.org:act:EC3756F7-33D3-432C-A9E2-684D93729258;

Condylocardia angusticostata n. sp. (Condylocardiidae): urn:lsid:zoobank.org:act:18F6E941-6BCA-4845-862A-024A81B03589;

Cuspidaria fernandezensis n. sp. (Cuspidariidae): urn:lsid:zoobank.org:act:3468E275-DE8E-4DD0-AB65-A364959B81AB,

Cuspidaria sanfelixensis n. sp. (Cuspidariidae): urn:lsid:zoobank.org:act:76F959F8-1BFA-41D8-A2C4-EF30EB8892B9;

Halonympha recurvirostris n. sp. (Halonymphidae): urn:lsid:zoobank.org:act:FDBBB75B-992A-448A-86C2-73706A7607E1;

Ledella costulata n. sp. (Nuculanidae):

urn:lsid:zoobank.org:act:BD240F8C-4278-4972-B8E4-8E724CC12E52;

Limaria crusoensis n. sp. (Limidae): urn:lsid:zoobank.org:act:C0E6AB15-FA72-458F-88B3-956064537288;

Limatula sanfelixensis n. sp. (Limidae): urn:lsid:zoobank.org:act:06000C16-F13A-46DE-94F0-4E2F7D58B545;

Malvinasia selkirkensis n. sp. (Lasaeidae): urn:lsid:zoobank.org:act:D31B6960-44BB-4882-9484-C0D470D778C1;

Neolepton sanfelixensis n. sp. (Neoleptonidae): urn:lsid:zoobank.org:act:06845FDE-3770-4538-82DA-83066A01A76E;

Pandora pyxis n. sp. (Pandoridae): urn:lsid:zoobank.org:act:65ABA641-3F86-46C3-B9E0-380ED8B0CA34.

Tellimya crusoensis n. sp. (Lasaeidae): urn:lsid:zoobank.org:act:52CEFE8D-2DF7-4BD9-8A30-72AA1ABE0271;

Thyasira fernandezensis n. sp. (Thyasiridae): urn:lsid:zoobank.org:act:A2352CAC-BC12-41A1-BD40-1EABC335BF51;

Timoclea sanfelixensis n. sp. (Veneridae): urn:lsid:zoobank.org:act:BC6F6B71-CB46-446B-AFFF-EBA6918FBF81;

Tindaria sanfelixensis n. sp. (Tindariidae): urn:lsid:zoobank.org:act:F28DBF89-B0E9-408C-9450-4B6215F6CB2A;

Tucetona sanfelixensis n. sp. (Glycymerididae): urn:lsid:zoobank.org:act:90A05C1E-C102-46B8-9684-22A3F9CB8540;

Verticipronus denticulatus n. sp. (Philobryidae): urn:lsid:zoobank.org:act:62D5FBC1-147C-4922-8813-0BC6F7CB9ECC.

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
