# Peer review of "Doubling the known diversity of a remote island fauna: marine bivalves of the Juan Fernández and Desventuradas oceanic archipelagos (Southeastern Pacific Ocean)"

_PeerJ, doi:10.7717/peerj.17305_

## Round 0.1 · original submission · Minor Revisions

All four reviewers highlighted the importance of your work and the high scientific standard. I am also very impressed by your comprehensive and thorough analysis. Most comments of the reviewers address minor points and should be resolved quite easily. However, the introduction and the discussion seem to be very long with unnecessary repetitions, and should be condensed.

I look forward to receiving your revised manuscript.

·

Basic reporting

This is a signification contribution by qualified workers on the fauna of some little known island faunas. The ms is well prepared, thorough and well illustrated.

I've made few minor notes on the attached (I hope it can be) PDF.

Gene Coan

Experimental design

Reviewing museum specimens if an "experimental design"?

Validity of the findings

Very good

·

Basic reporting

Almost no comment, I only found some minor points outlined on the MS-pdf

Experimental design

no comment

Validity of the findings

no comment

Additional comments

The paper is a long-term, very complete and well-done survey on the bivalve fauna on those 2 remote oceanic islands. It brings important reviews, novelties, and nomenclatural acts of species. It is very well based conceptually, illustrated, and discussed. It really there is nothing to criticize. I only find some few minor points which I put as comments on the original annexed PDF.

Reviewer 3 ·

Basic reporting

The manuscript is well-crafted with a robust taxonomic foundation, aligning seamlessly with its primary objective. The work is the culmination of years of diligent effort and unwavering dedication. The authors meticulously explored various museum collections, shedding light on specimens from past expeditions.

Experimental design

ABSTRACT

It is repeated continuously in this study. Try to improve the writing so that you don't have to repeat the same sentence so much, as it is understood that a large part of the information presented belongs to the present research see lines 10, 13, and 20.

Line 10 : . This study, including new expedition material…. It is not clear if the new material was collected by the authors during a new field trip or if they are referred to material mentioned in line 13. I think do mean the samples collected during the expedition IOC-97 in 1997, right.

Line 13 : This study investigates material collected during…
Line 20 : This study provides information on the diversity…

The authors should make it clearer in the text that the new expedition material (Line 10) comes from previous expeditions carried out on the island and is not the result of recent fieldwork. I consider it important to clarify that there have been no new expeditions conducted on the island since the work of Friedlander et al. (2016), as this reinforces the idea mentioned by the authors in Line 99 and 100: the marine fauna of Juan Fernández and Desventuradas has been widely regarded as poorly known. Also, in Line 102, it is mentioned that additional data are needed, especially in view of the suggested complex faunal affinities.

INTRODUCTION

Line 44: Much work remains to be done and the faunas of…
Change by Much work remains to be done, and the faunas of…

Line 121 to re-evaluate
122 the endemicity and faunistic affinities of these areas. This is more a discussion about the faunal affinities of the island, because there is not a quantitative analysis

RESULTS
Line 2334 Marcel Velasques change by Marcel Velásquez.


CONCLUSION

Lines 2705 and 2706: Is this number the same (9 bivalves) if we exclude species with doubtful or impossible identification (e.g., Perumytilus purpuratus and Bankia martensi)?

Validity of the findings

This study highlights the shortcomings of traditional taxonomy and the limited research conducted in the archipelago. The review of old material and examination of previously overlooked specimens reveal a diversity of species that had remained hidden. Furthermore, this study illustrates how previous studies have repeatedly cited older works without re-evaluating the examined material, thereby perpetuating the inclusion of incorrectly identified species in the biodiversity inventories of the archipelago.

Additional comments

In my opinion, some minor adjustments are advisable.

·

Basic reporting

Please see below

Experimental design

Please see below

Validity of the findings

Please see below

Additional comments

PeerJ 93091

General comments

This is an incredibly thorough work with many high quality and detailed descriptons of previously known and new taxa.

The Abstract, Introduction, and Discussion are all far too long, containing rambling and repetitive information that strays from the main strength of the paper.

The species descriptions are well done, but require some revision.
- the same species names are used ad infinitum, and need to be changed
- several species are recognised as new but are not named, they must be either named, or the currently accepted names used instead.


Introduction

the paper is overly long; for example the introduction could start at line 56 without losing meaning

Specimen sources and station data

You must spell out the names of the museums at first instance, especially because MNHN is more recognised as the acronym for Museum National d'histoire Naturelle in Paris, as included in your table later at line 295

line 277 - "For the previously unnamed species, only empty shell material was collected"
there is a difference between collections of dead shells (usually eroded, e.g. from the strandline) or specimens that were live collected but only the shell retained. The condition of the specimens should be clarified. Line 284-287, this is a non issue, shorten or delete

Discussion

lines 2642-2645 these data need to be shown or omitted. If this point is relevant, the data need to be included in a supplement.

line 2678-2680 this is totally unnecessary speculation, these are bivalves with a biphasic lifecycle, they are mobile.

2689 - same point - there is no evidence to support human mediated movement, delete

2690-2691 - it is not clear why you would expect SE Pacific and IWP overlap, this point comes a bit at random; you need to refer to the prior work earlier to make the case that this should be considered

2736-2739 - as phrased this is not a conclusion. The difference could be explained elsehwere, perhaps in the introduction, but as you do not make a conclusion, it is actually immaterial. The conclusion should end with a novel statement from this study, not reference to other work.


ZooBank registration

2741- onward - these DO NOT go at the end, the LSID need to be included with the description of each species



Species descriptions

It is difficult for me to comment on these species descriptions which should be checked by a bivalve expert. The relevant experts are few but Gennady Kamadev has the best knowledge by far for this group. So I have added only some comments on minor corrections and notes on the new species

Tindaria sanfelixensis n. sp.

Give the ZooBank link to the species

Here and all other new species - if the new species is known only from the type series please make an explicit statement

Comparison is made with other spp on the sculpture - the type series contains a lot of specimens so this seems reliable. But shell sculpture changes ontogenetically so please make some explicity comment about size range / developmental stage for the material studies of the new sp. And is there any additional material, specimens not from the 2 lots designated as types?

Here and elsewhere, do not make comparative remarks dependent on external websites. If photos of museum specimens provide relevant comparisons, they should be included in the present ms, perhaps as a supplement. The museums will gladly give permission.

Ledella costulata n. sp.

Give the ZooBank link to the species
See note about websites above

line 513-514 - "non" not "not" (and any other instances throughout)

line 517 - Orae Chilensis et Peruvianae.-- can you add a note to explain this... is there any way to constrain this? or is it really equivalent to "SE Pacific"?

Amygdalum sp.

line 483 - this must be a typo, 14.9 mmm

I understand your reasoning for not committing to an identification but this is misleading and will cause confusion. The most appropriate treatment is to refer to it as A. americanum but include this note to emphasise that your specimen looks like A peasei (I agree, it does), and this is more evidence for a more cosmopolitan species and that americanum may be a junior synonym. If you leave it as "sp" it appears to be a *new* species and this is the opposite of your intention.



Anadara stempelli n. sp.

Are you sure it isn't a juvenile - this is not sufficiently justified. It is clear from what is presented that it is not K. platei, but the justification that it is not a juvenile of other previously described Andara needs more information especially if it is only known from a single specimen!


Tucetona sanfelixensis n. sp.

I know this is a different family but please use a different species epithet, if only to introduce some variety


Verticipronus denticulatus n. sp.

This is a very nice addition to the genus!

Monia sp. B

If you are going to name a species based on a single <3 mm long dead shell, above, then you can definitely give this one a name. You state it is a new species. Please complete the description.

Limaria crusoensis n. sp.

No comments, there is a lot of detail here

Limatula sanfelixensis n. sp.

Please see comment above about the species epithet - same applies here - please select a less repetitive species name

The justification for separation of the new sp from japonica is not convincing as written, please expand

Cavilinga taylorgloverorum n. sp.

On the species epithet - you have enough species to give John and Emily each their own species... they have worked on other families :-)

Thyasira fernandezensis n. sp.

There are also several fernandesensis / fernandezensis names already in play, more variety please (Emily might like this one)

What is the distribution range of Thyasira succisa? If there is overlap in features it is possible that these are either the same, and/or that previous studies have included unidentified specimens of this new sp.
Descriptions for Thyasira really do need anatomical features - the shells look so similar - if there is any fresh or dried material you must add whatever notes will help future identifications

Malvinasia selkirkensis n. sp.

`no comments

Tellimya crusoensis n. sp.`

No comments

Condylocardia angusticostata n. sp.

For info there is a new paper on Carditidae phylogeny forthcoming in Palaeontology, but they also skirted the issue of Condylocardia ;-)

Condylocardiidae sp. A

In the case of such tiny nondescript specimens I have no objection to not naming it. This is justified. Well done in having useful descriptive notes for future workers

Gari sp. B

This case is very marginal, whether the material warrants naming it. I would urge you to name it, because having unnamed "A" or "B" taxa will continue to propagate. Fixing a name even with imperfect type material is the only basis to allow future workers to identify and understand the species.

Gari is very symmetrical so a single valve is a sufficient basis for ID and description (as you have basically already demonstrated here)

Timoclea sanfelixensis n. sp.

Please see notes above about names, this is the 3rd T. sanfelixensis in the same paper.

Neolepton sanfelixensis n. sp.

I have made the point about species names before. This is the 5th sanfelixensis

The characters separating this species and N hupei are a bit marginal as described, it is difficult to judge without seeing comparative images

Neolepton sp. A

I agree this material would be a poor basis for a new species description. But given the poor quality of the material it would be more appropriate to rephrase the comments (line 2307) to say it "could" or "might" prove to represent a new species.

Xylophaga sp. A

I understand why you do not want to describe this herein but this species should be listed as X globosa as that is the current status quo, and you can leave the remarks listed in the comments as further explanatory information.


Cuspidaria fernandezensis n. sp.

This is the second fernandezensis n. sp. please pick a different name

Cuspidaria sanfelixensis n. sp. / Cuspidaria sanfelixensis n. sp.

Please, with the names.

These two species are very similar to each other - there are some differences in shell outline but this should be described more clearly. Also the material illustrated for the second species is quite poor, so the case for its distinctiveness should be made a bit stronger.

Halonympha recurvirostris n. sp.

Can you please give some more information about the genus / family diagnosis, to support placement in this species, since it is such a significant range extension for the genus

Pandora pyxis n. sp.

Explicitly state this is known only from the holotype (right?)

Etymology - it's a jar, not a box - there is a fun explanation in the book by Natalie Haynes
https://www.theguardian.com/books/2020/oct/13/pandoras-jar-by-natalie-haynes-review-ancient-misogyny

So a better name on this theme would be pithos rather than pyxis! See also Wikipedia, there are pictures

The specimen does have a very strange shape, it is convincingly new.


Entodesma sp.

Unfortunately this seems hardly worth including but if completeness is important, here it is

Julia Sigwart

---

## Round 0.2 · accepted · Accept

Thank you for the revision of the manuscript. I hereby certify that you have adequately taken into account the reviewers' comments and improved the manuscript accordingly. Based on my assessment as an Academic Editor, your manuscript is now ready for publication. I am very happy that you have selected PeerJ for the publication of this excellent study.